# Randomized Exploration in Cooperative Multi-Agent Reinforcement Learning

Hao-Lun Hsu,* Weixin Wang,* Miroslav Pajic, Pan Xu
Duke University
{hao-lun.hsu,weixin.wang,miroslav.pajic,pan.xu}@duke.edu

## Abstract

We present the first study on provably efficient randomized exploration in cooperative multi-agent reinforcement learning (MARL). We propose a unified algorithm framework for randomized exploration in parallel Markov Decision Processes (MDPs), and two Thompson Sampling (TS)-type algorithms, CoopTS-PHE and CoopTS-LMC, incorporating the perturbed-history exploration (PHE) strategy and the Langevin Monte Carlo exploration (LMC) strategy respectively, which are flexible in design and easy to implement in practice. For a special class of parallel MDPs where the transition is (approximately) linear, we theoretically prove that both CoopTS-PHE and CoopTS-LMC achieve a $\widetilde{\mathcal{O}}(d^{3/2}H^2\sqrt{MK})$ regret bound with communication complexity $\widetilde{\mathcal{O}}(dHM^2)$, where $d$ is the feature dimension, $H$ is the horizon length, $M$ is the number of agents, and $K$ is the number of episodes. This is the first theoretical result for randomized exploration in cooperative MARL. We evaluate our proposed method on multiple parallel RL environments, including a deep exploration problem (*i.e., $N$*-chain), a video game, and a real-world problem in energy systems. Our experimental results support that our framework can achieve better performance, even under conditions of misspecified transition models. Additionally, we establish a connection between our unified framework and the practical application of federated learning.

## 1 Introduction

Multi-Agent Reinforcement Learning (MARL) has emerged as a potent tool with wide-ranging applications in diverse fields including robotics [23, 54], gaming [74, 92, 84], and numerous real-world systems [10, 25, 85]. This is particularly evident in cooperative scenarios, where MARL's effectiveness is enhanced through both direct and indirect communication channels among agents. This requires MARL algorithms to adeptly and flexibly coordinate communications to optimize the benefits of cooperation. One of the classical challenges in MARL is balancing exploration and exploitation so that agents effectively utilize existing information while acquiring new knowledge. Recent literature highlights the intricacies of this balance, focusing on cooperative exploration strategies [27] and dynamic exploitation tactics [68]. Achieving this equilibrium is crucial for the practical deployment of MARL systems in real-world scenarios, where unpredictability and the need for rapid adaptation are prevalent [27, 14, 55].

Optimism in the Face of Uncertainty (OFU) is a popular strategy to address the exploration-exploitation problem [1]. OFU strategy leads to numerous upper confidence bound (UCB)-type algorithms in contextual bandits [18, 1, 50], single-agent reinforcement learning [36, 76], and more recently multi-agent reinforcement learning [24, 56]. These algorithms compute statistical confidence regions for the model or the value function, given the observed history, and perform the greedy

---

*Equal contribution.

38th Conference on Neural Information Processing Systems (NeurIPS 2024).

policy with respect to these regions, or upper confidence bounds. Though UCB-based methods give out strong theoretical results, they often have poor performance in practice [61, 60]. For example, Wang et al. [76] demonstrates that computing the confidence bonus necessitates advanced sensitivity sampling and the expensive computation makes the practical applications inefficient. It is worth noting that UCB is mostly constructed based on a linear structure [18, 36]. NeuralUCB is a notable attempt at a nonlinear version while it is infeasible in terms of computational complexity [94, 82].

Inspired by Thompson Sampling (TS) [73], posterior sampling for reinforcement learning (RL) [7, 95] involves maintaining a posterior distribution over the parameters of the Markov Decision Processes (MDP) model parameters. Although conceptually simple, most existing TS methods require the exact posterior or a good Laplacian approximation [83]. Recently, there have been advancements in randomized exploration with approximate sampling. One important method is perturb-history exploration (PHE) strategy, which involves introducing random perturbations in the action history of the agent [45, 47, 32]. This randomized exploration approach diversifies the agent's experience, aiding in learning more robust strategies in environments with uncertainty and variability. Another effective method is Langevin Monte Carlo (LMC) method [83, 33, 31, 42, 58, 34]. Notably, Ishfaq et al. [33] maintains the simplicity and scalability of LMC, making it applicable in deep RL algorithms by approximating the posterior distribution of the $Q$ function.

Despite the aforementioned advancements of randomized exploration in bandits and single-agent RL, there remains a scarcity of research on randomized exploration within cooperative MARL, which motivates us to present the first investigation into provably efficient randomized exploration in cooperative MARL, with both theoretical and empirical evidence. We specifically focus on the applicability in parallel MDPs, aiming to facilitate faster learning and to improve policy optimization with the same state and action spaces, allowing for leveraging similarities across MDPs. We theoretically and empirically demonstrate that randomized exploration strategies can be extended to the multi-agent setting and the benefit of randomized exploration instead of UCB can be significant from single-agent to multi-agent setting.

In summary, **our contributions** are as follows:

- We propose a unified algorithm framework for learning parallel MDPs, and apply two TS-related strategies PHE and LMC for exploration, which leads to the CoopTS-PHE and CoopTS-LMC algorithms. Unlike conventional TS, which suffers from sampling errors due to Laplace approximation and expensive posterior computation [66, 46], our proposed algorithms only require adding standard Gaussian noises to the dataset (CoopTS-PHE) or the gradient (CoopTS-LMC) when performing Least-Square Value Iteration, which is efficient in computation and avoids sampling bias due to the Laplace approximation. Notably, both algorithms are easily implementable which are more practical than UCB-based algorithms in deep MARL.
- When reduced to linear parallel MDPs, we theoretically prove that both CoopTS-PHE and CoopTS-LMC with linear function approximation can achieve a regret bound $\widetilde{\mathcal{O}}\big(d^{3/2}H^2\sqrt{M}\big(\sqrt{dM\gamma} + \sqrt{K}\big)\big)$ with communication complexity $\widetilde{\mathcal{O}}\big((d + K/\gamma)MH\big)$, where $d$ is the feature dimension, $H$ is the horizon length, $M$ is the number of agents, $K$ is the number of episodes for each agent, and $\gamma$ is a parameter controlling the communication frequency. When $\gamma = \mathcal{O}(K/dM)$, our algorithms attain $\widetilde{\mathcal{O}}\big(d^{3/2}H^2\sqrt{MK}\big)$ regret with $\widetilde{\mathcal{O}}(dHM^2)$ communication complexity. This result matches the best communication complexity in cooperative MARL [56], and the best regret bounds for randomized RL in the single-agent setting ($M = 1$) [32, 33]. A comprehensive comparison with baseline algorithms on episodic, non-sationary, linear MDPs is presented in Table 1.
- We further extend our theoretical analysis to the misspecified setting where both the transition and reward are approximately linear up to an error $\zeta$ and the MDPs could be heterogeneous across agents, which is a generalized notion of misspecification [36]. We theoretically prove when $\zeta = \mathcal{O}\big(\sqrt{d/MK}\big)$, the cumulative regret for CoopTS-PHE matches the result in the linear homogeneous MDP setting. Simultaneously, when $\zeta = \mathcal{O}\big(\sqrt{1/MK}\big)$, the cumulative regret for CoopTS-LMC matches the result in the linear homogeneous MDP setting. This result indicates that CoopTS-PHE has a slightly higher tolerance on the model misspecification than CoopTS-LMC.
- We conduct extensive experiments on various benchmarks with comprehensive ablation studies, including $N$-chain that requires deep exploration, Super Mario Bros task in a misspecified setting, and a real-world problem in thermal control of building energy systems. Our empirical evaluation demonstrates that our randomized exploration strategies outperform existing deep $Q$-network

Table 1: Comparison on episodic, non-stationary, linear MDPs. We define the average regret as the cumulative regret divided by the total number of samples (transition pairs) used by the algorithm. Here $d$ is the feature dimension, $H$ is the episode length, $K$ is the number of episodes, and $M$ is the number of agents in a multi-agent setting.

| Setting | Algorithm | Regret | Average Regret | Randomized Exploration | Generalizable to Deep RL | Communication Complexity |
|---------|-----------|--------|----------------|:---------------------:|:-----------------------:|:------------------------:|
| single-agent | OPT-RLSVI [88] | $\widetilde{\mathcal{O}}(d^2 H^{\frac{5}{2}}\sqrt{K})$ | $\widetilde{\mathcal{O}}(d^2 H^{\frac{3}{2}}\sqrt{1/K})$ | ✓ | ✗ | – |
| | LSVI-UCB [36] | $\widetilde{\mathcal{O}}(d^{\frac{3}{2}} H^2\sqrt{K})$ | $\widetilde{\mathcal{O}}(d^{\frac{3}{2}} H\sqrt{1/K})$ | ✗ | ✗ | – |
| | LSVI-PHE [32] | $\widetilde{\mathcal{O}}(d^{\frac{3}{2}} H^2\sqrt{K})$ | $\widetilde{\mathcal{O}}(d^{\frac{3}{2}} H\sqrt{1/K})$ | ✓ | ✓ | – |
| | LMC-LSVI [33] | $\widetilde{\mathcal{O}}(d^{\frac{3}{2}} H^2\sqrt{K})$ | $\widetilde{\mathcal{O}}(d^{\frac{3}{2}} H\sqrt{1/K})$ | ✓ | ✓ | – |
| | LSVI-ASE [34] | $\widetilde{\mathcal{O}}(d H^2\sqrt{K})$ | $\widetilde{\mathcal{O}}(d H\sqrt{1/K})$ | ✓ | ✓ | – |
| multi-agent | Coop-LSVI [24] | $\widetilde{\mathcal{O}}(d^{\frac{3}{2}} H^2\sqrt{MK})$ | $\widetilde{\mathcal{O}}(d^{\frac{3}{2}} H\sqrt{1/MK})$ | ✗ | ✗ | $dHM^3$ |
| | Asyn-LSVI [56] | $\widetilde{\mathcal{O}}(d^{\frac{3}{2}} H^2\sqrt{K})$ | $\widetilde{\mathcal{O}}(d^{\frac{3}{2}} H\sqrt{1/K})$ | ✗ | ✗ | $dHM^2$ |
| | **CoopTS-PHE (Ours)** | $\widetilde{\mathcal{O}}(d^{\frac{3}{2}} H^2\sqrt{MK})$ | $\widetilde{\mathcal{O}}(d^{\frac{3}{2}} H\sqrt{1/MK})$ | ✓ | ✓ | $dHM^2$ |
| | **CoopTS-LMC (Ours)** | $\widetilde{\mathcal{O}}(d^{\frac{3}{2}} H^2\sqrt{MK})$ | $\widetilde{\mathcal{O}}(d^{\frac{3}{2}} H\sqrt{1/MK})$ | ✓ | ✓ | $dHM^2$ |

(DQN)-based baselines. We also show that these strategies in cooperative MARL can be adapted to the existing federated RL framework when data transitions are not shared.

## 2 Preliminary

In parallel Markov Decision Processes (MDPs), $M$ agents interact independently with their respective discrete-time MDPs, sharing the same but independent state and action spaces. Each agent might have its unique reward functions and transition kernels. Specifically, for agent $m \in \mathcal{M}$, the associated MDP is defined by the tuple $\text{MDP}(\mathcal{S}, \mathcal{A}, H, \mathbb{P}_m, r_m)$. Here $\mathcal{S}$ and $\mathcal{A}$ are the state and action spaces respectively, $H$ is the horizon length, $\mathbb{P}_m = \{\mathbb{P}_{m,h}\}_{h\in[H]}$ and $r_m = \{r_{m,h}\}_{h\in[H]}$ are the sets of transition kernels and reward functions. For step $h \in [H]$, $\mathbb{P}_{m,h}(\cdot|s,a)$ is the probability measure over the next state given current state-action pair $(s,a)$, $r_{m,h} : \mathcal{S} \times \mathcal{A} \to [0,1]$ is the deterministic reward function. The policy $\pi_m = \{\pi_{m,h}\}_{h\in[H]}$ is a sequences of decision rules where $\pi_{m,h} : \mathcal{S} \to \mathcal{A}$ is the deterministic policy at step $h$.

For agent $m \in \mathcal{M}$, given any policy $\pi$ and transition $\mathbb{P}$, to evaluate the policy effectiveness in the $m^{\text{th}}$ MDP, we define value function $V^\pi_{m,h}(s) := \mathbb{E}_\pi[\sum_{h'=h}^H r_{m,h'}(s_{m,h'}, a_{m,h'})|s_{m,h} = s]$ and $Q$ function $Q^\pi_{m,h}(s,a) := \mathbb{E}_\pi[\sum_{h'=h}^H r_{m,h'}(s_{m,h'}, a_{m,h'})|s_{m,h} = s, a_{m,h} = a]$ for any $(h,s,a) \in [H] \times \mathcal{S} \times \mathcal{A}$. The optimal policy is defined as $\pi^*_m$, and we denote $V^*_{m,h}(s) = V^{\pi^*_m}_{m,h}(s)$. For each $k \in [K]$, at the beginning of episode $k$, each agent $m \in \mathcal{M}$ receives the initial state $s^k_{m,1}$ chosen arbitrarily by the environment. For each step $h \in [H]$ in this episode, each agent $m$ observes its current state $s^k_{m,h}$, selects an action $a^k_{m,h}$ based on policy $\pi^k_{m,h}$, receives a reward $r_{m,h}(s^k_{m,h}, a^k_{m,h})$, and then transitions to the next state $s^k_{m,h+1}$ based on the transition probability measure $\mathbb{P}_{m,h}(\cdot|s^k_{m,h}, a^k_{m,h})$. The reward defaults to 0 when the episode terminates at step $H + 1$. The goal of agents is to minimize the cumulative group regret after $K$ episodes, which is defined as

$$\text{Regret}(K) = \sum_{m\in\mathcal{M}} \sum_{k=1}^K \left[ V^*_{m,1}(s^k_{m,1}) - V^{\pi^k_m}_{m,1}(s^k_{m,1}) \right].$$

## 3 Algorithm Design

In this section, we first present a unified algorithm framework for conducting randomized exploration in cooperative MARL. Then we introduce two practical randomized exploration strategies.

### 3.1 Unified Algorithm Framework

A unified algorithm framework is presented in Algorithm 1, where each agent executes Least-Square Value Iteration (LSVI) in parallel and makes decisions based on collective data obtained from communication between each agent and the server. Before we describe the details of our algorithm, we first define notations about the datasets stored on each agent's local machine and the server.

---

**Algorithm 1** Unified Algorithm Framework for Randomized Exploration in Parallel MDPs

---

1:  Initialization: set $U_h^{\mathrm{ser}}(k), U_{m,h}^{\mathrm{loc}}(k) = \emptyset$.
2:  **for** episode $k = 1, ..., K$ **do**
3:     **for** agent $m \in \mathcal{M}$ **do**
4:        Receive initial state $s_{m,1}^k$.
5:        $V_{m,H+1}^k(\cdot) \leftarrow 0$.
6:        $\{Q_{m,h}^k(\cdot,\cdot)\}_{h=1}^H \leftarrow$ **Randomized Exploration**          ⊲ Algorithm 2 or Algorithm 3
7:        **for** step $h = 1, ..., H$ **do**
8:           $a_{m,h}^k \leftarrow \mathrm{argmax}_{a \in \mathcal{A}} Q_{m,h}^k(s_{m,h}^k, a)$.
9:           Receive $s_{m,h+1}^k$ and $r_{m,h}$.
10:          $U_{m,h}^{\mathrm{loc}}(k) \leftarrow U_{m,h}^{\mathrm{loc}}(k) \bigcup \left( s_{m,h}^k, a_{m,h}^k, s_{m,h+1}^k \right)$.
11:          **if** Condition **then**
12:             SYNCHRONIZE $\leftarrow$ True.
13:          **end if**
14:       **end for**
15:    **end for**
16:    **if** SYNCHRONIZE **then**
17:       **for** step $h = H, ..., 1$ **do**
18:          $\forall AGENT$: Send $U_{m,h}^{\mathrm{loc}}(k)$ to $SERVER$.
19:          $SERVER$: $U_h^{\mathrm{loc}}(k) \leftarrow \bigcup_{m \in \mathcal{M}} U_{m,h}^{\mathrm{loc}}(k)$.
20:          $SERVER$: $U_h^{\mathrm{ser}}(k) \leftarrow U_h^{\mathrm{ser}}(k) \bigcup U_h^{\mathrm{loc}}(k)$.
21:          $SERVER$: Send $U_h^{\mathrm{ser}}(k)$ to each $AGENT$.
22:          $\forall AGENT$: Set $U_{m,h}^{\mathrm{loc}}(k) \leftarrow \emptyset$.
23:       **end for**
24:    **end if**
25: **end for**

---

**Index notation** We define $k_s(k)$ (denoted as $k_s$ when no ambiguity arises) as the last episode before episode $k$ where synchronization happens. For episode $k$ and step $h$, we define three datasets:

$$U_h^{\mathrm{ser}}(k) = \left\{ \left( s_{n,h}^\tau, a_{n,h}^\tau, s_{n,h+1}^\tau \right) \right\}_{n \in \mathcal{M}, \tau \in [k_s]}, \tag{3.1a}$$

$$U_{m,h}^{\mathrm{loc}}(k) = \left\{ \left( s_{m,h}^\tau, a_{m,h}^\tau, s_{m,h+1}^\tau \right) \right\}_{\tau = k_s + 1}^{k-1}, \tag{3.1b}$$

$$U_{m,h}(k) = U_h^{\mathrm{ser}}(k) \bigcup U_{m,h}^{\mathrm{loc}}(k). \tag{3.1c}$$

By definition, $U_h^{\mathrm{ser}}(k)$ is the dataset that is shared across all agents due to the latest synchronization at episode $k_s$. $U_{m,h}^{\mathrm{loc}}(k)$ is the unique data collected by agent $m$ since episode $k_s$. Then $U_{m,h}(k)$ is the total dataset available for agent $m$ at the current time. Let $\mathcal{K}(k) = |U_{m,h}(k)|$ be the total number of data points. For the simplicity of notation, we also re-order the data points in $U_{m,h}(k)$, and rename the tuple $(s_{m,h}^\tau, a_{m,h}^\tau, s_{m,h+1}^\tau)$ as $(s^l, a^l, s'^l)$ such that we have $U_{m,h}(k) = \bigcup_{l=1}^{\mathcal{K}(k)} (s^l, a^l, s'^l)$. In fact, this can be done by the following one-to-one mapping

$$l_{m,k}(n, \tau) = \begin{cases} (\tau - 1)M + n & \tau \le k_s, \\ (M-1)k_s + \tau & k_s < \tau \le k - 1. \end{cases} \tag{3.2}$$

Therefore, we use indices $(s, a, s') \in U_{m,h}(k)$ and $l \in [\mathcal{K}(k)]$ interchangeably for the summation over set $U_{m,h}(k)$.

**Algorithm interpretation** At a high level, each episode $k$ in Algorithm 1 consists of two stages. The first stage (Lines 3-15) is parallelly executed by all agents and the second stage (Lines 16-24) involves the communication among agents and the server.

In the first stage (Lines 3-15) of Algorithm 1, each agent $m$ operates in two parts. The first part (Line 6) updates estimated $Q$ functions $\{Q_{m,h}^k\}_{h=1}^H$ through LSVI with a randomized exploration strategy (Algorithm 2 or Algorithm 3, which will be introduced in Section 3.2). In particular, given the estimated value functions $V_{m,h+1}^k(\cdot) = \max_{a \in A} Q_{m,h}^k(\cdot, a)$ at step $h + 1$, we perform one step

robust backward Bellman update to obtain $V_{m,h}^k(\cdot)$ at step $h$. And we initialize $V_{m,H+1}^k(\cdot)$ to be $0$ (Line 5). In the second part (Lines 7-14), after obtaining the estimated $Q$ functions, in each step $h$ we execute the greedy policy with respect to $Q_{m,h}^k$ and collect new data points which are added to the local dataset $U_{m,h}^{\mathrm{loc}}(k)$ (Lines 8-10). Then we verify the synchronization condition (Lines 11-13). In this paper, we mainly use three types of synchronization rules. (1) We can synchronize every $c$ episode where $c$ is a user-defined constant, which is easy to implement in practice. (2) We can also synchronize at the episode of $b^1, b^2, ..., b^n$, with $b$ representing the base of the exponential function. This is guided by the intuition that agents require more transitions urgently at the early learning stages. (3) Additionally, if we have a feature mapping $\phi(s, a) : \mathcal{S} \times \mathcal{A} \to \mathbb{R}^d$, based on (3.1), we define the following empirical covariance matrices.

$$^{\mathrm{ser}}\mathbf{\Lambda}_h^k = \sum_{(s^l, a^l, s'^l) \in U_h^{\mathrm{ser}}(k)} \phi(s^l, a^l) \phi(s^l, a^l)^\top,$$

$$^{\mathrm{loc}}\mathbf{\Lambda}_{m,h}^k = \sum_{(s^l, a^l, s'^l) \in U_{m,h}^{\mathrm{loc}}(k)} \phi(s^l, a^l) \phi(s^l, a^l)^\top,$$

$$\mathbf{\Lambda}_{m,h}^k = {}^{\mathrm{ser}}\mathbf{\Lambda}_h^k + {}^{\mathrm{loc}}\mathbf{\Lambda}_{m,h}^k + \lambda \mathbf{I}.$$

We synchronize as long as the following condition is met:

$$\log \frac{\det\left({}^{\mathrm{ser}}\mathbf{\Lambda}_h^k + {}^{\mathrm{loc}}\mathbf{\Lambda}_{m,h}^k + \lambda \mathbf{I}\right)}{\det\left({}^{\mathrm{ser}}\mathbf{\Lambda}_h^k + \lambda \mathbf{I}\right)} \geq \frac{\gamma}{(k - k_s)}, \tag{3.3}$$

where $\gamma$ is a communication control factor. In our experiments, we try all three rules and compare their performance, which is discussed in detail in Appendix K.1.

The second stage (Lines 16-24) is executed only when the synchronization condition is satisfied. First, all the agents upload their local transition set $U_{m,h}^{\mathrm{loc}}(k)$, i.e., the newly collected local data after the last synchronization, to the server. Then, the server gathers all information together in $U_h^{\mathrm{ser}}(k)$ and sends it back to each agent. Finally, each agent resets the local transition set $U_{m,h}^{\mathrm{loc}}(k) \leftarrow \emptyset$. Now agent $m$ can access the dataset $U_{m,h}(k) = U_h^{\mathrm{ser}}(k) \bigcup U_{m,h}^{\mathrm{loc}}(k)$, which contains the historical data of all agents up to last synchronization and its local dataset.

## 3.2 Randomized Exploration Strategies

When we update the model parameter and estimate $Q$ functions in Algorithm 1 (Line 6), we use exploration strategies to avoid suboptimal policies. Previous work adopted Upper Confidence Bound (UCB) exploration in the linear function class [24, 56] to estimate the $Q$ function $\{Q_{m,h}^k\}_{h=1}^H$. Although UCB-based methods come with strong theoretical guarantees, they often perform poorly in practice [16, 61, 60]. Moreover, UCB requires precise computation of the confidence set, which is usually hard to be implemented beyond the linear structure. In contrast, randomized exploration strategies offer more robust performance, flexibility in design, ease of implementation, and do not require a linear structure.

We approximate the $Q$ functions with the following function class $\mathcal{F} = \{f_\mathbf{w} : \mathcal{S} \times \mathcal{A} \to \mathbb{R} | f_\mathbf{w}(s, a) = f(\mathbf{w}; \phi(s, a))\}$, where $\mathbf{w} \in \mathbb{R}^d$ is the parameter and $\phi \in \mathbb{R}^d$ is a feature mapping associated with state-action pairs. Now we define the loss function for estimating the $Q$ functions.

$$L_{m,h}^k(\mathbf{w}) = \sum_{l=1}^{\mathcal{K}(k)} L\left(r_h^l + V_{m,h+1}^k(s'^l), f(\mathbf{w}; \phi^l)\right) + \lambda \|\mathbf{w}\|^2, \tag{3.4}$$

where $r_h^l = r_h(s^l, a^l)$, $\phi^l = \phi(s^l, a^l)$, and $L$ is a user-specified loss function.

**Perturbed-History Exploration** The first strategy we use in Algorithm 1 is called the perturbed-history exploration [45, 47, 32], displayed in Algorithm 2. We refer to the resulting algorithm as CoopTS-PHE. In particular, we optimize the following randomized loss function, where we add random Gaussian noises to the rewards and regularizer in (3.4).

$$\widetilde{L}_{m,h}^{k,n}(\mathbf{w}) = \sum_{l=1}^{\mathcal{K}(k)} L\left(\left(r_h^l + \epsilon_h^{k,l,n}\right) + V_{m,h+1}^k(s'^l), f(\mathbf{w}; \phi^l)\right) + \lambda \|\mathbf{w} + \boldsymbol{\xi}_h^{k,n}\|^2, \tag{3.5}$$

where $\epsilon_h^{k,l,n} \overset{\mathrm{i.i.d}}{\sim} \mathcal{N}(0, \sigma^2)$, $\boldsymbol{\xi}_h^{k,n} \sim \mathcal{N}(\mathbf{0}, \sigma^2 \mathbf{I})$, and $n \in [N]$. Then we obtain the following perturbed estimated parameter

$$\widetilde{\mathbf{w}}_{m,h}^{k,n} = \mathrm{argmin}_{\mathbf{w} \in \mathbb{R}^d} \widetilde{L}_{m,h}^{k,n}(\mathbf{w}). \tag{3.6}$$

Note that we repeat the above steps for $n = 1, \ldots, N$ to obtain independent copies of parameters, which is referred to as the multi-sampling process [32, 33]. Then we obtain the estimated $Q$ function $Q_{m,h}^k$ based on Line 7 in Algorithm 2. Finally, by maximizing $Q_{m,h}^k$ over action space $\mathcal{A}$, we obtain the estimated value function $V_{m,h}^k$.

---

**Algorithm 2** Perturbed-History Exploration

---
1: Input: multi-sampling number $N \in \mathbb{N}^+$, function class $\mathcal{F} = \{f_{\mathbf{w}} : \mathcal{S} \times \mathcal{A} \to \mathbb{R} | f_{\mathbf{w}}(s, a) = f(\mathbf{w}; \phi(s, a))\}$.
2: **for** step $h = H, \ldots, 1$ **do**
3:     **for** $n = 1, \ldots, N$ **do**
4:         Sample $\{\epsilon_h^{k,l,n}\}_{l \in [\mathcal{K}(k)]} \overset{\text{i.i.d}}{\sim} \mathcal{N}(0, \sigma^2)$ and $\boldsymbol{\xi}_h^{k,n} \sim \mathcal{N}(\mathbf{0}, \sigma^2 \mathbf{I})$ independently.
5:         Solve $\widetilde{\mathbf{w}}_{m,h}^{k,n}$ according to (3.6).
6:     **end for**
7:     $Q_{m,h}^k \leftarrow \min\{\max_{n \in [N]} f(\widetilde{\mathbf{w}}_{m,h}^{k,n}; \phi), H - h + 1\}^+$.
8:     $V_{m,h}^k(\cdot) \leftarrow \max_{a \in \mathcal{A}} Q_{m,h}^k(\cdot, a)$.
9: **end for**
10: Output: $\{Q_{m,h}^k(\cdot, \cdot), V_{m,h}^k(\cdot, \cdot)\}_{h=1}^H$.

---

**Langevin Monte Carlo Exploration**     Next we introduce the Langevin Monte Carlo exploration strategy [83, 33] in Algorithm 3, which stems from the Langevin dynamics [67, 8, 19, 81, 96]. Combining it with Algorithm 1 leads to our second proposed algorithm, CoopTS-LMC. Specifically, we update the model parameter iteratively. For iterate $j = 1, \ldots, J_k$, the update is given by

$$\mathbf{w}_{m,h}^{k,j,n} = \mathbf{w}_{m,h}^{k,j-1,n} - \eta_{m,k} \nabla L_{m,h}^k(\mathbf{w}_{m,h}^{k,j-1,n}) + \sqrt{2\eta_{m,k}\beta_{m,k}^{-1}} \boldsymbol{\epsilon}_{m,h}^{k,j,n}, \tag{3.7}$$

where $L_{m,h}^k$ is defined in (3.4), $\boldsymbol{\epsilon}_{m,h}^{k,j,n} \in \mathbb{R}^d$ is a standard Gaussian noise, $\eta_{m,k}$ is the learning rate, and $\beta_{m,k}$ is the inverse temperature parameter. We similarly use the multi-sampling trick to obtain $N$ independent estimators and estimate $Q$ function $Q_{m,h}^k$ by truncation based on Line 10 in Algorithm 3.

---

**Algorithm 3** Langevin Monte Carlo Exploration

---
1: Input: multi-sampling number $N \in \mathbb{N}^+$, function class $\mathcal{F} = \{f_{\mathbf{w}} : \mathcal{S} \times \mathcal{A} \to \mathbb{R} | f_{\mathbf{w}}(s, a) = f(\mathbf{w}; \phi(s, a))\}$, step sizes $\{\eta_{m,k}\}_{m \in \mathcal{M}, k \in [K]}$, inverse temperature parameters $\{\beta_{m,k}\}_{m \in \mathcal{M}, k \in [K]}$.
2: **for** step $h = H, \ldots, 1$ **do**
3:     **for** $n = 1, \ldots, N$ **do**
4:         $\mathbf{w}_{m,h}^{k,0,n} = \mathbf{w}_{m,h}^{k-1,J_{k-1},n}$.
5:         **for** $j = 1, \ldots, J_k$ **do**
6:             Sample $\boldsymbol{\epsilon}_{m,h}^{k,j,n} \overset{\text{i.i.d}}{\sim} \mathcal{N}(\mathbf{0}, \mathbf{I})$.
7:             Update $\mathbf{w}_{m,h}^{k,j,n}$ by (3.7).
8:         **end for**
9:     **end for**
10:     $Q_{m,h}^k \leftarrow \min\{\max_{n \in [N]} f(\mathbf{w}_{m,h}^{k,J_k,n}; \phi), H - h + 1\}^+$.
11:     $V_{m,h}^k(\cdot) \leftarrow \max_{a \in A} Q_{m,h}^k(\cdot, a)$.
12: **end for**
13: Output: $\{Q_{m,h}^k(\cdot, \cdot), V_{m,h}^k(\cdot, \cdot)\}_{h=1}^H$.

---

## 4 Theoretical Analysis

### 4.1 Homogeneous Parallel Linear MDPs

We provide theoretical analyses of our algorithms in the linear structure under the assumption of linear function approximation and linear MDP setting. We first present the definition of linear MDPs.

**Definition 4.1** (Linear MDP [36]). An MDP$(\mathcal{S}, \mathcal{A}, H, \mathbb{P}, r)$ is a linear MDP with feature map $\phi : \mathcal{S} \times \mathcal{A} \to \mathbb{R}^d$, if for any $h \in [H]$, there exist $d$ unknown measures $\boldsymbol{\mu}_h = (\mu_h^1, ..., \mu_h^d)$ over $\mathcal{S}$ and an unknown vector $\boldsymbol{\theta}_h \in \mathbb{R}^d$ such that for any $(s, a) \in \mathcal{S} \times \mathcal{A}$,

$$\mathbb{P}_h(\cdot | s, a) = \langle \boldsymbol{\phi}(s, a), \boldsymbol{\mu}_h(\cdot) \rangle, \quad r_h(s, a) = \langle \boldsymbol{\phi}(s, a), \boldsymbol{\theta}_h \rangle.$$

Without loss of generality, we assume that for all $(s, a) \in \mathcal{S} \times \mathcal{A}$, $\|\boldsymbol{\phi}(s, a)\| \leq 1$ and $\max\{\|\boldsymbol{\mu}_h(\mathcal{S})\|, \|\boldsymbol{\theta}_h\|\} \leq \sqrt{d}$.

Throughout the analyses in this section, we assume the homogeneous parallel MDPs setting where all agents share the same linear MDP defined in Definition 4.1. We also provide the results when the MDPs across agents are approximately linear and heterogeneous, which is deferred to Section 4.2 due to the space limit. Under the linear MDP assumption, it is known that the $Q$-function admits a linear form [36, Proposition 2.3]. Consequently, we choose the loss function $L$ in (3.4) to be the $l_2$ loss and approximate the $Q$ function in the linear function class $f(\mathbf{w}; \boldsymbol{\phi}^l) = \mathbf{w}^\top \boldsymbol{\phi}^l$.

Now we first present the regret bound for CoopTS-PHE.

**Theorem 4.2.** Under Definition 4.1, choose $L$ to be $l_2$ loss and linear function class $f(\mathbf{w}; \boldsymbol{\phi}^l) = \mathbf{w}^\top \boldsymbol{\phi}^l$ in (3.4). In CoopTS-PHE (Algorithm 1+Algorithm 2), let $N = \widetilde{C} \log(\delta) / \log(c_0)$ where $\widetilde{C} = \widetilde{\mathcal{O}}(d)$ and $c_0 = \Phi(1)$, $\Phi(\cdot)$ is the cumulative distribution function (CDF) of the standard normal distribution. Let $\lambda = 1$ and $0 < \delta < 1$. Under the determinant synchronization condition (3.3), we obtain the following cumulative regret

$$\mathrm{Regret}(K) = \widetilde{\mathcal{O}}\big(d^{\frac{3}{2}} H^2 \sqrt{M} \big(\sqrt{dM\gamma} + \sqrt{K}\big)\big),$$

with probability at least $1 - \delta$.

**Remark 4.3.** When we choose $\gamma = \mathcal{O}(K/dM)$ in the synchronization condition (3.3), the cumulative regret of CoopTS-PHE becomes $\widetilde{\mathcal{O}}(d^{3/2} H^2 \sqrt{MK})$, which matches the result of UCB exploration [24]. When $M = 1$, the regret becomes $\widetilde{\mathcal{O}}(d^{3/2} H^2 \sqrt{K})$, which matches the existing best randomized single-agent result [32, 33]. Note that if there is no communication at all and agents act independently, with the same number of learning rounds (or samples), the cumulative regret becomes $\widetilde{\mathcal{O}}(M \cdot d^{3/2} H^2 \sqrt{K})$. By incorporating communication, our regret bound in Theorem 4.2 is lower than that of the independent setting by a factor $\sqrt{M}$. A similar strategy called rare-switching update with a determinant synchronization condition has also been adopted in parallel bandit problems [69, 15].

Similarly, we have the following result for CoopTS-LMC.

**Theorem 4.4.** Under Definition 4.1, choose $L$ to be $l_2$ loss and linear function class $f(\mathbf{w}; \boldsymbol{\phi}^l) = \mathbf{w}^\top \boldsymbol{\phi}^l$ in (3.4). In CoopTS-LMC (Algorithm 1+Algorithm 3), let $N = \bar{C} \log(\delta) / \log(c_0')$ where $c_0' = 1 - 1/2\sqrt{2e\pi}$ and $\bar{C} = \widetilde{\mathcal{O}}(d)$. Let $1/\sqrt{\beta_{m,k}} = \widetilde{\mathcal{O}}(H\sqrt{d})$ for all $m \in \mathcal{M}$, $\lambda = 1$, and $0 < \delta < 1$. For any episode $k \in [K]$ and agent $m \in \mathcal{M}$, let the learning rate $\eta_{m,k} = 1/\big(4\lambda_{\max}\big(\boldsymbol{\Lambda}_{m,h}^k\big)\big)$, the update number $J_k = 2\kappa_k \log(4HKMd)$ where $\kappa_k = \lambda_{\max}\big(\boldsymbol{\Lambda}_{m,h}^k\big)/\lambda_{\min}\big(\boldsymbol{\Lambda}_{m,h}^k\big)$ is the condition number of $\boldsymbol{\Lambda}_{m,h}^k$. Under the determinant synchronization condition (3.3), we have

$$\mathrm{Regret}(K) = \widetilde{\mathcal{O}}\big(d^{\frac{3}{2}} H^2 \sqrt{M} \big(\sqrt{dM\gamma} + \sqrt{K}\big)\big),$$

with probability at least $1 - \delta$.

**Remark 4.5.** Note that CoopTS-PHE and CoopTS-LMC have the same order of regret. Hence the discussion in Remark 4.3 also applies to CoopTS-LMC. We would also like to highlight that our results are the first rigorous regret bounds for randomized MARL algorithms.

From the perspective of technical novelty, our analysis of randomized MARL algorithms is different from that of UCB-based algorithms [24] because the model prediction error here contains randomness, causing a more complex probability analysis and an additional approximation error. We would also like to point out that in proofs for both CoopTS-LMC and CoopTS-PHE we use a new $\varepsilon$-covering technique to prove that the optimism lemma holds for all $(s, a) \in \mathcal{S} \times \mathcal{A}$ instead of just the state-action pairs encountered by the algorithm, which is essential for the regret analysis. This was ignored by previous works [13] and its follow-up works [93, 33] that use the same regret decomposition technique. Furthermore, the multi-agent setting and the communications from synchronization in our algorithms also significantly increase the challenges in our analysis compared to randomized exploration in the single-agent setting [32, 33].

Next we present the communication complexity of Algorithm 1 with synchronization condition (3.3).

**Lemma 4.6.** The total number of communication rounds between the agents and the server in Algorithm 1 is bounded by $\mathrm{CPX} = \widetilde{\mathcal{O}}((d + K/\gamma)MH)$. Moreover, the total number of transferred random bits only has a logarithmic dependence on the number of episodes $K$.

**Remark 4.7.** We provide a refined analysis in Appendix C to get this improved result based on that of [24], which studied the same communication procedure as ours. When we choose $\gamma = \mathcal{O}(K/dM)$, the communication complexity reduces to $\widetilde{\mathcal{O}}(dHM^2)$, which only has a logarithmic dependence on the number of episodes $K$. Additionally, we provide a rigorous analysis to show that the algorithm only needs to communicate logarithm number of random bits throughout the learning process.

Note that Min et al. [56] studied the asynchronous setting where only one agent is active in each episode, giving out the regret $\widetilde{\mathcal{O}}(d^{3/2}H^2\sqrt{K})$ with the communication complexity $\widetilde{\mathcal{O}}(dHM^2)$. It is interesting to see that our algorithm, though in the synchronous setting, has the same communication complexity as the asynchronous variant. This implies that the asynchronous algorithm can only circumvent current communication by delaying it to the future but does not decrease the communication complexity. In fact, the synchronous setting can learn the policy better in our work, which is indicated by comparison of the average regret (the cumulative regret divided by the total number of samples used by the algorithm) in Table 1. By achieving a matched communication complexity, we find that synchronous and asynchronous settings have their own advantages and cannot replace each other. This phenomenon can help us better understand the properties of these two communication schemes.

## 4.2 Misspecified Setting

In this part, we extend our theoretical analysis to the misspecified setting. In this setting, the transition functions $\mathbb{P}_{m,h}$ and the reward functions $r_{m,h}$ are heterogeneous across different MDPs, which is slightly more complicated than the homogeneous setting. Moreover, instead of assuming the transition and reward are linear, we only require each individual MDP is a $\zeta$-approximate linear MDP [36] where both the transition and reward are approximately linear up to an controlled error $\zeta$.

**Definition 4.8** (Misspecified Parallel MDPs). For any $0 < \zeta \le 1$, and for any agent $m \in \mathcal{M}$, the corresponding $\mathrm{MDP}(\mathcal{S}, \mathcal{A}, H, \mathbb{P}_m, r_m)$ is a $\zeta$-approximate linear MDP with a feature map $\phi : \mathcal{S} \times \mathcal{A} \to \mathbb{R}^d$, for any $h \in [H]$, there exist $d$ unknown (signed) measures $\boldsymbol{\mu}_h = \left(\mu_h^{(1)}, \ldots, \mu_h^{(d)}\right)$ over $\mathcal{S}$ and an unknown vector $\boldsymbol{\theta}_h \in \mathbb{R}^d$ such that for any $(s,a) \in \mathcal{S} \times \mathcal{A}$, we have

$$\big\|\mathbb{P}_{m,h}(\cdot \mid s,a) - \big\langle\phi(s,a), \boldsymbol{\mu}_h(\cdot)\big\rangle\big\|_{\mathrm{TV}} \le \zeta,$$
$$\big|r_{m,h}(s,a) - \big\langle\phi(s,a), \boldsymbol{\theta}_h\big\rangle\big| \le \zeta,$$

where $\|\cdot\|_{\mathrm{TV}}$ is the total variation norm, for two distributions $P_1$ and $P_2$, we define it as: $\|P_1 - P_2\|_{\mathrm{TV}} = \frac{1}{2}\sum_{\mathbf{x}\in\Omega}|P_1(\mathbf{x}) - P_2(\mathbf{x})|$. Without loss of generality, we assume that $\|\phi(s,a)\| \le 1$ for all $(s,a) \in \mathcal{S} \times \mathcal{A}$, and $\max\big\{\|\boldsymbol{\mu}_h(\mathcal{S})\|, \|\boldsymbol{\theta}_h\|\big\} \le \sqrt{d}$ for all $h \in [H]$ and $m \in \mathcal{M}$.

**Remark 4.9.** Note that our misspecified setting defined in Definition 4.8 is a generalized notion of misspecification in [36]. Moreover, our misspecified setting is also more general and cover the small heterogeneous setting mentioned in [24]. The triangle inequality can easily be used to derive small heterogeneous setting from our misspecified setting, but not vice versa.

Next we state our regret bound for CoopTS-PHE in the misspecified setting.

**Theorem 4.10** (Misspecified Regret Bound for CoopTS-PHE). In CoopTS-PHE (Algorithm 1+Algorithm 2), under Definition 4.8 and determinant synchronization condition (3.3), with the same initialization with Theorem 4.2, we obtain the following cumulative regret

$$\mathrm{Regret}(K) = \widetilde{\mathcal{O}}\Big(d^{\frac{3}{2}}H^2\sqrt{M}\big(\sqrt{dM\gamma} + \sqrt{K}\big) + dH^2M\sqrt{K}\big(\sqrt{dM\gamma} + \sqrt{K}\big)\zeta\Big),$$

with probability at least $1 - \delta$.

**Remark 4.11.** When we choose $\zeta = \mathcal{O}\big(\sqrt{d/MK}\big)$, the cumulative regret becomes $\widetilde{\mathcal{O}}\big(d^{\frac{3}{2}}H^2\sqrt{M}\big(\sqrt{dM\gamma} + \sqrt{K}\big)\big)$. This matches the result of Theorem 4.2 in the linear MDP setting.

Similarly, we can have the following result for CoopTS-LMC.

**Theorem 4.12** (Misspecified Regret Bound for CoopTS-LMC). In CoopTS-LMC (Algorithm 1+Algorithm 3), under Definition 4.8 and determinant synchronization condition (3.3), with the same initialization with Theorem 4.4 except that $1/\sqrt{\beta_{m,k}} = \widetilde{\mathcal{O}}(H\sqrt{d} + H\sqrt{MK}d\zeta)$, we obtain the following cumulative regret

$$\text{Regret}(K) = \widetilde{\mathcal{O}}\Big(d^{\frac{3}{2}}H^2\sqrt{M}\big(\sqrt{dM\gamma} + \sqrt{K}\big) + d^{\frac{3}{2}}H^2M\sqrt{K}\big(\sqrt{dM\gamma} + \sqrt{K}\big)\zeta\Big),$$

with probability at least $1 - \delta$.

**Remark 4.13.** When $\zeta = \mathcal{O}\big(\sqrt{1/MK}\big)$, the cumulative regret becomes $\widetilde{\mathcal{O}}\big(d^{\frac{3}{2}}H^2\sqrt{M}\big(\sqrt{dM\gamma} + \sqrt{K}\big)\big)$. This matches the result of Theorem 4.4 in the linear MDP setting. By comparing Theorems 4.10 and 4.12, we find the result of CoopTS-LMC has an extra $\sqrt{d}$ factor worse than that of CoopTS-PHE, causing the chosen $\zeta$ in CoopTS-PHE has an extra $\sqrt{d}$ order over that in CoopTS-LMC. This indicates that CoopTS-PHE has better performance tolerance for the misspecified setting.

## 5 Experiments

In this section, we present an empirical evaluation of our proposed randomized exploration strategies (*i.e.,* CoopTS-PHE and CoopTS-LMC) with deep $Q$-networks (DQNs) [57] as the core algorithm on varying tasks under multi-agent settings compared with several baselines: vanilla DQN, Double DQN [28], Bootstrapped DQN [62], and Noisy-Net [26]). Given that all experiments are conducted under multi-agent settings unless explicitly specified as a single-agent or centralized scenario, we denote CoopTS-PHE as "PHE" and CoopTS-LMC as "LMC" in both experimental contexts and figures. Note that we run all our experiments on Nvidia RTX A5000 with 24GB RAM. The implementation of this work can be found at https://github.com/panxulab/MARL-CoopTS

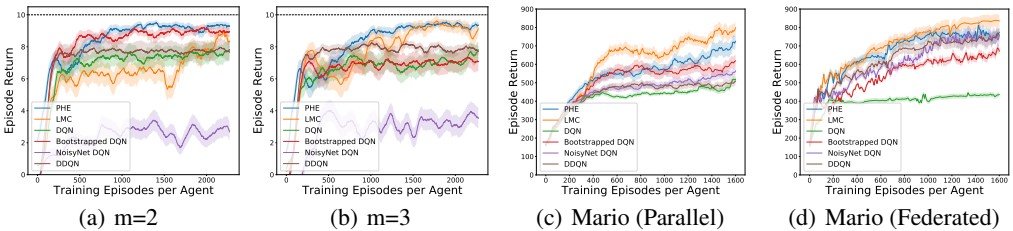

(a) m=2      (b) m=3      (c) Mario (Parallel)      (d) Mario (Federated)

Figure 1: Comparison among different exploration strategies in different environments. (a)-(b): $N$-chain with $N = 25$. (c)-(d): Super Mario Bros. All results are averaged over 10 runs and the shaded area represents the standard deviation.

### 5.1 $N$-chain

The $N$-chain [62] comprises a sequence of $N$ states denoted as $\{s_l\}_{l=1}^N$. Assuming the existence of $m$ agents, all initiating their trajectories from $s_2$, this study explores the dynamics of their movement within the chain. At each time step, agents face the decision to move either left or right. Notably, each agent incurs a nominal reward of $r = 0.001$ upon reaching state $s_1$, while a more substantial reward of $r = 1$ is obtained upon reaching the terminal state $s_N$. The illustration of $N$-chain environment is shown in Appendix K.1. With a horizon length of $N + 9$, the optimal return is 10. We consider $N = 25$ with the communication among agents in Figure 1 following the synchronization approach in Algorithm 1. In Figure 1(a), we show that PHE and Bootstrapped DQN result in higher average episode return among all agents while LMC can also eventually converge to a similar reward.

Upon increasing the number of agents to $m = 3$, we show in Figure 1(b) that our randomized exploration methods outperform all other baselines. Notably, the fluctuation in PHE is observed to be less pronounced against LMC. This observation lends support to our theoretical framework regarding performance tolerance in the misspecified setting, as detailed in Section 4.2. The complete results for $N$-chain and ablation studies can be found in Appendix K.1.

### 5.2 Super Mario Bros

Environmental heterogeneity, arising from various sources, is a prevalent challenge in practical scenarios. In Section 4.2, we illustrate the extension of homogeneous parallel MDP to the misspecified

setting. In the Super Mario Bros task [74], we examine a scenario where four agents, denoted as $m = 4$, engage in learning within distinct environments. Despite these environments sharing the same state space $\mathcal{S}$, action space $\mathcal{A}$, and reward function, their characteristics are different described in Appendix K.2. The primary objective of the Super Mario Bros task is to train an agent capable of advancing as far-right and rapidly as possible without collisions or falls. Utilizing preprocessed images as input states, agents aim to select optimal actions from a set of 7 discrete actions.

Figure 1(c) visually depicts that both randomized exploration strategies outperform other baselines in cooperative parallel learning. Notably, We observe that the superiority of LMC gets significant against PHE unlike the results in $N$-chain in Figures 1(a) and 1(b). In the case of PHE, Gaussian noise is introduced to the reward before applying the Bellman update, which can be viewed as a method empirically approximating the posterior distribution of the $Q$ function using a Gaussian distribution. However, it is crucial to note that in practical scenarios, unlike the $N$-chain setting, Gaussian distributions may not always provide an accurate approximation of the true posterior of the $Q$ function [33]. Here, transitions are shared among the four agents whenever the synchronization condition in (3.3) is met. We also conducted extra experiments in this task extending our proposed method to federated learning shown in Figure 1(d) with details in Appendix K.2.

### 5.3 Thermal Control of Building Energy Systems

Finally, we assess the efficacy of our randomized exploration strategies through their application to a practical task within a sustainable energy system: BuildingEnv, as outlined in [85]. BuildingEnv is designed to manage the heating supply in a multi-zone building, which involves addressing real-world physical constraints and accounting for environmental shifts over time. The objective is to meet user-defined temperature specifications while simultaneously minimizing overall electricity consumption. We defer the environment details to Appendix K.3.

With the availability of different cities in varying weather types, we conduct experiments on multiple cities in parallel and share their data following Algorithm 1 for each exploration strategy. During the evaluation, we deploy those trained policies to the environment of each city/weather respectively. We include all methods as well as random action in Figure 2 for a fair comparison. Specifically, we sample action randomly from action space for random action. We display the distribution of the return with probability density in violin plots, indicating that our PHE and LMC can perform better with a higher mean. Additional results for other cities can be found in Appendix K.3.

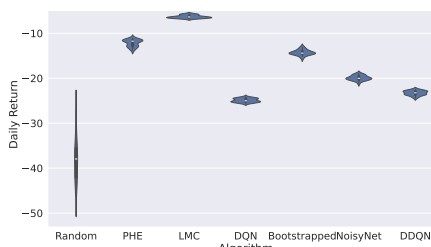

Figure 2: Evaluation performance at Tampa (hot humid) in building energy systems. All results are averaged over 10 runs.

## 6 Conclusion

We proposed a unified algorithm framework for provably efficient randomized exploration in parallel MDPs. By combining this unified algorithm framework with two TS-type randomized exploration strategies, PHE and LMC, we obtained two algorithms for parallel MDPs: CoopTS-PHE and CoopTS-LMC. These two algorithms are both flexible in design and easy to implement in practice. Under the linear MDP setting, we derived the theoretical regret bounds and communication complexities of CoopTS-PHE and CoopTS-LMC. This is the first result for randomized exploration in cooperative MARL, matching the best existing regret bounds for single-agent RL [32, 33]. We also extended our theoretical analysis to the misspecified setting. Our experiments on diverse RL parallel environments verified that randomized exploration improves the balance between exploration and exploitation in both homogeneous and heterogeneous settings. Future research directions includes extending our randomized exploration algorithm to fully decentralized or federated learning settings. Additionally, developing a more communication-efficient algorithm to reduce the substantial communication costs in the general function class setting is another potential direction.

## Acknowledgments

We would like to thank the anonymous reviewers for their helpful comments. HH and MP were supported in part by the ONR under agreement N00014-23-1-2206, AFOSR under the award number FA9550-19-1-0169, and by the NSF under NAIAD Award 2332744 as well as the National AI Institute for Edge Computing Leveraging Next Generation Wireless Networks, Grant CNS-2112562. WW and PX were supported in part by the National Science Foundation (DMS-2323112) and the Whitehead Scholars Program at the Duke University School of Medicine. The views and conclusions in this paper are those of the authors and should not be interpreted as representing any funding agency.

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

# A   Related Work

**Cooperative Multi-Agent Reinforcement Learning**   Cooperative MARL is closely intertwined with the domain of multi-agent multi-armed bandits, exemplified by decentralized algorithms featuring communication across a network or hypergraphs [48, 91, 41] and distributed settings [29, 77]. Cooperative MARL manifests primarily in two categories: multi-agent MDPs [12, 89, 79, 24] and parallel MDPs [11, 24, 53, 11, 56]. In the realm of cooperative multi-agent robotics, the former is employed to formulate optimal multi-agent policies across the distributed system [86, 87]. On the other hand, homogeneous parallel MDPs leverage inter-agent communication to expedite learning processes [43]. Additionally, heterogeneous parallel MDPs establish connections to heterogeneous federated learning [51] and exhibit improved generalizability in transfer learning scenarios [72].

We focus on parallel MDPs in this paper, where agents interact with the environment simultaneously to tackle shared challenges within extensive and distributed systems [43]. Recently, Dubey and Pentland [24] proposed the Coop-LSVI algorithm, extending the LSVI-UCB algorithm [36] in single-agent RL to MARL with linear MDPs. In a parallel RL setting with asynchronous communication, Min et al. [56] builds upon Coop-LSVI while relinquishing compatibility with heterogeneous MDPs. Meanwhile, Lidard et al. [53] focuses on fully decentralized multi-agent UCB $Q$-learning in a tabular setting, maintaining polynomial space complexity even as the number of agents increases. However, it is worth noting that neither of the previous works [24, 56] in non-tabular cooperative MRAL provides experimental validation for the efficacy of their proposed communication strategies. The gap arises from their reliance on LSVI-UCB as the core algorithm, wherein optimism is instantiated through UCB. Empirical evidence suggests that UCB-based approaches tend to underperform in practical scenarios [61, 60, 33]. Moreover, the computational demands of LSVI-UCB become untenable due to the necessity of recurrently computing the feature covariance matrix for updating the UCB bonus function. On the other hand, distributed applications of parallel MDPs in TS-based concurrent RL algorithms have been explored [21, 22, 17]. Specifically, Dimakopoulou and Roy [21] proposed a tabular model learning method based on seed sampling for coordinated exploration. This approach was further generalized to address intractable state spaces in [22] and supported by a **Bayesian** regret bound in [17]. However, none of these studies consider the communication complexity associated with efficient cooperative strategies. Therefore, randomized exploration in this work is critical to make these algorithm designs practical.

**Randomized Exploration**   The roots of randomized exploration, particularly TS, can be traced back to its success in bandit problems [73]. Randomized exploration strategies can typically exhibit superior performance in practical applications due to avoidance of early convergence to suboptimal actions [38–40]. Furthermore, these strategies demonstrate robustness in the face of noise and uncertainty, particularly within non-stationary environments [78, 9]. This success has extended to Langevin Monte Carlo Thompson Sampling (LMCTS), which has been applied to various domains, including linear bandits, generalized linear bandits, and neural contextual bandits [83]. The exploration of posterior sampling techniques in RL has gained prominence, building upon the foundation laid by TS [71, 7]. Randomized Least-Square Value Iteration (RLSVI) is an approach that leverages random perturbations to approximate the posterior, with frequentist regret analysis applied under the tabular MDP setting [63], inspiring subsequent works focusing on theoretical analyses aimed at improving worst-case regret under tabular MDPs [70, 4], with extensions to the linear setting [88, 32, 20]. In addition to theoretical advancements, several practical algorithms have been proposed based on RLSVI to approximate posterior samples of $Q$ functions in deep RL. These approaches involve ensembles of randomly initialized neural networks [62, 64] and noise injection into the parameters of the neural network [26, 52]. With the success of LMCTS [83] in bandit domains, the exploration of randomized methods has expanded to alternative approaches like LMC in tabular RL [42] and linear MDPs with neural network approximation [33]. Further works delve into the realm of random exploration from the perspectives of delayed feedback [44] and offline RL [59].

While posterior sampling demonstrates superiority in various contexts, its theoretical foundations in the multi-agent setting remain underexplored. Existing research predominantly focuses on two-player zero-sum games, considering both Bayesian [95, 35] and frequentist regrets [80, 65]. There is no existing work studying randomized exploration for cooperative multi-agent settings.

# B Instantiation of the Proposed Algorithms in the Linear Function Class

In this section, we specifically discuss our TS-related algorithms in the linear structure, which is under the assumption of linear function approximation and linear MDP setting.

Recall from the loss function in (3.4), here we choose $L$ to be $l_2$ loss and linear function class $f(\mathbf{w}; \boldsymbol{\phi}^l) = \mathbf{w}^\top \boldsymbol{\phi}^l$. By solving this least-square regression problem, we obtain the unperturbed regression estimator $\widehat{\mathbf{w}}_{m,h}^k$. In the linear setting, we have the closed-form solution

$$\widehat{\mathbf{w}}_{m,h}^k = (\boldsymbol{\Lambda}_{m,h}^k)^{-1} \boldsymbol{b}_{m,h}^k, \tag{B.1}$$

where $\boldsymbol{\Lambda}_{m,h}^k$ and $\boldsymbol{b}_{m,h}^k$ are defined as follows

$$\boldsymbol{\Lambda}_{m,h}^k = \sum_{l=1}^{\mathcal{K}(k)} \boldsymbol{\phi}(s^l, a^l) \boldsymbol{\phi}(s^l, a^l)^\top + \lambda \mathbf{I},$$

$$\boldsymbol{b}_{m,h}^k = \sum_{l=1}^{\mathcal{K}(k)} \big[ r_h(s^l, a^l) + V_{m,h+1}^k(s'^l) \big] \boldsymbol{\phi}(s^l, a^l).$$

A natural way of doing randomized exploration is to add a noise $\mathcal{N}(\mathbf{0}, \sigma^2 (\boldsymbol{\Lambda}_{m,h}^k)^{-1})$ to $\widehat{\mathbf{w}}_{m,h}^k$ and get the estimated parameter $\bar{\mathbf{w}}_{m,h}^k$. Then we can construct estimated $Q$ function $Q_{m,h}^k(\cdot, \cdot) = \min\{\boldsymbol{\phi}(\cdot, \cdot)^\top \bar{\mathbf{w}}_{m,h}^k, H - h + 1\}^+$. We call this method as CoopTS, which is aligned with other linear TS algorithms [6, 2]. In what follows, we theoretically show that our proposed algorithms are equivalent or approximately converge to the CoopTS algorithm in the linear function approximation setting.

For CoopTS-PHE (Algorithm 1+Algorithm 2), let the function approximation in (3.5) be linear and choose $L$ to be the squared loss. By solving this least-square regression problem, we obtain the perturbed regression estimator $\widetilde{\mathbf{w}}_{m,h}^{k,n}$ in CoopTS-PHE. The following proposition conveys that CoopTS-PHE is actually equivalent to CoopTS.

**Proposition B.1** (Equivalent to CoopTS). *The output $\widetilde{\mathbf{w}}_{m,h}^{k,n}$ by CoopTS-PHE is equivalent to adding a Gaussian vector to the unperturbed regression estimator $\widehat{\mathbf{w}}_{m,h}^k$, i.e., $\widetilde{\mathbf{w}}_{m,h}^{k,n} = \widehat{\mathbf{w}}_{m,h}^k + \boldsymbol{\zeta}_{m,h}^{k,n}$, where $\boldsymbol{\zeta}_{m,h}^{k,n} \sim \mathcal{N}(\mathbf{0}, \sigma^2 (\boldsymbol{\Lambda}_{m,h}^k)^{-1})$.*

For CoopTS-LMC (Algorithm 1+Algorithm 3), let function approximation in (3.4) be linear and choose $L$ to be $l_2$ loss to get the loss function. Then after finishing the LMC update, we get the estimated parameter $\mathbf{w}_{m,h}^{k,J_k,n}$ and construct the model approximation of $Q$ function. The following proposition conveys that the distribution of $\mathbf{w}_{m,h}^{k,J_k}$ converges to the posterior distribution of Thompson Sampling exploration. The proof of this proposition is given in [83].

**Proposition B.2** (Approximately equivalent to CoopTS [83]). *If the epoch length $J_k$ in Algorithm 3 is sufficiently large, the distribution of $\mathbf{w}_{m,h}^{k,J_k}$ converges to Gaussian distribution $\mathcal{N}(\widehat{\mathbf{w}}_{m,h}^k, \beta_{m,k}^{-1} (\boldsymbol{\Lambda}_{m,h}^k)^{-1})$.*

Propositions B.1 and B.2 indicate that the results of our two randomized exploration strategies are closely related to CoopTS. As we have mentioned above, in CoopTS, the estimated parameter $\bar{\mathbf{w}}_{m,h}^k$ is sampled from the normal distribution $\mathcal{N}(\widehat{\mathbf{w}}_{m,h}^k, \sigma^2 (\boldsymbol{\Lambda}_{m,h}^k)^{-1})$. However, in practice, this sampling is often executed in this way: we sample $\boldsymbol{\beta} \sim \mathcal{N}(\mathbf{0}, \mathbf{I})$ first, then we calculate $\bar{\mathbf{w}}_{m,h}^k = \widehat{\mathbf{w}}_{m,h}^k + \sigma (\boldsymbol{\Lambda}_{m,h}^k)^{-\frac{1}{2}} \boldsymbol{\beta}$ and obtain the estimated parameter. Nevertheless, computing $(\boldsymbol{\Lambda}_{m,h}^k)^{-\frac{1}{2}}$ can be computationally expensive, often requiring at least $\mathcal{O}(d^3)$ operations with the Cholesky decomposition, making it impractical for high-dimensional machine learning challenges. Additionally, the Gaussian distribution used in Thompson Sampling may not effectively approximate the posterior distribution in more complex bandit models than the linear MDP due to their intricate structures.

Moreover, as pointed out by recent work [16, 66, 46, 83], the Laplace approximation-based Thompson Sampling exhibits a constant approximation error in the estimation of the posterior distribution. Therefore, it necessitates a careful redesign of the covariance matrix to ensure effective performance.

**Advantages of PHE and LMC**   As mentioned above, computing $\left(\mathbf{\Lambda}_{m,h}^k\right)^{-\frac{1}{2}}$ can be computationally expensive. However, Perturbed-History exploration and Langevin Monte Carlo exploration can avoid this. For PHE, by only adding i.i.d random Gaussian noise to perturb reward and regularizer, its performance will be equivalent to TS. For LMC, by only performing noisy gradient descent, we can do the randomized exploration, resulting in similar performance compared with TS. Additionally, these two methods can easily be implemented to general function class while Thompson Sampling usually cannot be generalized except for the linear setting. In summary, these two methods are both flexible in design and easy to implement in practice.

**Communication cost**   We emphasize that agents can just send compressed statistics to the server under the linear setting, which can largely reduce communication cost. In the linear function class, we can calculate the closed-form solution of the regression problem (B.1). In this case, when synchronization process is met, all the agents will only need to send their calculated local statistics $^{\mathrm{loc}}\mathbf{\Lambda}_{m,h}^k$ and $^{\mathrm{loc}}\boldsymbol{b}_{m,h}^k$ to help solve the regression problem. This communication cost is much smaller because $\mathbf{\Lambda}$ is only a $d \times d$ matrix and $\boldsymbol{b}$ is only a $d$-dimensional vector, where $d$ is the feature dimension in linear MDP assumption. This can also avoid privacy disclosure through communications.

Nevertheless, in the general function class setting, our proposed algorithms still require sharing all the collected datasets, which will cause relatively large communication cost. Additionally, in Appendix K.2, we also propose a federated setting algorithm Algorithm 4. In this setting, instead of sharing collected datasets, agents can just share the weight of the collected estimated $Q$ functions, which can largely reduce the communication cost.

## C   Analysis of the Communication Complexity of Algorithm 1

The proof of the communication complexity is largely inspired by that in [24]. However, we provide a refined analysis here, and thus obtain an improved communication complexity $\widetilde{\mathcal{O}}(dHM^2)$, in contrast with the $\widetilde{\mathcal{O}}(dHM^3)$ complexity in their paper. We also discussed this in Remark 4.7 and showed that our result matches that of a recently proposed asynchronous algorithm. Moreover, we do a careful calculation of the total number of transferred random bits and show it only has a dependence on the number of episodes $K$.

*Proof of Lemma 4.6.*  We assume $\sigma = \{\sigma_1, \ldots, \sigma_n\}$ as the synchronization episodes, where $\sigma_i \in [K]$, we also denote $\sigma_0 = 0$. To bound the number of synchronization $n$, we separate $\sigma$ into two parts with an undetermined term $\alpha$

$$I_1 = \{i \in [n] | \sigma_i - \sigma_{i-1} \le \alpha\},$$
$$I_2 = \{i \in [n] | \sigma_i - \sigma_{i-1} > \alpha\}.$$

Then we have $n = |I_1| + |I_2|$. Note that

$$K \ge \sigma_n = \sum_{i=1}^{n} (\sigma_i - \sigma_{i-1}) \ge \sum_{i \in I_2} (\sigma_i - \sigma_{i-1}) > |I_2|\alpha.$$

Then we have $|I_2| < K/\alpha$. Then note that

$$
\begin{aligned}
\sum_{i=1}^{n} \log\left(\frac{\det(\mathbf{\Lambda}_{m,h}^{\sigma_i})}{\det(\mathbf{\Lambda}_{m,h}^{\sigma_{i-1}})}\right) &\ge \sum_{i \in I_1} \log\left(\frac{\det(\mathbf{\Lambda}_{m,h}^{\sigma_i})}{\det(\mathbf{\Lambda}_{m,h}^{\sigma_{i-1}})}\right) \\
&\ge \sum_{i \in I_1} \frac{\gamma}{\sigma_i - \sigma_{i-1}} \\
&\ge |I_1|\frac{\gamma}{\alpha}.
\end{aligned}
\tag{C.1}
$$

Define $\mathbf{\Lambda}_h^K = \sum_{m \in \mathcal{M}} \sum_{k=1}^{K} \phi(z_{m,h}^k) \phi(z_{m,h}^k)^\top + \lambda \mathbf{I}$ where $z_{m,h}^k = (s_{m,h}^k, a_{m,h}^k)$. On the other hand, we have

$$\sum_{i=1}^{n} \log\left(\frac{\det(\mathbf{\Lambda}_{m,h}^{\sigma_i})}{\det(\mathbf{\Lambda}_{m,h}^{\sigma_{i-1}})}\right) = \log\left(\frac{\det(\mathbf{\Lambda}_{m,h}^{\sigma_n})}{\det(\mathbf{\Lambda}_{m,h}^{\sigma_0})}\right)$$

$$\leq \log\left(\frac{\det(\boldsymbol{\Lambda}_h^K)}{\det(\lambda\mathbf{I})}\right)$$
$$\leq d\log(1 + MK/d), \tag{C.2}$$

where the first inequality holds due to the trivial fact that $\mathbf{A} \preccurlyeq \mathbf{B} \Rightarrow \det(\mathbf{A}) \leq \det(\mathbf{B})$, the second inequality follow from Lemma J.2 and the fact that $\|\boldsymbol{\phi}(\cdot)\|_2 \leq 1$. Combine (C.1) and (C.2), then we have $|I_1| \leq d\alpha/\gamma\log(1 + MK/d)$. Finally, we choose $\alpha = K/d$, then we have

$$n \leq \frac{K}{\alpha} + \frac{d\alpha}{\gamma}\log\left(1 + \frac{MK}{d}\right) = \left(d + \frac{K}{\gamma}\right)\log\left(1 + \frac{MK}{d}\right).$$

When one synchronization occurs, communications between agents and the server will occur $M$ times because we have $M$ agents in total. Recall from Lines 16-24 in Algorithm 1, also note that in one synchronization episode, communications will happen $H$ times between every agent and the server. Finally, the upper bound of communication complexity is

$$\text{CPX} = \widetilde{\mathcal{O}}\big((d + K/\gamma)MH\big).$$

Next we consider the total number of transferred random bits. We first calculate the communication bits per round. Under the linear setting, we can calculate the closed-form solution of the regression problem $\hat{\mathbf{w}}_{m,h}^k = (\boldsymbol{\Lambda}_{m,h}^k)^{-1}\boldsymbol{b}_{m,h}^k$, where

$$\boldsymbol{\Lambda}_{m,h}^k = \sum_{l=1}^{\mathcal{K}(k)} \boldsymbol{\phi}\big(s^l, a^l\big)\boldsymbol{\phi}\big(s^l, a^l\big)^\top + \lambda\mathbf{I},$$
$$\boldsymbol{b}_{m,h}^k = \sum_{l=1}^{\mathcal{K}(k)} [r_h\big(s^l, a^l\big) + V_{m,h+1}^k(s'^l)]\boldsymbol{\phi}\big(s^l, a^l\big).$$

Note that $l \in [\mathcal{K}(k)]$ is equivalent to $(s, a, s') \in U_{m,h}(k)$, and the index set $U_{m,h}(k)$ consists of $U_h^{\text{ser}}(k)$ and $U_{m,h}^{\text{loc}}(k)$. Therefore, the empirical covariance matrix $\boldsymbol{\Lambda}_{m,h}^k$ and the vector $\boldsymbol{b}_{m,h}^k$ can be decomposed into the summation of the local matrices and vectors on each agent. When the synchronization occurs, agents just need to send their local statistics $^{\text{loc}}\boldsymbol{\Lambda}_{m,h}^k$ and $^{\text{loc}}\boldsymbol{b}_{m,h}^k$ to the server to help solve the regression problem on each agent.

For the local empirical covariance matrix $^{\text{loc}}\boldsymbol{\Lambda}_{m,h}^k$

$$^{\text{loc}}\boldsymbol{\Lambda}_{m,h}^k = \sum_{(s^l, a^l, s'^l) \in U_{m,h}^{\text{loc}}(k)} \boldsymbol{\phi}\big(s^l, a^l\big)\boldsymbol{\phi}\big(s^l, a^l\big)^\top,$$

this is the summation of up to $K$ $d \times d$ matrices. Note that $\|\boldsymbol{\phi}(s, a)\| \leq 1$, thus it is easy to see that the entries of each matrix, namely, $\boldsymbol{\phi}\big(s^l, a^l\big)\boldsymbol{\phi}\big(s^l, a^l\big)^\top$, are bounded by 1. Therefore, the entries in $^{\text{loc}}\boldsymbol{\Lambda}_{m,h}^k$ are bounded by $K$. For each entry in this matrix, it suffices to use $\mathcal{O}(\log K)$ bits to communicate between the server and the agent. Thus in each round, $\mathcal{O}(d^2 \log K)$ bits are needed to send the matrix $^{\text{loc}}\boldsymbol{\Lambda}_{m,h}^k$.

For the local vector $^{\text{loc}}\boldsymbol{b}_{m,h}^k$

$$^{\text{loc}}\boldsymbol{b}_{m,h}^k = \sum_{(s^l, a^l, s'^l) \in U_{m,h}^{\text{loc}}(k)} [r_h\big(s^l, a^l\big) + V_{m,h+1}^k(s'^l)]\boldsymbol{\phi}\big(s^l, a^l\big),$$

this is a $d$-dimensional vector. Note that $r_h$ is bounded by 1, $V_{m,h+1}^k$ is bounded by $H$ and is linear with $\boldsymbol{\phi}$ by definition, which indicates we only need to communicate a $d$-dimensional vector $\bar{\mathbf{w}}_{m,h}^k$ to obtain $V_{m,h+1}^k$. Similar to the above analysis, in each round, $\mathcal{O}(d\log(K(H + 1)))$ bits are needed to send the vector $^{\text{loc}}\boldsymbol{b}_{m,h}^k$.

Therefore, the total bits of communication still only has a logarithmic dependency on the number of episodes $K$. This completes the proof. $\qquad\square$

# D  Proof of the Regret Bound for CoopTS-LMC

The general framework for CoopTS-LMC and CoopTS-PHE is closely similar. To make the article more concise, we first prove CoopTS-LMC completely, which is a bit more complicated. Then we can simplify the following similar proof for CoopTS-PHE in Appendix G.

## D.1 Supporting Lemmas

Before deriving the regret bound for CoopTS-LMC, we first provide the necessary technical lemmas for our regret analysis. Note that the loop (Line 3-9) in Algorithm 3 is to do multi-sampling for $N$ times. To simplify the notations, we eliminate the index $n$ before Lemma D.7 because the previous lemmas have nothing to do with multi-sampling.

**Definition D.1** (Model prediction error). For any $(m, k, h) \in \mathcal{M} \times [K] \times [H]$, we define the model error associated with the reward $r_h$,

$$l_{m,h}^k(s,a) = r_h(s,a) + \mathbb{P}_h V_{m,h+1}^k(s,a) - Q_{m,h}^k(s,a).$$

**Definition D.2** (Filtration). For any $(m, k, h) \in \mathcal{M} \times [K] \times [H]$, we define the filtration $\mathcal{F}_{m,k,h}$ as

$$\mathcal{F}_{m,k,h} = \sigma\Big(\big\{(s_{n,i}^\tau, a_{n,i}^\tau)\big\}_{(n,\tau,i) \in \mathcal{M} \times [k-1] \times [H]} \bigcup \big\{(s_{n,i}^k, a_{n,i}^k)\big\}_{(n,i) \in [m-1] \times [H]} \bigcup \big\{(s_{m,i}^k, a_{m,i}^k)\big\}_{i \in [h]}\Big).$$

**Proposition D.3.** In Algorithm 3, the parameter $\mathbf{w}_{m,h}^{k,J_k}$ satisfies the Gaussian distribution $\mathcal{N}\big(\boldsymbol{\mu}_{m,h}^{k,J_k}, \boldsymbol{\Sigma}_{m,h}^{k,J_k}\big)$, where mean vector and the covariance matrix are defined as

$$\boldsymbol{\mu}_{m,h}^{k,J_k} = \mathbf{A}_k^{J_k}...\mathbf{A}_1^{J_1}\mathbf{w}_{m,h}^{1,0} + \sum_{i=1}^k \mathbf{A}_k^{J_k}...\mathbf{A}_{i+1}^{J_{i+1}}\big(\mathbf{I} - \mathbf{A}_i^{J_i}\big)\widehat{\mathbf{w}}_{m,h}^i,$$

$$\boldsymbol{\Sigma}_{m,h}^{k,J_k} = \sum_{i=1}^k \frac{1}{\beta_{m,i}} \mathbf{A}_k^{J_k}...\mathbf{A}_{i+1}^{J_{i+1}}\big(\mathbf{I} - \mathbf{A}_i^{2J_i}\big)(\boldsymbol{\Lambda}_{m,h}^i)^{-1}(\mathbf{I} + \mathbf{A}_i)^{-1}\mathbf{A}_{i+1}^{J_{i+1}}...\mathbf{A}_k^{J_k},$$

where $\mathbf{A}_i = \mathbf{I} - 2\eta_{m,i}\boldsymbol{\Lambda}_{m,h}^i$ for $i \in [k]$.

**Lemma D.4.** For any $(m, k, h) \in \mathcal{M} \times [K] \times [H]$, the unperturbed estimated parameter $\widehat{\mathbf{w}}_{m,h}^k$ satisfies

$$\big\|\widehat{\mathbf{w}}_{m,h}^k\big\| \leq 2H\sqrt{Mkd/\lambda}.$$

**Lemma D.5.** Let $\lambda = 1$ in Algorithm 3. For any fixed $0 < \delta < 1$, with probability at least $1 - \delta^2$, for any $(m, k, h) \in \mathcal{M} \times [K] \times [H]$ and for any $(s, a) \in \mathcal{S} \times \mathcal{A}$, we have

$$\Big|\boldsymbol{\phi}(s,a)^\top \mathbf{w}_{m,h}^{k,J_k} - \boldsymbol{\phi}(s,a)^\top \widehat{\mathbf{w}}_{m,h}^k\Big| \leq \left(5\sqrt{\frac{2d\log(1/\delta)}{3\beta_K}} + \frac{4}{3}\right)\|\boldsymbol{\phi}(s,a)\|_{(\boldsymbol{\Lambda}_{m,h}^k)^{-1}}.$$

**Lemma D.6.** Let $\lambda = 1$ in Algorithm 3. For any fixed $0 < \delta < 1$, with probability at least $1 - \delta$, for any $(m, k, h) \in \mathcal{M} \times [K] \times [H]$, we have

$$\big\|\mathbf{w}_{m,h}^{k,J_k}\big\| \leq \frac{16}{3}Hd\sqrt{MK} + \sqrt{\frac{2K}{3\beta_K\delta}}d^{3/2} \overset{\text{def}}{=} B_\delta,$$

**Lemma D.7.** Let $\lambda = 1$ in Algorithm 3. For any fixed $0 < \delta < 1$, with probability at least $1 - \delta$, for all $(m, k, h) \in \mathcal{M} \times [K] \times [H]$, we have

$$\left\|\sum_{(s^l, a^l, s'^l) \in U_{m,h}(k)} \boldsymbol{\phi}(s^l, a^l)\big[\big(V_{m,h+1}^k - \mathbb{P}_h V_{m,h+1}^k\big)(s^l, a^l)\big]\right\|_{(\boldsymbol{\Lambda}_{m,h}^k)^{-1}} \leq 3H\sqrt{d}C_\delta,$$

where $C_\delta = \left[\frac{1}{2}\log(K+1) + \log\left(\frac{2\sqrt{2}KB_{\delta/2NMHK}}{H}\right) + \log\frac{3}{\delta}\right]^{1/2}$ and $B_\delta$ is defined in Lemma D.6.

**Lemma D.8.** Let $\lambda = 1$ in Algorithm 3. Under Definition 4.1, for any fixed $0 < \delta < 1$, with probability at least $1 - \delta$, for all $(m, k, h) \in \mathcal{M} \times [K] \times [H]$ and for any $(s, a) \in \mathcal{S} \times \mathcal{A}$, we have

$$\Big|\boldsymbol{\phi}(s,a)^\top \widehat{\mathbf{w}}_{m,h}^k - r_h(s,a) - \mathbb{P}_h V_{m,h+1}^k(s,a)\Big| \leq 5H\sqrt{d}C_\delta\|\boldsymbol{\phi}(s,a)\|_{(\boldsymbol{\Lambda}_{m,h}^k)^{-1}}.$$

**Lemma D.9** (Error bound). Let $\lambda = 1$ in Algorithm 3. Under Definition 4.1, for any fixed $0 < \delta < 1$, with probability at least $1 - \delta - \delta^2$, for any $(m, k, h) \in \mathcal{M} \times [K] \times [H]$ and for any $(s, a) \in \mathcal{S} \times \mathcal{A}$, we have

$$-l_{m,h}^k(s,a) \leq \left(5H\sqrt{d}C_\delta + 5\sqrt{\frac{2d\log\left(\sqrt{N}/\delta\right)}{3\beta_K}} + \frac{4}{3}\right)\|\boldsymbol{\phi}(s,a)\|_{(\boldsymbol{\Lambda}_{m,h}^k)^{-1}},$$

where $C_\delta$ is defined in Lemma D.7.

**Lemma D.10** (Optimism). Let $\lambda = 1$ in Algorithm 3 and $c_0' = 1 - \frac{1}{2\sqrt{2e\pi}}$. Under Definition 4.1, for any fixed $0 < \delta < 1$, with probability at least $1 - |\mathcal{C}(\varepsilon)|c_0'^N - 2\delta$ where $|\mathcal{C}(\varepsilon)| \leq (3/\varepsilon)^d$, for all $(m,h,k) \in \mathcal{M} \times [H] \times [K]$ and for all $(s,a) \in \mathcal{S} \times \mathcal{A}$, we have

$$l_{m,h}^k(s,a) \leq \alpha_\delta \varepsilon,$$

where $\alpha_\delta = \sqrt{MK}\big(2H\sqrt{d} + B_{\delta/NMHK}\big)$.

**Remark D.11.** Here we point out that in our proofs for both CoopTS-LMC and CoopTS-PHE, we use a new $\varepsilon$-covering technique to prove that the optimism lemma holds for all $(s,a) \in \mathcal{S} \times \mathcal{A}$ instead of just the state-action pairs encountered by the algorithm, which is essential in applying this lemma to bound the term $\mathbb{E}_{\pi^*}[l_{m,h}^k(s_{m,h}, a_{m,h})|s_{m,1} = s_{m,1}^k]$ in (D.2) in the regret analysis. This was ignored by previous works [13, 33] that use the same regret decomposition technique in the single-agent setting.

The following lemma gives the upper bound of self-normalized term summation in the multi-agent setting, which is first introduced by Lemma 9 in [24]. To make our analysis complete, we give out the proof in the Appendix E.9 where we make some necessary modifications compared with Lemma 9 in [24].

**Lemma D.12.** Let Algorithm 2 run for any $K > 0$, $M \geq 1$, and $\gamma$ as the communication control factor. Define $\mathbf{\Lambda}_h^K = \sum_{m\in\mathcal{M}} \sum_{k=1}^K \phi\big(s_{m,h}^k, a_{m,h}^k\big)\phi\big(s_{m,h}^k, a_{m,h}^k\big)^\top + \lambda\mathbf{I}$, then we have

$$\sum_{m\in\mathcal{M}} \sum_{k=1}^K \big\|\phi(s_{m,h}^k, a_{m,h}^k)\big\|_{(\mathbf{\Lambda}_{m,h}^k)^{-1}} \leq \left(\log\left(\frac{\det(\mathbf{\Lambda}_h^K)}{\det(\lambda\mathbf{I})}\right) + 1\right)M\sqrt{\gamma} + 2\sqrt{MK\log\left(\frac{\det(\mathbf{\Lambda}_h^K)}{\det(\lambda\mathbf{I})}\right)}.$$

The following lemma shows that we can decompose the regret of Algorithm 2 into three different components. The proof of this lemma closely resembles Lemma 4.2 in [13] for the single-agent setting. When we fix the agent $m \in \mathcal{M}$, it is totally same as Lemma 4.2 in [13].

**Lemma D.13.** [13, Lemma 4.2] Define the operators and the following terms:

$$(\mathbb{J}_{m,h}f)(s) = \big\langle f(s,\cdot), \pi_{m,h}^*(\cdot|s)\big\rangle, \quad (\mathbb{J}_{m,k,h}f)(s) = \big\langle f(s,\cdot), \pi_{m,h}^k(\cdot|s)\big\rangle,$$
$$D_{m,k,h,1} = \big(\mathbb{J}_{m,k,h}(Q_{m,h}^k - Q_{m,h}^{\pi_{m,k}})\big)(s_{m,h}^k) - \big(Q_{m,h}^k - Q_{m,h}^{\pi_{m,k}}\big)(s_{m,h}^k, a_{m,h}^k), \qquad (\text{D.1})$$
$$D_{m,k,h,2} = \big(\mathbb{P}_{m,h}(V_{m,h+1}^k - V_{m,h+1}^{\pi_{m,k}})\big)(s_{m,h}^k, a_{m,h}^k) - \big(V_{m,h+1}^k - V_{m,h+1}^{\pi_{m,k}}\big)(s_{m,h+1}^k).$$

Then we can decompose the regret into the following form:

$$\text{Regret}(K) = \sum_{m\in\mathcal{M}} \sum_{k=1}^K V_{m,1}^*\big(s_{m,1}^k\big) - V_{m,1}^{\pi_m^k}\big(s_{m,1}^k\big)$$

$$= \underbrace{\sum_{m\in\mathcal{M}} \sum_{k=1}^K \sum_{h=1}^H \mathbb{E}_{\pi^*}\big[\big\langle Q_{m,h}^k(s_{m,h}, \cdot), \pi_{m,h}^*(\cdot,|s_{m,h}) - \pi_{m,h}^k(\cdot|s_{m,h})\big\rangle|s_{m,1} = s_{m,1}^k\big]}_{(i)}$$

$$+ \underbrace{\sum_{m\in\mathcal{M}} \sum_{k=1}^K \sum_{h=1}^H (D_{m,k,h,1} + D_{m,k,h,2})}_{(ii)}$$

$$+ \underbrace{\sum_{m\in\mathcal{M}} \sum_{k=1}^K \sum_{h=1}^H \big(\mathbb{E}_{\pi^*}\big[l_{m,h}^k(s_{m,h}, a_{m,h})|s_{m,1} = s_{m,1}^k\big] - l_{m,h}^k\big(s_{m,h}^k, a_{m,h}^k\big)\big)}_{(iii)}.$$

## D.2 Regret Analysis

In this part, we give out the proof of Theorem 4.4, the regret bound for CoopTS-LMC.

*Proof of Theorem 4.4.* Based on the result from Lemma D.13, we do the regret decomposition first

$$
\begin{aligned}
\text{Regret}(K) &= \sum_{m \in \mathcal{M}} \sum_{k=1}^{K} V_{m,1}^{*}\big(s_{m,1}^{k}\big) - V_{m,1}^{\pi_{m}^{k}}\big(s_{m,1}^{k}\big) \\
&= \underbrace{\sum_{m \in \mathcal{M}} \sum_{k=1}^{K} \sum_{h=1}^{H} \mathbb{E}_{\pi^{*}}\big[\big\langle Q_{m,h}^{k}(s_{m,h}, \cdot), \pi_{m,h}^{*}(\cdot | s_{m,h}) - \pi_{m,h}^{k}(\cdot | s_{m,h}) \big\rangle | s_{m,1} = s_{m,1}^{k}\big]}_{(i)} \\
&\quad + \underbrace{\sum_{m \in \mathcal{M}} \sum_{k=1}^{K} \sum_{h=1}^{H} (D_{m,k,h,1} + D_{m,k,h,2})}_{(ii)} \\
&\quad + \underbrace{\sum_{m \in \mathcal{M}} \sum_{k=1}^{K} \sum_{h=1}^{H} \big(\mathbb{E}_{\pi^{*}}\big[l_{m,h}^{k}(s_{m,h}, a_{m,h}) | s_{m,1} = s_{m,1}^{k}\big] - l_{m,h}^{k}\big(s_{m,h}^{k}, a_{m,h}^{k}\big)\big)}_{(iii)} .
\end{aligned}
\tag{D.2}
$$

Next, we will bound the above three terms respectively.

**Bounding Term (i) in** (D.2)**:** for the policy $\pi_{m,h}^{k}$, we have

$$
\sum_{m \in \mathcal{M}} \sum_{k=1}^{K} \sum_{h=1}^{H} \mathbb{E}_{\pi^{*}}\big[\big\langle Q_{m,h}^{k}(s_{m,h}, \cdot), \pi_{m,h}^{*}(\cdot | s_{m,h}) - \pi_{m,h}^{k}(\cdot | s_{m,h}) \big\rangle | s_{m,1} = s_{m,1}^{k}\big] \leq 0. \tag{D.3}
$$

This is because by definition $\pi_{m,h}^{k}$ is the greedy policy for $Q_{m,h}^{k}$.

**Bounding Term (ii) in** (D.2)**:** note that $0 \leq Q_{m,h}^{k} \leq H - h + 1 \leq H$, based on (D.1), for any $(m, k, h) \in \mathcal{M} \times [K] \times [H]$, we have $|D_{m,k,h,1}| \leq 2H$ and $|D_{m,k,h,2}| \leq 2H$. Note that $D_{m,k,h,1}$ is a martingale difference sequence $\mathbb{E}[D_{m,k,h,1} | \mathcal{F}_{m,k,h}] = 0$. By applying Azuma-Hoeffding inequality, with probability at least $1 - \delta/3$, we have

$$
\sum_{m \in \mathcal{M}} \sum_{k=1}^{K} \sum_{h=1}^{H} D_{m,k,h,1} \leq 2\sqrt{2MH^{3}K \log(6/\delta)}.
$$

Note that $D_{m,k,h,2}$ is also a martingale difference sequence. By applying Azuma-Hoeffding inequality, with probability at least $1 - \delta/3$, we have

$$
\sum_{m \in \mathcal{M}} \sum_{k=1}^{K} \sum_{h=1}^{H} D_{m,k,h,2} \leq 2\sqrt{2MH^{3}K \log(6/\delta)}.
$$

By taking union bound, with probability at least $1 - 2\delta/3$, we have

$$
\sum_{m \in \mathcal{M}} \sum_{k=1}^{K} \sum_{h=1}^{H} D_{m,k,h,1} + \sum_{m \in \mathcal{M}} \sum_{k=1}^{K} \sum_{h=1}^{H} D_{m,k,h,2} \leq 4\sqrt{2MH^{3}K \log(6/\delta)}. \tag{D.4}
$$

**Bounding Term (iii) in** (D.2)**:** based on Lemmas D.9 and D.10, by taking union bound, with probability at least $1 - |\mathcal{C}(\varepsilon)|c_{0}'^{N} - 2\delta' - MHK(\delta' + \delta'^{2})$, we have

$$
\begin{aligned}
&\sum_{m \in \mathcal{M}} \sum_{k=1}^{K} \sum_{h=1}^{H} \big(\mathbb{E}_{\pi^{*}}\big[l_{m,h}^{k}(s_{m,h}, a_{m,h}) | s_{m,1} = s_{m,1}^{k}\big] - l_{m,h}^{k}\big(s_{m,h}^{k}, a_{m,h}^{k}\big)\big) \\
&\leq \sum_{m \in \mathcal{M}} \sum_{k=1}^{K} \sum_{h=1}^{H} \big(\alpha_{\delta'}\varepsilon - l_{m,h}^{k}\big(s_{m,h}^{k}, a_{m,h}^{k}\big)\big) \\
&\leq HMK\alpha_{\delta'}\varepsilon + \sum_{m \in \mathcal{M}} \sum_{k=1}^{K} \sum_{h=1}^{H} \left(5H\sqrt{d}C_{\delta'} + 5\sqrt{\frac{2d \log(\sqrt{N}/\delta')}{3\beta_{K}}} + \frac{4}{3}\right) \big\|\phi(s_{m,h}^{k}, a_{m,h}^{k})\big\|_{(\mathbf{\Lambda}_{m,h}^{k})^{-1}}
\end{aligned}
$$

$$= HMK\alpha_{\delta'}\varepsilon + \left(5H\sqrt{d}C_{\delta'} + 5\sqrt{\frac{2d\log(\sqrt{N}/\delta')}{3\beta_K}} + \frac{4}{3}\right)\sum_{h=1}^{H}\sum_{m\in\mathcal{M}}\sum_{k=1}^{K}\left\|\phi(s_{m,h}^k, a_{m,h}^k)\right\|_{(\mathbf{\Lambda}_{m,h}^k)^{-1}}$$

$$\leq HMK\alpha_{\delta'}\varepsilon + \left(5H\sqrt{d}C_{\delta'} + 5\sqrt{\frac{2d\log(\sqrt{N}/\delta')}{3\beta_K}} + \frac{4}{3}\right)$$

$$\times \sum_{h=1}^{H}\left(\log\left(\frac{\det(\mathbf{\Lambda}_h^K)}{\det(\lambda\mathbf{I})}\right) + 1\right)M\sqrt{\gamma} + 2\sqrt{MK\log\left(\frac{\det(\mathbf{\Lambda}_h^K)}{\det(\lambda\mathbf{I})}\right)}$$

$$\leq HMK\alpha_{\delta'}\varepsilon + \left(5H\sqrt{d}C_{\delta'} + 5\sqrt{\frac{2d\log(\sqrt{N}/\delta')}{3\beta_K}} + \frac{4}{3}\right)$$

$$\times H\Big(d(\log(1 + MK/d) + 1)M\sqrt{\gamma} + 2\sqrt{MKd\log(1 + MK/d)}\Big).$$

The first inequality follows from Lemma D.10, the second inequality follows from Lemma D.9, the third inequality follows from Lemma D.12, the last inequality holds due to Lemma J.2 and the fact that $\|\phi(\cdot)\|_2 \leq 1$.

Here we choose $\varepsilon = dH\sqrt{d/MK}/\alpha_{\delta'} = \widetilde{\mathcal{O}}(\sqrt{1/dHM^3K^4N})$ and choose $\frac{1}{\sqrt{\beta_K}} = 20H\sqrt{d}C_{\delta'} + \frac{16}{3}$, we have

$$\sum_{m\in\mathcal{M}}\sum_{k=1}^{K}\sum_{h=1}^{H}\big(\mathbb{E}_{\pi^*}\big[l_{m,h}^k(s_{m,h}, a_{m,h})|s_{m,1} = s_{m,1}^k\big] - l_{m,h}^k\big(s_{m,h}^k, a_{m,h}^k\big)\big) \leq \widetilde{\mathcal{O}}\big(dH^2\big(dM\sqrt{\gamma} + \sqrt{dMK}\big)\big),$$
(D.5)

occurs with probability at least $1 - |\mathcal{C}(\varepsilon)|c_0'^N - 2\delta' - MHK(\delta' + \delta'^2)$.

We set $\delta' = \delta/12(MHK + 1)$ and choose $N = \bar{C}\log(\delta)/\log(c_0')$ where $\bar{C} = \widetilde{\mathcal{O}}(d)$, then we have

$$1 - |\mathcal{C}(\varepsilon)|c_0'^N - 2\delta' - MHK(\delta' + \delta'^2) \geq 1 - \delta/3.$$

**Combining Terms (i)(ii)(iii) together:** Based on (D.3), (D.4) and (D.5). By taking union bound, we get that the final regret bound for CoopTS-LMC is $\widetilde{\mathcal{O}}\big(dH^2\big(dM\sqrt{\gamma} + \sqrt{dMK}\big)\big)$ with probability at least $1 - \delta$. □

# E Proof of Supporting Lemmas in Appendix D

## E.1 Proof of Proposition D.3

Recall from Algorithm 3, the LMC update rule is

$$\mathbf{w}_{m,h}^{k,j} = \mathbf{w}_{m,h}^{k,j-1} - \eta_{m,k}\nabla L_{m,h}^k\big(\mathbf{w}_{m,h}^{k,j-1}\big) + \sqrt{2\eta_{m,k}\beta_{m,k}^{-1}}\,\boldsymbol{\epsilon}_{m,h}^{k,j},$$

where we have $\nabla L_{m,h}^k\big(\mathbf{w}_{m,h}^{k,j-1}\big) = 2\big(\mathbf{\Lambda}_{m,h}^k\mathbf{w}_{m,h}^{k,j-1} - \boldsymbol{b}_{m,h}^k\big)$. Plug in the above formula, then we can calculate that

$$\mathbf{w}_{m,h}^{k,J_k} = \mathbf{w}_{m,h}^{k,J_k-1} - 2\eta_{m,k}\big(\mathbf{\Lambda}_{m,h}^k\mathbf{w}_{m,h}^{k,J_k-1} - \boldsymbol{b}_{m,h}^k\big) + \sqrt{2\eta_{m,k}\beta_{m,k}^{-1}}\,\boldsymbol{\epsilon}_{m,h}^{k,J_k}$$

$$= \big(\mathbf{I} - 2\eta_{m,k}\mathbf{\Lambda}_{m,h}^k\big)\mathbf{w}_{m,h}^{k,J_k-1} + 2\eta_{m,k}\boldsymbol{b}_{m,h}^k + \sqrt{2\eta_{m,k}\beta_{m,k}^{-1}}\,\boldsymbol{\epsilon}_{m,h}^{k,J_k}$$

$$= \big(\mathbf{I} - 2\eta_{m,k}\mathbf{\Lambda}_{m,h}^k\big)^{J_k}\mathbf{w}_{m,h}^{k,0} + \sum_{l=0}^{J_k-1}(\mathbf{I} - 2\eta_{m,k}\mathbf{\Lambda}_{m,h}^k)^l\Big(2\eta_{m,k}\boldsymbol{b}_{m,h}^k + \sqrt{2\eta_{m,k}\beta_{m,k}^{-1}}\,\boldsymbol{\epsilon}_{m,h}^{k,J_k-l}\Big)$$

$$= \big(\mathbf{I} - 2\eta_{m,k}\mathbf{\Lambda}_{m,h}^k\big)^{J_k}\mathbf{w}_{m,h}^{k,0} + 2\eta_{m,k}\sum_{l=0}^{J_k-1}\big(\mathbf{I} - 2\eta_{m,k}\mathbf{\Lambda}_{m,h}^k\big)^l\boldsymbol{b}_{m,h}^k$$

$$+ \sqrt{2\eta_{m,k}\beta_{m,k}^{-1}} \sum_{l=0}^{J_k-1} \left(\mathbf{I} - 2\eta_{m,k}\mathbf{\Lambda}_{m,h}^k\right)^l \boldsymbol{\epsilon}_{m,h}^{k,J_k-l},$$

where the third equality follows from iteration. Denote that $\mathbf{A}_i = \mathbf{I} - 2\eta_{m,i}\mathbf{\Lambda}_{m,h}^i$. Moreover, we choose the step size such that $0 < \eta_{m,i} < 1/\big(2\lambda_{\max}\big(\mathbf{\Lambda}_{m,h}^i\big)\big)$. Thus we have

$$\mathbf{w}_{m,h}^{k,J_k} = \mathbf{A}_k^{J_k}\mathbf{w}_{m,h}^{k-1,J_{k-1}} + 2\eta_{m,k}\sum_{l=0}^{J_k-1}\mathbf{A}_k^l\mathbf{\Lambda}_{m,h}^k\widehat{\mathbf{w}}_{m,h}^k + \sqrt{2\eta_{m,k}\beta_{m,k}^{-1}}\sum_{l=0}^{J_k-1}\mathbf{A}_k^l\boldsymbol{\epsilon}_{m,h}^{k,J_k-l}$$

$$= \mathbf{A}_k^{J_k}\mathbf{w}_{m,h}^{k-1,J_{k-1}} + (\mathbf{I} - \mathbf{A}_k)\big(\mathbf{I} + \mathbf{A}_k + ... + \mathbf{A}_k^{J_k-1}\big)\widehat{\mathbf{w}}_{m,h}^k + \sqrt{2\eta_{m,k}\beta_{m,k}^{-1}}\sum_{l=0}^{J_k-1}\mathbf{A}_k^l\boldsymbol{\epsilon}_{m,h}^{k,J_k-l}$$

$$= \mathbf{A}_k^{J_k}\mathbf{w}_{m,h}^{k-1,J_{k-1}} + \big(\mathbf{I} - \mathbf{A}_k^{J_k}\big)\widehat{\mathbf{w}}_{m,h}^k + \sqrt{2\eta_{m,k}\beta_{m,k}^{-1}}\sum_{l=0}^{J_k-1}\mathbf{A}_k^l\boldsymbol{\epsilon}_{m,h}^{k,J_k-l}$$

$$= \mathbf{A}_k^{J_k}...\mathbf{A}_1^{J_1}\mathbf{w}_{m,h}^{1,0} + \sum_{i=1}^k \mathbf{A}_k^{J_k}...\mathbf{A}_{i+1}^{J_{i+1}}\big(\mathbf{I} - \mathbf{A}_i^{J_i}\big)\widehat{\mathbf{w}}_{m,h}^i$$

$$+ \sum_{i=1}^k \sqrt{2\eta_{m,i}\beta_{m,i}^{-1}}\mathbf{A}_k^{J_k}...\mathbf{A}_{i+1}^{J_{i+1}}\sum_{l=0}^{J_i-1}\mathbf{A}_i^l\boldsymbol{\epsilon}_{m,h}^{i,J_i-l},$$

where the first equality holds because $\boldsymbol{b}_{m,h}^k = \mathbf{\Lambda}_{m,h}^k\widehat{\mathbf{w}}_{m,h}^k$ and $\mathbf{w}_{m,h}^{k-1,J_{k-1}} = \mathbf{w}_{m,h}^{k,0}$, the third equality follows from the fact that $\mathbf{I} + \mathbf{A} + ... + \mathbf{A}^{n-1} = (\mathbf{I} - \mathbf{A}^n)(\mathbf{I} - \mathbf{A})^{-1}$, and the fourth equality holds because of iteration.

Note that $\boldsymbol{\epsilon}_{m,h}^{i,J_i-l} \sim \mathcal{N}(\mathbf{0}, \mathbf{I})$, based on the property of multivariate Gaussian distribution, we have $\mathbf{w}_{m,h}^{k,J_k} \sim \mathcal{N}\big(\boldsymbol{\mu}_{m,h}^{k,J_k}, \mathbf{\Sigma}_{m,h}^{k,J_k}\big)$. Then we can directly get the mean vector

$$\boldsymbol{\mu}_{m,h}^{k,J_k} = \mathbf{A}_k^{J_k}...\mathbf{A}_1^{J_1}\mathbf{w}_{m,h}^{1,0} + \sum_{i=1}^k \mathbf{A}_k^{J_k}...\mathbf{A}_{i+1}^{J_{i+1}}\big(\mathbf{I} - \mathbf{A}_i^{J_i}\big)\widehat{\mathbf{w}}_{m,h}^i.$$

Next we will calculate the covariance matrix $\mathbf{\Sigma}_{m,h}^{k,J_k}$. For simplicity, we define $\mathbf{M}_i = \sqrt{2\eta_{m,i}\beta_{m,i}^{-1}}\mathbf{A}_k^{J_k}...\mathbf{A}_{i+1}^{J_{i+1}}$, thus we have

$$\mathbf{M}_i\sum_{l=0}^{J_i-1}\mathbf{A}_i^l\boldsymbol{\epsilon}_{m,h}^{i,J_i-l} \sim \mathcal{N}\left(\mathbf{0}, \sum_{l=0}^{J_i-1}\mathbf{M}_i\mathbf{A}_i^l\big(\mathbf{M}_i\mathbf{A}_i^l\big)^\top\right) \sim \mathcal{N}\left(\mathbf{0}, \mathbf{M}_i\left(\sum_{l=0}^{J_i-1}\mathbf{A}_i^{2l}\right)\mathbf{M}_i^\top\right).$$

Thus we get the covariance matrix $\mathbf{\Sigma}_{m,h}^{k,J_k}$,

$$\mathbf{\Sigma}_{m,h}^{k,J_k} = \sum_{i=1}^k \mathbf{M}_i\left(\sum_{l=0}^{J_i-1}\mathbf{A}_i^{2l}\right)\mathbf{M}_i^\top$$

$$= \sum_{i=1}^k 2\eta_{m,i}\beta_{m,i}^{-1}\mathbf{A}_k^{J_k}...\mathbf{A}_{i+1}^{J_{i+1}}\left(\sum_{l=0}^{J_i-1}\mathbf{A}_i^{2l}\right)\mathbf{A}_{i+1}^{J_{i+1}}...\mathbf{A}_k^{J_k}$$

$$= \sum_{i=1}^k 2\eta_{m,i}\beta_{m,i}^{-1}\mathbf{A}_k^{J_k}...\mathbf{A}_{i+1}^{J_{i+1}}\big(\mathbf{I} - \mathbf{A}_i^{2J_i}\big)(\mathbf{I} - \mathbf{A}_i^2)^{-1}\mathbf{A}_{i+1}^{J_{i+1}}...\mathbf{A}_k^{J_k}$$

$$= \sum_{i=1}^k \frac{1}{\beta_{m,i}}\mathbf{A}_k^{J_k}...\mathbf{A}_{i+1}^{J_{i+1}}\big(\mathbf{I} - \mathbf{A}_i^{2J_i}\big)\big(\mathbf{\Lambda}_{m,h}^i\big)^{-1}(\mathbf{I} + \mathbf{A}_i)^{-1}\mathbf{A}_{i+1}^{J_{i+1}}...\mathbf{A}_k^{J_k},$$

where the third equality follows from the fact that $\mathbf{I} + \mathbf{A} + ... + \mathbf{A}^{n-1} = (\mathbf{I} - \mathbf{A}^n)(\mathbf{I} - \mathbf{A})^{-1}$. Here we complete the proof.

## E.2 Proof of Lemma D.4

*Proof.* Note that $\widehat{\mathbf{w}}_{m,h}^k = \left(\boldsymbol{\Lambda}_{m,k}^k\right)^{-1}\boldsymbol{b}_{m,h}^k$, we can calculate that

$$
\begin{aligned}
\left\|\widehat{\mathbf{w}}_{m,h}^k\right\| &= \left\|\left(\boldsymbol{\Lambda}_{m,h}^k\right)^{-1}\boldsymbol{b}_{m,h}^k\right\| \\
&= \left\|\left(\boldsymbol{\Lambda}_{m,h}^k\right)^{-1}\sum_{(s^l,a^l,s'^l)\in U_{m,h}(k)}\left[r_h\left(s^l,a^l\right)+V_{m,h+1}^k(s'^l)\right]\boldsymbol{\phi}\left(s^l,a^l\right)\right\| \\
&\le \frac{1}{\sqrt{\lambda}}\sqrt{\mathcal{K}(k)}\left(\sum_{(s^l,a^l,s'^l)\in U_{m,h}(k)}\left\|\left[r_h\left(s^l,a^l\right)+V_{m,h+1}^k(s'^l)\right]\boldsymbol{\phi}\left(s^l,a^l\right)\right\|_{(\boldsymbol{\Lambda}_{m,h}^k)^{-1}}^2\right)^{1/2} \\
&\le \frac{2H}{\sqrt{\lambda}}\sqrt{\mathcal{K}(k)}\left(\sum_{(s^l,a^l,s'^l)\in U_{m,h}(k)}\left\|\boldsymbol{\phi}(s^l,a^l)\right\|_{(\boldsymbol{\Lambda}_{m,h}^k)^{-1}}^2\right)^{1/2} \\
&\le 2H\sqrt{\mathcal{K}(k)d/\lambda} \\
&\le 2H\sqrt{Mkd/\lambda},
\end{aligned}
$$

where the first inequality follows from Lemma J.3, the second inequality is due to $0 \le V_{m,h}^k \le H - h + 1$, $0 \le r_h \le 1$ and $\|\boldsymbol{\phi}(s,a)\| \le 1$, the third inequality follows from Lemma J.4, and the last inequality holds because $\mathcal{K}(k) = (M-1)k_s + k - 1 \le Mk$. $\qquad\square$

## E.3 Proof of Lemma D.5

*Proof.* We separate the error into two terms and bound them respectively,

$$
\left|\boldsymbol{\phi}(s,a)^\top\mathbf{w}_{m,h}^{k,J_k} - \boldsymbol{\phi}(s,a)^\top\widehat{\mathbf{w}}_{m,h}^k\right| \le \underbrace{\left|\boldsymbol{\phi}(s,a)^\top\left(\mathbf{w}_{m,h}^{k,J_k} - \boldsymbol{\mu}_{m,h}^{k,J_k}\right)\right|}_{I_1} + \underbrace{\left|\boldsymbol{\phi}(s,a)^\top\left(\boldsymbol{\mu}_{m,h}^{k,J_k} - \widehat{\mathbf{w}}_{m,h}^k\right)\right|}_{I_2}.
$$
(E.1)

**Bounding Term $I_1$ in** (E.1)**:** by Cauchy-Schwarz inequality, we have

$$
\left|\boldsymbol{\phi}(s,a)^\top\left(\mathbf{w}_{m,h}^{k,J_k} - \boldsymbol{\mu}_{m,h}^{k,J_k}\right)\right| \le \left\|\boldsymbol{\phi}(s,a)\right\|_{\boldsymbol{\Sigma}_{m,h}^{k,J_k}} \cdot \left\|\mathbf{w}_{m,h}^{k,J_k} - \boldsymbol{\mu}_{m,h}^{k,J_k}\right\|_{(\boldsymbol{\Sigma}_{m,h}^{k,J_k})^{-1}}.
$$

By choosing $\eta_{m,k} \le 1/(4\lambda_{\max}(\boldsymbol{\Lambda}_{m,h}^k))$ for all $k$ and $m$, then we have

$$
\begin{aligned}
\frac{1}{2}\mathbf{I} &\preccurlyeq \mathbf{A}_k = \mathbf{I} - 2\eta_{m,k}\boldsymbol{\Lambda}_{m,h}^k \preccurlyeq (1 - 2\eta_{m,k}\lambda_{\min}(\boldsymbol{\Lambda}_{m,h}^k))\mathbf{I}, \\
\frac{3}{2}\mathbf{I} &\preccurlyeq \mathbf{I} + \mathbf{A}_k = 2\mathbf{I} - 2\eta_{m,k}\boldsymbol{\Lambda}_{m,h}^k \preccurlyeq 2\mathbf{I}.
\end{aligned}
$$
(E.2)

Recall the definition of $\boldsymbol{\Sigma}_{m,h}^{k,J_k}$ in Proposition D.3. By choosing $\beta_{m,i} = \beta_K$ for all $i \in [k]$ and $m \in \mathcal{M}$, then we have

$$
\begin{aligned}
&\boldsymbol{\phi}(s,a)^\top\boldsymbol{\Sigma}_{m,h}^{k,J_k}\boldsymbol{\phi}(s,a) \\
&= \sum_{i=1}^k\frac{1}{\beta_{m,i}}\boldsymbol{\phi}(s,a)^\top\mathbf{A}_k^{J_k}...\mathbf{A}_{i+1}^{J_{i+1}}\left(\mathbf{I}-\mathbf{A}_i^{2J_i}\right)\left(\boldsymbol{\Lambda}_{m,h}^i\right)^{-1}(\mathbf{I}+\mathbf{A}_i)^{-1}\mathbf{A}_{i+1}^{J_{i+1}}...\mathbf{A}_k^{J_k}\boldsymbol{\phi}(s,a) \\
&\le \frac{2}{3\beta_{m,i}}\sum_{i=1}^k\boldsymbol{\phi}(s,a)^\top\mathbf{A}_k^{J_k}...\mathbf{A}_{i+1}^{J_{i+1}}\left(\left(\boldsymbol{\Lambda}_{m,h}^i\right)^{-1} - \mathbf{A}_i^{J_i}\left(\boldsymbol{\Lambda}_{m,h}^i\right)^{-1}\mathbf{A}_i^{J_i}\right)\mathbf{A}_{i+1}^{J_{i+1}}...\mathbf{A}_k^{J_k}\boldsymbol{\phi}(s,a) \\
&= \frac{2}{3\beta_K}\sum_{i=1}^{k-1}\boldsymbol{\phi}(s,a)^\top\mathbf{A}_k^{J_k}...\mathbf{A}_{i+1}^{J_{i+1}}\left(\left(\boldsymbol{\Lambda}_{m,h}^i\right)^{-1} - \left(\boldsymbol{\Lambda}_{m,h}^{i+1}\right)^{-1}\right)\mathbf{A}_{i+1}^{J_{i+1}}...\mathbf{A}_k^{J_k}\boldsymbol{\phi}(s,a) \\
&\quad - \frac{2}{3\beta_K}\boldsymbol{\phi}(s,a)^\top\mathbf{A}_k^{J_k}...\mathbf{A}_1^{J_1}(\boldsymbol{\Lambda}_{m,h}^1)^{-1}\mathbf{A}_1^{J_1}...\mathbf{A}_k^{J_k}\boldsymbol{\phi}(s,a)
\end{aligned}
$$

$$+ \frac{2}{3\beta_K} \phi(s,a)^\top (\mathbf{\Lambda}_{m,h}^k)^{-1} \phi(s,a),$$

where the first inequality follows from (E.2). By the definition of $\mathbf{\Lambda}_{m,h}^i$ and Woodbury formula, we have

$$\left(\mathbf{\Lambda}_{m,h}^i\right)^{-1} - \left(\mathbf{\Lambda}_{m,h}^{i+1}\right)^{-1} = \left(\mathbf{\Lambda}_{m,h}^i\right)^{-1} - \left(\mathbf{\Lambda}_{m,h}^i + \sum_{(s^l,a^l,s'^l)\in U_{m,h}(k)} \phi(s^l,a^l)\phi(s^l,a^l)^\top\right)^{-1}$$

$$= (\mathbf{\Lambda}_{m,h}^i)^{-1}\boldsymbol{\varphi}(\mathbf{I}_n + \boldsymbol{\varphi}^\top(\mathbf{\Lambda}_{m,h}^i)^{-1}\boldsymbol{\varphi})^{-1}\boldsymbol{\varphi}^\top(\mathbf{\Lambda}_{m,h}^i)^{-1},$$

where $\boldsymbol{\varphi}$ is a matrix with the dimension of $d \times n$, $n$ is the number difference of $\phi(s^l,a^l)$ between $\left(\mathbf{\Lambda}_{m,h}^i\right)^{-1}$ and $\left(\mathbf{\Lambda}_{m,h}^{i+1}\right)^{-1}$ (*i.e.* we concatenate all $\phi(s^l,a^l)$ into the matrix $\boldsymbol{\varphi}$). Note that $n \leq M$, then we have

$$\phi(s,a)^\top \mathbf{A}_k^{J_k}...\mathbf{A}_{i+1}^{J_{i+1}}\left(\left(\mathbf{\Lambda}_{m,h}^i\right)^{-1} - \left(\mathbf{\Lambda}_{m,h}^{i+1}\right)^{-1}\right)\mathbf{A}_{i+1}^{J_{i+1}}...\mathbf{A}_k^{J_k}\phi(s,a)$$

$$= \phi(s,a)^\top \mathbf{A}_k^{J_k}...\mathbf{A}_{i+1}^{J_{i+1}}\left(\left(\mathbf{\Lambda}_{m,h}^i\right)^{-1}\boldsymbol{\varphi}(\mathbf{I}_n + \boldsymbol{\varphi}^\top\left(\mathbf{\Lambda}_{m,h}^i\right)^{-1}\boldsymbol{\varphi})^{-1}\boldsymbol{\varphi}^\top\left(\mathbf{\Lambda}_{m,h}^i\right)^{-1}\right)\mathbf{A}_{i+1}^{J_{i+1}}...\mathbf{A}_k^{J_k}\phi(s,a)$$

$$\leq \phi(s,a)^\top \mathbf{A}_k^{J_k}...\mathbf{A}_{i+1}^{J_{i+1}}\left(\mathbf{\Lambda}_{m,h}^i\right)^{-1}\boldsymbol{\varphi}\boldsymbol{\varphi}^\top\left(\mathbf{\Lambda}_{m,h}^i\right)^{-1}\mathbf{A}_{i+1}^{J_{i+1}}...\mathbf{A}_k^{J_k}\phi(s,a)$$

$$= \left\|\phi(s,a)^\top \mathbf{A}_k^{J_k}...\mathbf{A}_{i+1}^{J_{i+1}}\left(\mathbf{\Lambda}_{m,h}^i\right)^{-1}\boldsymbol{\varphi}\right\|_2^2$$

$$\leq \left\|\mathbf{A}_k^{J_k}...\mathbf{A}_{i+1}^{J_{i+1}}(\mathbf{\Lambda}_{m,h}^i)^{-1/2}\phi(s,a)\right\|_2^2 \cdot \left\|(\mathbf{\Lambda}_{m,h}^i)^{-1/2}\boldsymbol{\varphi}\right\|_F^2$$

$$\leq \prod_{j=i+1}^k \left(1 - 2\eta_{m,j}\lambda_{\min}\left(\mathbf{\Lambda}_{m,h}^j\right)\right)^{2J_j} \text{tr}\left(\boldsymbol{\varphi}^\top\left(\mathbf{\Lambda}_{m,h}^i\right)^{-1}\boldsymbol{\varphi}\right)\|\phi(s,a)\|_{(\mathbf{\Lambda}_{m,h}^i)^{-1}}^2,$$

where $\|\cdot\|_F$ is Frobenius norm and the last inequality is due to $\|\mathbf{\Lambda}^{-\frac{1}{2}}\mathbf{X}\|_F^2 = \text{tr}(\mathbf{X}^\top\mathbf{\Lambda}^{-1}\mathbf{X})$ and (E.2). Thus we have

$$\|\phi(s,a)\|_{\mathbf{\Sigma}_{m,h}^{k,J_k}}^2 \leq \frac{2}{3\beta_K}\sum_{i=1}^k\prod_{j=i+1}^k\left(1 - 2\eta_{m,j}\lambda_{\min}\left(\mathbf{\Lambda}_{m,h}^j\right)\right)^{2J_j}\text{tr}\left(\boldsymbol{\varphi}^\top\left(\mathbf{\Lambda}_{m,h}^i\right)^{-1}\boldsymbol{\varphi}\right)\|\phi(s,a)\|_{(\mathbf{\Lambda}_{m,h}^i)^{-1}}^2$$

$$+ \frac{2}{3\beta_K}\|\phi(s,a)\|_{(\mathbf{\Lambda}_{m,h}^k)^{-1}}^2.$$

Using the inequality $\sqrt{a^2 + b^2} \leq a + b$ for $a, b > 0$, we get

$$\|\phi(s,a)\|_{\mathbf{\Sigma}_{m,h}^{k,J_k}} \leq \sqrt{\frac{2}{3\beta_K}}\Bigg(\sum_{i=1}^k\prod_{j=i+1}^k\left(1 - 2\eta_{m,j}\lambda_{\min}(\mathbf{\Lambda}_{m,h}^j)\right)^{J_j}\text{tr}(\boldsymbol{\varphi}^\top(\mathbf{\Lambda}_{m,h}^i)^{-1}\boldsymbol{\varphi})^{\frac{1}{2}}\|\phi(s,a)\|_{(\mathbf{\Lambda}_{m,h}^i)^{-1}}$$

$$+ \|\phi(s,a)\|_{(\mathbf{\Lambda}_{m,h}^k)^{-1}}\Bigg)$$

$$\stackrel{\text{def}}{=} \widehat{g}_{m,h}^k(\phi(s,a)).$$

Note that $\left(\mathbf{\Sigma}_{m,h}^{k,J_k}\right)^{-1/2}\left(\mathbf{w}_{m,h}^{k,J_k} - \boldsymbol{\mu}_{m,h}^{k,J_k}\right) \sim \mathcal{N}(\mathbf{0}, \mathbf{I}_d)$. By the Gaussian concentration property, we have

$$\mathbb{P}\left(\left\|\left(\mathbf{\Sigma}_{m,h}^{k,J_k}\right)^{-1/2}\left(\mathbf{w}_{m,h}^{k,J_k} - \boldsymbol{\mu}_{m,h}^{k,J_k}\right)\right\| \geq \sqrt{4d\log(1/\delta)}\right) \leq \delta^2.$$

Then we have

$$\mathbb{P}\left(\left|\phi(s,a)^\top\mathbf{w}_{m,h}^{k,J_k} - \phi(s,a)^\top\boldsymbol{\mu}_{m,h}^{k,J_k}\right| \geq 2\widehat{g}_{m,h}^k(\phi(s,a))\sqrt{d\log(1/\delta)}\right)$$

$$\leq \mathbb{P}\left(\left|\phi(s,a)^\top\mathbf{w}_{m,h}^{k,J_k} - \phi(s,a)^\top\boldsymbol{\mu}_{m,h}^{k,J_k}\right| \geq 2\sqrt{d\log(1/\delta)}\|\phi(s,a)\|_{\mathbf{\Sigma}_{m,h}^{k,J_k}}\right)$$

$$\leq \mathbb{P}\left(\left\|\phi(s,a)\right\|_{\mathbf{\Sigma}_{m,h}^{k,J_k}}\cdot\left\|\mathbf{w}_{m,h}^{k,J_k} - \boldsymbol{\mu}_{m,h}^{k,J_k}\right\|_{(\mathbf{\Sigma}_{m,h}^{k,J_k})^{-1}} \geq 2\sqrt{d\log(1/\delta)}\|\phi(s,a)\|_{\mathbf{\Sigma}_{m,h}^{k,J_k}}\right)$$

$$= \mathbb{P}\left(\left\|\left(\mathbf{\Sigma}_{m,h}^{k,J_k}\right)^{-1/2}\left(\mathbf{w}_{m,h}^{k,J_k} - \boldsymbol{\mu}_{m,h}^{k,J_k}\right)\right\| \geq 2\sqrt{d\log(1/\delta)}\right)$$

$$\leq \delta^2. \tag{E.3}$$

**Bounding Term $I_2$ in** (E.1)**:** Recall from Proposition D.3, we have

$$
\begin{aligned}
\boldsymbol{\mu}_{m,h}^{k,J_k} &= \mathbf{A}_k^{J_k}...\mathbf{A}_1^{J_1}\mathbf{w}_{m,h}^{1,0} + \sum_{i=1}^{k} \mathbf{A}_k^{J_k}...\mathbf{A}_{i+1}^{J_{i+1}}\big(\mathbf{I} - \mathbf{A}_i^{J_i}\big)\widehat{\mathbf{w}}_{m,h}^{i} \\
&= \mathbf{A}_k^{J_k}...\mathbf{A}_1^{J_1}\mathbf{w}_{m,h}^{1,0} + \sum_{i=1}^{k-1} \mathbf{A}_k^{J_k}...\mathbf{A}_{i+1}^{J_{i+1}}(\widehat{\mathbf{w}}_{m,h}^{i} - \widehat{\mathbf{w}}_{m,h}^{i+1}) - \mathbf{A}_k^{J_k}...\mathbf{A}_1^{J_1}\widehat{\mathbf{w}}_{m,h}^{1} + \widehat{\mathbf{w}}_{m,h}^{k} \\
&= \mathbf{A}_k^{J_k}...\mathbf{A}_1^{J_1}(\mathbf{w}_{m,h}^{1,0} - \widehat{\mathbf{w}}_{m,h}^{1}) + \sum_{i=1}^{k-1} \mathbf{A}_k^{J_k}...\mathbf{A}_{i+1}^{J_{i+1}}(\widehat{\mathbf{w}}_{m,h}^{i} - \widehat{\mathbf{w}}_{m,h}^{i+1}) + \widehat{\mathbf{w}}_{m,h}^{k}.
\end{aligned}
$$

Then we can get

$$
\begin{aligned}
&\boldsymbol{\phi}(s,a)^{\top}(\boldsymbol{\mu}_{m,h}^{k,J_k} - \widehat{\mathbf{w}}_{m,h}^{k}) \\
&= \underbrace{\boldsymbol{\phi}(s,a)^{\top}\mathbf{A}_k^{J_k}...\mathbf{A}_1^{J_1}(\mathbf{w}_{m,h}^{1,0} - \widehat{\mathbf{w}}_{m,h}^{1})}_{I_{21}} + \underbrace{\boldsymbol{\phi}(s,a)^{\top}\sum_{i=1}^{k-1}\mathbf{A}_k^{J_k}...\mathbf{A}_{i+1}^{J_{i+1}}(\widehat{\mathbf{w}}_{m,h}^{i} - \widehat{\mathbf{w}}_{m,h}^{i+1})}_{I_{22}}.
\end{aligned}
$$

In Algorithm 3, we choose $\mathbf{w}_{m,h}^{1,0} = \mathbf{0}$ and $\widehat{\mathbf{w}}_{m,h}^{1} = (\boldsymbol{\Lambda}_{m,h}^{1})^{-1}\boldsymbol{b}_{m,h}^{1} = \mathbf{0}$. Thus we have $I_{21} = 0$. To bound term $I_{22}$, we use the inequalities in (E.2) and Lemma D.4, we have

$$
\begin{aligned}
I_{22} &\leq \Big| \sum_{i=1}^{k-1} \boldsymbol{\phi}(s,a)^{\top}\mathbf{A}_k^{J_k}...\mathbf{A}_{i+1}^{J_{i+1}}(\widehat{\mathbf{w}}_{m,h}^{i} - \widehat{\mathbf{w}}_{m,h}^{i+1})\Big| \\
&\leq \sum_{i=1}^{k-1} \prod_{j=i+1}^{k} \Big(1 - 2\eta_{m,j}\lambda_{\min}(\boldsymbol{\Lambda}_{m,h}^{j})\Big)^{J_j} \|\boldsymbol{\phi}(s,a)\|(\|\widehat{\mathbf{w}}_{m,h}^{i}\| + \|\widehat{\mathbf{w}}_{m,h}^{i+1}\|) \\
&\leq \sum_{i=1}^{k-1} \prod_{j=i+1}^{k} \Big(1 - 2\eta_{m,j}\lambda_{\min}(\boldsymbol{\Lambda}_{m,h}^{j})\Big)^{J_j} \|\boldsymbol{\phi}(s,a)\|\big(2H\sqrt{Mid/\lambda} + 2H\sqrt{M(i+1)d/\lambda}\big) \\
&\leq 4H\sqrt{MKd/\lambda}\sum_{i=1}^{k-1} \prod_{j=i+1}^{k} \Big(1 - 2\eta_{m,j}\lambda_{\min}(\boldsymbol{\Lambda}_{m,h}^{j})\Big)^{J_j} \|\boldsymbol{\phi}(s,a)\|.
\end{aligned}
$$

Thus we get

$$
\boldsymbol{\phi}(s,a)^{\top}\big(\boldsymbol{\mu}_{m,h}^{k,J_k} - \widehat{\mathbf{w}}_{m,h}^{k}\big) \leq 4H\sqrt{MKd/\lambda}\sum_{i=1}^{k-1} \prod_{j=i+1}^{k} \Big(1 - 2\eta_{m,j}\lambda_{\min}(\boldsymbol{\Lambda}_{m,h}^{j})\Big)^{J_j} \|\boldsymbol{\phi}(s,a)\|. \tag{E.4}
$$

Substituting (E.3) and (E.4) into (E.1), with probability at least $1 - \delta^2$, we have

$$
\begin{aligned}
&\Big|\boldsymbol{\phi}(s,a)^{\top}\mathbf{w}_{m,h}^{k,J_k} - \boldsymbol{\phi}(s,a)^{\top}\widehat{\mathbf{w}}_{m,h}^{k}\Big| \\
&\leq 4H\sqrt{MKd/\lambda}\sum_{i=1}^{k-1} \prod_{j=i+1}^{k} \Big(1 - 2\eta_{m,j}\lambda_{\min}(\boldsymbol{\Lambda}_{m,h}^{j})\Big)^{J_j} \|\boldsymbol{\phi}(s,a)\| + 2\sqrt{\frac{2d\log(1/\delta)}{3\beta_K}}\|\boldsymbol{\phi}(s,a)\|_{(\boldsymbol{\Lambda}_{m,h}^{k})^{-1}} \\
&\quad + 2\sqrt{\frac{2d\log(1/\delta)}{3\beta_K}}\sum_{i=1}^{k} \prod_{j=i+1}^{k} \Big(1 - 2\eta_{m,j}\lambda_{\min}(\boldsymbol{\Lambda}_{m,h}^{j})\Big)^{J_j} \mathrm{tr}\big(\boldsymbol{\varphi}^{\top}(\boldsymbol{\Lambda}_{m,h}^{i})^{-1}\boldsymbol{\varphi}\big)^{\frac{1}{2}}\|\boldsymbol{\phi}(s,a)\|_{(\boldsymbol{\Lambda}_{m,h}^{i})^{-1}} \\
&\stackrel{\text{def}}{=} W. \tag{E.5}
\end{aligned}
$$

Here we choose $\eta_{m,j} = 1/(4\lambda_{\max}(\boldsymbol{\Lambda}_{m,h}^{j}))$ and set $\kappa_j = \lambda_{\max}(\boldsymbol{\Lambda}_{m,h}^{j})/\lambda_{\min}(\boldsymbol{\Lambda}_{m,h}^{j})$, then we have

$$
\Big(1 - 2\eta_{m,j}\lambda_{\min}(\boldsymbol{\Lambda}_{m,h}^{j})\Big)^{J_j} = (1 - 1/2\kappa_j)^{J_j}.
$$

We want to have $(1 - 1/2\kappa_j)^{J_j} < \epsilon$, it suffices to choose $J_j$ such that

$$J_j \geq \frac{\log(1/\epsilon)}{\log\left(\frac{1}{1-1/2\kappa_j}\right)}.$$

Note that $1/2\kappa_j \leq 1/2$, we have $\log(1/(1 - 1/2\kappa_j)) \geq 1/2\kappa_j$ because $e^{-x} > 1 - x$ for $0 < x < 1$. Therefore, we only need to pick $J_j \geq 2\kappa_j \log(1/\epsilon)$.

Also note that $1 \geq \|\phi(s,a)\| \geq \sqrt{\lambda}\|\phi(s,a)\|_{(\Lambda_{m,h}^i)^{-1}}$ and $\mathrm{tr}\left(\varphi^\top\left(\Lambda_{m,h}^i\right)^{-1}\varphi\right) \leq M$ due to the fact that $n \leq M$. By setting $\epsilon = 1/(4HMKd)$ and $\lambda = 1$, we obtain

$$W \leq \sum_{i=1}^{k-1}\epsilon^{k-i}4H\sqrt{MKd/\lambda}\|\phi(s,a)\| + 2\sqrt{\frac{2d\log(1/\delta)}{3\beta_K}}\left(\|\phi(s,a)\|_{(\Lambda_{m,h}^k)^{-1}} + \sum_{i=1}^{k-1}\epsilon^{k-i}\sqrt{M}\|\phi(s,a)\|\right)$$

$$\leq \sum_{i=1}^{k-1}\epsilon^{k-i}4H\sqrt{MKd/\lambda}\sqrt{MK}\|\phi(s,a)\|_{(\Lambda_{m,h}^k)^{-1}}$$

$$\quad + 2\sqrt{\frac{2d\log(1/\delta)}{3\beta_K}}\left(\|\phi(s,a)\|_{(\Lambda_{m,h}^k)^{-1}} + \sum_{i=1}^{k-1}\epsilon^{k-i}M\sqrt{K}\|\phi(s,a)\|_{(\Lambda_{m,h}^k)^{-1}}\right)$$

$$\leq \sum_{i=1}^{k-1}\epsilon^{k-i-1}\|\phi(s,a)\|_{(\Lambda_{m,h}^k)^{-1}} + 2\sqrt{\frac{2d\log(1/\delta)}{3\beta_K}}\left(\|\phi(s,a)\|_{(\Lambda_{m,h}^k)^{-1}} + \sum_{i=1}^{k-1}\epsilon^{k-i-1}\|\phi(s,a)\|_{(\Lambda_{m,h}^k)^{-1}}\right)$$

$$\leq \left(5\sqrt{\frac{2d\log(1/\delta)}{3\beta_K}} + \frac{4}{3}\right)\|\phi(s,a)\|_{(\Lambda_{m,h}^k)^{-1}},$$

where the second inequality follows from $\|\phi(s,a)\|_{(\Lambda_{m,h}^k)^{-1}} \geq 1/\sqrt{\mathcal{K}(k)+1}\|\phi(s,a)\| \geq 1/\sqrt{MK}\|\phi(s,a)\|$, the fourth inequality follows from $\sum_{i=1}^{k-1}\epsilon^{k-i-1} = \sum_{i=0}^{k-2}\epsilon^i < 1/(1-\epsilon) \leq 4/3$. Finally we have

$$\mathbb{P}\left(\left|\phi(s,a)^\top\mathbf{w}_{m,h}^{k,J_k} - \phi(s,a)^\top\widehat{\mathbf{w}}_{m,h}^k\right| \leq \left(5\sqrt{\frac{2d\log(1/\delta)}{3\beta_K}} + \frac{4}{3}\right)\|\phi(s,a)\|_{(\Lambda_{m,h}^k)^{-1}}\right)$$

$$\geq \mathbb{P}\left(\left|\phi(s,a)^\top\mathbf{w}_{m,h}^{k,J_k} - \phi(s,a)^\top\widehat{\mathbf{w}}_{m,h}^k\right| \leq W\right)$$

$$\geq 1 - \delta^2.$$

This completes the proof. $\qquad\square$

### E.4  Proof of Lemma D.6

*Proof.* Recall that $\mathbf{w}_{m,h}^{k,J_k} \sim \mathcal{N}\left(\boldsymbol{\mu}_{m,h}^{k,J_k}, \boldsymbol{\Sigma}_{m,h}^{k,J_k}\right)$. Let $\boldsymbol{\xi}_{m,h}^{k,J_k} = \mathbf{w}_{m,h}^{k,J_k} - \boldsymbol{\mu}_{m,h}^{k,J_k} \sim \mathcal{N}(\mathbf{0}, \boldsymbol{\Sigma}_{m,h}^{k,J_k})$, thus we have

$$\|\mathbf{w}_{m,h}^{k,J_k}\| = \|\boldsymbol{\mu}_{m,h}^{k,J_k} + \boldsymbol{\xi}_{m,h}^{k,J_k}\| \leq \|\boldsymbol{\mu}_{m,h}^{k,J_k}\| + \|\boldsymbol{\xi}_{m,h}^{k,J_k}\|. \tag{E.6}$$

**Bounding** $\|\boldsymbol{\mu}_{m,h}^{k,J_k}\|$ **in** (E.6): Based on Proposition D.3, we have

$$\|\boldsymbol{\mu}_{m,h}^{k,J_k}\| = \left\|\mathbf{A}_k^{J_k}\ldots\mathbf{A}_1^{J_1}\mathbf{w}_{m,h}^{1,0} + \sum_{i=1}^k \mathbf{A}_k^{J_k}\ldots\mathbf{A}_{i+1}^{J_{i+1}}\left(\mathbf{I} - \mathbf{A}_i^{J_i}\right)\widehat{\mathbf{w}}_{m,h}^i\right\|$$

$$\leq \sum_{i=1}^k\left\|\mathbf{A}_k^{J_k}\ldots\mathbf{A}_{i+1}^{J_{i+1}}\left(\mathbf{I} - \mathbf{A}_i^{J_i}\right)\right\|_F \cdot \left\|\widehat{\mathbf{w}}_{m,h}^i\right\|$$

$$\leq 2H\sqrt{MKd/\lambda}\sum_{i=1}^k\left\|\mathbf{A}_k^{J_k}\ldots\mathbf{A}_{i+1}^{J_{i+1}}\left(\mathbf{I} - \mathbf{A}_i^{J_i}\right)\right\|_F$$

$$\leq 2Hd\sqrt{MK/\lambda}\sum_{i=1}^k\|\mathbf{A}_k\|_2^{J_k}\ldots\|\mathbf{A}_{i+1}\|_2^{J_{i+1}}\left\|\left(\mathbf{I} - \mathbf{A}_i^{J_i}\right)\right\|_2$$

$$\leq 2Hd\sqrt{MK/\lambda}\sum_{i=1}^{k}\prod_{j=i+1}^{k}\left(1-2\eta_{m,j}\lambda_{\min}\left(\mathbf{\Lambda}_{m,h}^{j}\right)\right)^{J_j}\left(\|\mathbf{I}\|_2+\left\|\mathbf{A}_i\right\|_2^{J_i}\right)$$

$$\leq 2Hd\sqrt{MK/\lambda}\sum_{i=1}^{k}\prod_{j=i+1}^{k}\left(1-2\eta_{m,j}\lambda_{\min}\left(\mathbf{\Lambda}_{m,h}^{j}\right)\right)^{J_j}\left(1+\left(1-2\eta_{m,i}\lambda_{\min}\left(\mathbf{\Lambda}_{m,h}^{i}\right)\right)^{J_j}\right),$$

where the second inequality holds from Lemma D.4, the third inequality follows from the fact that $\operatorname{rank}\left(\mathbf{A}_k^{J_k}\ldots\mathbf{A}_{i+1}^{J_{i+1}}\left(\mathbf{I}-\mathbf{A}_i^{J_i}\right)\right)\leq d$ and $\|\mathbf{X}\|_2\leq\|\mathbf{X}\|_F\leq\operatorname{rank}(\mathbf{X})\|\mathbf{X}\|_2$ where $\|\mathbf{X}\|_2=\sigma_{\max}(\mathbf{X})$.

Recall that in Lemma D.5, we set $J_j\geq 2\kappa_j\log(1/\epsilon)$ where $\kappa_j=\lambda_{\max}\left(\mathbf{\Lambda}_{m,h}^{j}\right)/\lambda_{\min}\left(\mathbf{\Lambda}_{m,h}^{j}\right)$, $\epsilon=1/(4HMKd)$ and $\lambda=1$, thus we get

$$\left\|\boldsymbol{\mu}_{m,h}^{k,J_k}\right\|\leq 2Hd\sqrt{MK/\lambda}\sum_{i=1}^{k}(\epsilon^{k-i}+\epsilon^{k-i+1})$$

$$\leq 4Hd\sqrt{MK/\lambda}\sum_{i=0}^{\infty}\epsilon^i$$

$$\leq\frac{16}{3}Hd\sqrt{MK}.$$

**Bounding** $\left\|\boldsymbol{\xi}_{m,h}^{k,J_k}\right\|$ **in** (E.6): Note that $\boldsymbol{\xi}_{m,h}^{k,J_k}\sim\mathcal{N}\left(\mathbf{0},\mathbf{\Sigma}_{m,h}^{k,J_k}\right)$, using Gaussian concentration Lemma J.5, we have

$$\mathbb{P}\left(\left\|\boldsymbol{\xi}_{m,h}^{k,J_k}\right\|\leq\sqrt{\frac{1}{\delta}\operatorname{tr}\left(\mathbf{\Sigma}_{m,h}^{k,J_k}\right)}\right)\geq 1-\delta.$$

Recall from Proposition D.3, we have

$$\operatorname{tr}\left(\mathbf{\Sigma}_{m,h}^{k,J_k}\right)=\sum_{i=1}^{k}\frac{1}{\beta_{m,i}}\operatorname{tr}\left(\mathbf{A}_k^{J_k}\ldots\mathbf{A}_{i+1}^{J_{i+1}}\left(\mathbf{I}-\mathbf{A}_i^{2J_i}\right)(\mathbf{\Lambda}_{m,h}^{i})^{-1}(\mathbf{I}+\mathbf{A}_i)^{-1}\mathbf{A}_{i+1}^{J_{i+1}}\ldots\mathbf{A}_k^{J_k}\right)$$

$$\leq\sum_{i=1}^{k}\frac{1}{\beta_{m,i}}\operatorname{tr}\left(\mathbf{A}_k^{J_k}\right)\ldots\operatorname{tr}\left(\mathbf{A}_{i+1}^{J_{i+1}}\right)\operatorname{tr}\left(\mathbf{I}-\mathbf{A}_i^{2J_i}\right)\operatorname{tr}\left(\left(\mathbf{\Lambda}_{m,h}^{i}\right)^{-1}\right)\operatorname{tr}\left(\left(\mathbf{I}+\mathbf{A}_i\right)^{-1}\right)$$

$$\times\operatorname{tr}\left(\mathbf{A}_{i+1}^{J_{i+1}}\right)\ldots\operatorname{tr}\left(\mathbf{A}_k^{J_k}\right),$$

where the inequality holds due to Lemma J.6. Recall from (E.2) that, when $\eta_{m,k}\leq 1/(4\lambda_{\max}(\mathbf{\Lambda}_{m,h}^{k}))$ for all $k$ and $m$, we have $\mathbf{A}_i^{J_i}\preccurlyeq(1-2\eta_{m,k}\lambda_{\min}(\mathbf{\Lambda}_{m,h}^{k}))^{J_j}\mathbf{I}$, set $\lambda=1$, then we obtain

$$\operatorname{tr}(\mathbf{A}_i^{J_i})\leq\operatorname{tr}\left(\left(1-2\eta_{m,k}\lambda_{\min}\left(\mathbf{\Lambda}_{m,h}^{k}\right)\right)^{J_j}\mathbf{I}\right)\leq d\left(1-2\eta_{m,k}\lambda_{\min}\left(\mathbf{\Lambda}_{m,h}^{k}\right)\right)^{J_j}\leq d\epsilon\leq 1.$$

Similarly, we have $\mathbf{I}-\mathbf{A}_i^{2J_i}\preccurlyeq\left(1-\frac{1}{2^{2J_i}}\right)\mathbf{I}$, then we get

$$\operatorname{tr}(\mathbf{I}-\mathbf{A}_i^{2J_i})\leq\left(1-\frac{1}{2^{2J_i}}\right)d<d.$$

Also, based on $(\mathbf{I}+\mathbf{A}_i)^{-1}\preccurlyeq\frac{2}{3}\mathbf{I}$, we have

$$\operatorname{tr}\left((\mathbf{I}+\mathbf{A}_i)^{-1}\right)\leq\frac{2}{3}d.$$

Note that $\lambda_{\max}\left(\left(\mathbf{\Lambda}_{m,h}^{i}\right)^{-1}\right)\leq 1$, we have

$$\operatorname{tr}\left(\left(\mathbf{\Lambda}_{m,h}^{i}\right)^{-1}\right)\leq\sum\lambda\left(\left(\mathbf{\Lambda}_{m,h}^{i}\right)^{-1}\right)\leq d.$$

Combine the above results together and choose $\beta_{m,i}=\beta_K$ for all $i\in[K]$ and $m\in\mathcal{M}$, we have

$$\operatorname{tr}\left(\mathbf{\Sigma}_{m,h}^{k,J_k}\right)\leq\sum_{i=1}^{K}\frac{1}{\beta_{m,i}}\cdot\frac{2}{3}\cdot d^3=\frac{2}{3\beta_K}Kd^3.$$

Then we have

$$\mathbb{P}\left(\big\|\boldsymbol{\xi}_{m,h}^{k,J_k}\big\| \le \sqrt{\frac{1}{\delta} \cdot \frac{2}{3\beta_K} K d^3}\right) \ge \mathbb{P}\left(\big\|\boldsymbol{\xi}_{m,h}^{k,J_k}\big\| \le \sqrt{\frac{1}{\delta}\operatorname{tr}\left(\boldsymbol{\Sigma}_{m,h}^{k,J_k}\right)}\right) \ge 1 - \delta.$$

**Combine above results together:** with probability at least $1 - \delta$, we have

$$\big\|\mathbf{w}_{m,h}^{k,J_k}\big\| \le \frac{16}{3} H d\sqrt{MK} + \sqrt{\frac{2K}{3\beta_K\delta}} d^{3/2}.$$

This completes the proof. □

## E.5 Proof of Lemma D.7

*Proof.* Based on Lemma D.6, for any fixed $n \in [N]$, with probability at least $1 - \delta$, for any $(m, k, h) \in \mathcal{M} \times [K] \times [H]$, we have

$$\big\|\mathbf{w}_{m,h}^{k,J_k,n}\big\| \le \frac{16}{3} H d\sqrt{MK} + \sqrt{\frac{2K}{3\beta_K\delta}} d^{3/2}.$$

By taking union over $n, m, k, h$, we have for all $(m, k, h) \in \mathcal{M} \times [K] \times [H]$ and for all $n \in [N]$, with probability $1 - \delta/2$, we have

$$\big\|\mathbf{w}_{m,h}^{k,J_k,n}\big\| \le \frac{16}{3} H d\sqrt{MK} + \sqrt{\frac{4NMHK^2}{3\beta_K\delta}} d^{3/2} = B_{\delta/2NMHK}. \tag{E.7}$$

Based on Lemma J.7 and Lemma J.9, we have that for any $\varepsilon > 0$ and $\delta > 0$, with probability at least $1 - \delta/2$,

$$\left\|\sum_{(s^l,a^l,s'^l)\in U_{m,h}(k)} \phi(s^l, a^l)\left[\left(V_{m,h+1}^k - \mathbb{P}_h V_{m,h+1}^k\right)(s^l, a^l)\right]\right\|_{(\boldsymbol{\Lambda}_{m,h}^k)^{-1}}$$

$$\le \left(4H^2\left[\frac{d}{2}\log\left(\frac{k+\lambda}{\lambda}\right) + d\log\left(\frac{B_{\delta/2NMHK}}{\varepsilon}\right) + \log\frac{3}{\delta}\right] + \frac{8k^2\varepsilon^2}{\lambda}\right)^{1/2}$$

$$\le 2H\left[\frac{d}{2}\log\left(\frac{k+\lambda}{\lambda}\right) + d\log\left(\frac{B_{\delta/2NMHK}}{\varepsilon}\right) + \log\frac{3}{\delta}\right]^{1/2} + \frac{2\sqrt{2}k\varepsilon}{\sqrt{\lambda}}.$$

Here we set $\lambda = 1, \varepsilon = \frac{H}{2\sqrt{2}k}$, with probability at least $1 - \delta/2$, we have

$$\left\|\sum_{(s^l,a^l,s'^l)\in U_{m,h}(k)} \phi(s^l, a^l)\left[\left(V_{m,h+1}^k - \mathbb{P}_h V_{m,h+1}^k\right)(s^l, a^l)\right]\right\|_{(\boldsymbol{\Lambda}_{m,h}^k)^{-1}}$$

$$\le 2H\sqrt{d}\left[\frac{1}{2}\log(k+1) + \log\left(\frac{B_{\delta/2NMHK}}{\frac{H}{2\sqrt{2}k}}\right) + \log\frac{3}{\delta}\right]^{1/2} + H$$

$$\le 3H\sqrt{d}\left[\frac{1}{2}\log(K+1) + \log\left(\frac{2\sqrt{2}KB_{\delta/2NMHK}}{H}\right) + \log\frac{3}{\delta}\right]^{1/2}. \tag{E.8}$$

By applying union bound between (E.7) and (E.8), and define that $C_\delta = \left[\frac{1}{2}\log(K+1) + \log\frac{3}{\delta} + \log\left(\frac{2\sqrt{2}KB_{\delta/2NMHK}}{H}\right)\right]^{1/2}$, finally we obtain that for all $(m, k, h) \in \mathcal{M} \times [K] \times [H]$,

$$\left\|\sum_{(s^l,a^l,s'^l)\in U_{m,h}(k)} \phi(s^l, a^l)\left[\left(V_{m,h+1}^k - \mathbb{P}_h V_{m,h+1}^k\right)(s^l, a^l)\right]\right\|_{(\boldsymbol{\Lambda}_{m,h}^k)^{-1}} \le 3H\sqrt{d}C_\delta,$$

with probability at least $1 - \delta$. □

## E.6 Proof of Lemma D.8

*Proof.* We denote the inner product over $\mathcal{S}$ by $\langle \cdot, \cdot \rangle_{\mathcal{S}}$. Based on $\mathbb{P}_h(\cdot|s,a) = \langle \phi(s,a), \mu_h(\cdot) \rangle_{\mathcal{S}}$ in Definition 4.1, we have

$$
\begin{aligned}
\mathbb{P}_h V_{m,h+1}^k(s,a) &= \phi(s,a)^\top \langle \mu_h, V_{m,h+1}^k \rangle_{\mathcal{S}} \\
&= \phi(s,a)^\top \left(\Lambda_{m,h}^k\right)^{-1} \left(\Lambda_{m,h}^k\right) \langle \mu_h, V_{m,h+1}^k \rangle_{\mathcal{S}} \\
&= \phi(s,a)^\top \left(\Lambda_{m,h}^k\right)^{-1} \left( \sum_{(s^l,a^l,s'^l)\in U_{m,h}(k)} \phi(s^l,a^l)\phi(s^l,a^l)^\top + \lambda \mathbf{I} \right) \langle \mu_h, V_{m,h+1}^k \rangle_{\mathcal{S}} \\
&= \phi(s,a)^\top \left(\Lambda_{m,h}^k\right)^{-1} \left( \sum_{(s^l,a^l,s'^l)\in U_{m,h}(k)} \phi(s^l,a^l)\left(\mathbb{P}_h V_{m,h+1}^k\right)(s^l,a^l) + \lambda \mathbf{I} \langle \mu_h, V_{m,h+1}^k \rangle_{\mathcal{S}} \right).
\end{aligned}
\tag{E.9}
$$

Here the last equality uses $\mathbb{P}_h(\cdot|s,a) = \langle \phi(s,a), \mu_h(\cdot) \rangle_{\mathcal{S}}$ again. Then we can separate the following error into three parts,

$$
\begin{aligned}
&\phi(s,a)^\top \widehat{\mathbf{w}}_{m,h}^k - r_h(s,a) - \mathbb{P}_h V_{m,h+1}^k(s,a) \\
&= \phi(s,a)^\top \left(\Lambda_{m,h}^k\right)^{-1} \sum_{(s^l,a^l,s'^l)\in U_{m,h}(k)} \left[ r_h(s^l,a^l) + V_{m,h+1}^k(s'^l) \right] \phi(s^l,a^l) - r_h(s,a) \\
&\quad - \phi(s,a)^\top \left(\Lambda_{m,h}^k\right)^{-1} \left( \sum_{(s^l,a^l,s'^l)\in U_{m,h}(k)} \phi(s^l,a^l)\left(\mathbb{P}_h V_{m,h+1}^k\right)(s^l,a^l) + \lambda \mathbf{I} \langle \mu_h, V_{m,h+1}^k \rangle_{\mathcal{S}} \right) \\
&= \underbrace{\phi(s,a)^\top \left(\Lambda_{m,h}^k\right)^{-1} \left( \sum_{(s^l,a^l,s'^l)\in U_{m,h}(k)} \phi(s^l,a^l)\left[\left(V_{m,h+1}^k - \mathbb{P}_h V_{m,h+1}^k\right)(s^l,a^l)\right] \right)}_{(i)} \\
&\quad + \underbrace{\phi(s,a)^\top \left(\Lambda_{m,h}^k\right)^{-1} \left( \sum_{(s^l,a^l,s'^l)\in U_{m,h}(k)} r_h(s^l,a^l)\phi(s^l,a^l) \right) - r_h(s,a)}_{(ii)} \\
&\quad - \underbrace{\lambda \phi(s,a)^\top \left(\Lambda_{m,h}^k\right)^{-1} \langle \mu_h, V_{m,h+1}^k \rangle_{\mathcal{S}}}_{(iii)}.
\end{aligned}
\tag{E.10}
$$

Here the first equality holds due to (E.9). We now provide an upper bound for each of the terms in (E.10).

**Bounding Term (i) in** (E.10): using Cauchy-Schwarz inequality and Lemma D.7, with probability at least $1 - \delta$, for all $(m,k,h) \in \mathcal{M} \times [K] \times [H]$ and for any $(s,a) \in \mathcal{S} \times \mathcal{A}$, we have

$$
\begin{aligned}
&\phi(s,a)^\top \left(\Lambda_{m,h}^k\right)^{-1} \left( \sum_{(s^l,a^l,s'^l)\in U_{m,h}(k)} \phi(s^l,a^l)\left[\left(V_{m,h+1}^k - \mathbb{P}_h V_{m,h+1}^k\right)(s^l,a^l)\right] \right) \\
&\leq \left\| \sum_{(s^l,a^l,s'^l)\in U_{m,h}(k)} \phi(s^l,a^l)\left[\left(V_{m,h+1}^k - \mathbb{P}_h V_{m,h+1}^k\right)(s^l,a^l)\right] \right\|_{\left(\Lambda_{m,h}^k\right)^{-1}} \|\phi(s,a)\|_{\left(\Lambda_{m,h}^k\right)^{-1}} \\
&\leq 3H\sqrt{d}C_\delta \|\phi(s,a)\|_{\left(\Lambda_{m,h}^k\right)^{-1}}.
\end{aligned}
\tag{E.11}
$$

**Bounding Term (ii) in** (E.10): we first note that

$$
\begin{aligned}
&\phi(s,a)^\top \left(\Lambda_{m,h}^k\right)^{-1} \left( \sum_{(s^l,a^l,s'^l)\in U_{m,h}(k)} r_h(s^l,a^l)\phi(s^l,a^l) \right) - r_h(s,a) \\
&= \phi(s,a)^\top \left(\Lambda_{m,h}^k\right)^{-1} \left( \sum_{(s^l,a^l,s'^l)\in U_{m,h}(k)} r_h(s^l,a^l)\phi(s^l,a^l) \right) - \phi(s,a)^\top \theta_h
\end{aligned}
$$

$$= \phi(s,a)^\top \big(\mathbf{\Lambda}_{m,h}^k\big)^{-1} \left( \sum_{(s^l,a^l,s'^l) \in U_{m,h}(k)} r_h\big(s^l,a^l\big)\phi\big(s^l,a^l\big) - \mathbf{\Lambda}_{m,h}^k\boldsymbol{\theta}_h \right)$$

$$= \phi(s,a)^\top \big(\mathbf{\Lambda}_{m,h}^k\big)^{-1} \left( \sum_{(s^l,a^l,s'^l) \in U_{m,h}(k)} r_h\big(s^l,a^l\big)\phi\big(s^l,a^l\big) - \sum_{(s^l,a^l,s'^l) \in U_{m,h}(k)} \phi\big(s^l,a^l\big)\phi\big(s^l,a^l\big)^\top\boldsymbol{\theta}_h - \lambda\mathbf{I}\boldsymbol{\theta}_h \right)$$

$$= \phi(s,a)^\top \big(\mathbf{\Lambda}_{m,h}^k\big)^{-1} \left( \sum_{(s^l,a^l,s'^l) \in U_{m,h}(k)} r_h\big(s^l,a^l\big)\phi\big(s^l,a^l\big) - \sum_{(s^l,a^l,s'^l) \in U_{m,h}(k)} \phi\big(s^l,a^l\big) r_h\big(s^l,a^l\big) - \lambda\boldsymbol{\theta}_h \right)$$

$$= -\lambda\phi(s,a)^\top \big(\mathbf{\Lambda}_{m,h}^k\big)^{-1}\boldsymbol{\theta}_h, \tag{E.12}$$

where the first and fourth equality holds due to the definition $r_h(s,a) = \big\langle \phi(s,a), \boldsymbol{\theta}_h \big\rangle$ from Definition 4.1, the third equality uses the definition of $\mathbf{\Lambda}_{m,h}^k$. Next we can obtain that

$$-\lambda\phi(s,a)^\top \big(\mathbf{\Lambda}_{m,h}^k\big)^{-1}\boldsymbol{\theta}_h \leq \lambda\|\phi(s,a)\|_{(\mathbf{\Lambda}_{m,h}^k)^{-1}}\|\boldsymbol{\theta}_h\|_{(\mathbf{\Lambda}_{m,h}^k)^{-1}}$$
$$\leq \sqrt{\lambda}\|\phi(s,a)\|_{(\mathbf{\Lambda}_{m,h}^k)^{-1}}\|\boldsymbol{\theta}_h\|$$
$$\leq \sqrt{\lambda d}\|\phi(s,a)\|_{(\mathbf{\Lambda}_{m,h}^k)^{-1}}, \tag{E.13}$$

where we use the fact that $\lambda_{\max}\big(\big(\mathbf{\Lambda}_{m,h}^k\big)^{-1}\big) \leq 1/\lambda$ and $\|\boldsymbol{\theta}_h\| \leq \sqrt{d}$ from Definition 4.1. By Combining (E.12) and (E.13), we obtain

$$\phi(s,a)^\top \big(\mathbf{\Lambda}_{m,h}^k\big)^{-1} \left( \sum_{(s^l,a^l,s'^l) \in U_{m,h}(k)} r_h\big(s^l,a^l\big)\phi\big(s^l,a^l\big) \right) - r_h(s,a) \leq \sqrt{\lambda d}\|\phi(s,a)\|_{(\mathbf{\Lambda}_{m,h}^k)^{-1}}.$$
$$\tag{E.14}$$

**Bounding Term (iii) in** (E.10): we have

$$\lambda\phi(s,a)^\top \big(\mathbf{\Lambda}_{m,h}^k\big)^{-1}\big\langle \boldsymbol{\mu}_h, V_{m,h+1}^k \big\rangle_{\mathcal{S}} \leq \lambda\|\phi(s,a)\|_{(\mathbf{\Lambda}_{m,h}^k)^{-1}}\big\|\big\langle \boldsymbol{\mu}_h, V_{m,h+1}^k \big\rangle_{\mathcal{S}}\big\|_{(\mathbf{\Lambda}_{m,h}^k)^{-1}}$$
$$\leq \sqrt{\lambda}\|\phi(s,a)\|_{(\mathbf{\Lambda}_{m,h}^k)^{-1}}\big\|\big\langle \boldsymbol{\mu}_h, V_{m,h+1}^k \big\rangle_{\mathcal{S}}\big\|$$
$$\leq H\sqrt{\lambda}\|\phi(s,a)\|_{(\mathbf{\Lambda}_{m,h}^k)^{-1}}\|\boldsymbol{\mu}_h\|$$
$$\leq H\sqrt{\lambda d}\|\phi(s,a)\|_{(\mathbf{\Lambda}_{m,h}^k)^{-1}}, \tag{E.15}$$

where the second inequality holds due to the fact that $\lambda_{\max}\big(\big(\mathbf{\Lambda}_{m,h}^k\big)^{-1}\big) \leq 1/\lambda$, the third inequality uses the fact that $V_{m,h+1}^k \leq H$ and the last inequality follows from $\|\boldsymbol{\mu}_h\| \leq \sqrt{d}$ in Definition 4.1.

**Combine Terms (i)(ii)(iii) together:** combine (E.11), (E.14) and (E.15), then set $\lambda = 1$, with probability at least $1 - \delta$, we get

$$\left| \phi(s,a)^\top \widehat{\mathbf{w}}_{m,h}^k - r_h(s,a) - \mathbb{P}_h V_{m,h+1}^k(s,a) \right| \leq \big( 3HC_\delta + \sqrt{\lambda d} + H\sqrt{\lambda d} \big)\|\phi(s,a)\|_{(\mathbf{\Lambda}_{m,h}^k)^{-1}}$$
$$\leq 5H\sqrt{d}C_\delta\|\phi(s,a)\|_{(\mathbf{\Lambda}_{m,h}^k)^{-1}},$$

This completes the proof. $\qquad\qquad\square$

### E.7 Proof of Lemma D.9

*Proof.* Recall from Definition D.1,

$$-l_{m,h}^k(s,a) = Q_{m,h}^k(s,a) - r_h(s,a) - \mathbb{P}_h V_{m,h+1}^k(s,a)$$
$$= \min\left\{ \max_{n \in [N]} \phi(s,a)^\top \mathbf{w}_{m,h}^{k,J_k,n}, H - h + 1 \right\}^+ - r_h(s,a) - \mathbb{P}_h V_{m,h+1}^k(s,a)$$
$$\leq \max_{n \in [N]} \phi(s,a)^\top \mathbf{w}_{m,h}^{k,J_k,n} - r_h(s,a) - \mathbb{P}_h V_{m,h+1}^k(s,a)$$
$$= \max_{n \in [N]} \phi(s,a)^\top \mathbf{w}_{m,h}^{k,J_k,n} - \phi(s,a)^\top \widehat{\mathbf{w}}_{m,h}^k + \phi(s,a)^\top \widehat{\mathbf{w}}_{m,h}^k - r_h(s,a) - \mathbb{P}_h V_{m,h+1}^k(s,a)$$

$$\leq \underbrace{\max_{n \in [N]} \left| \phi(s,a)^\top \mathbf{w}_{m,h}^{k,J_k,n} - \phi(s,a)^\top \widehat{\mathbf{w}}_{m,h}^k \right|}_{I_1} + \underbrace{\left| \phi(s,a)^\top \widehat{\mathbf{w}}_{m,h}^k - r_h(s,a) - \mathbb{P}_h V_{m,h+1}^k(s,a) \right|}_{I_2}.$$

**Bounding Term $I_1$:** based on Lemma D.5, for any fixed $n \in [N]$, for any $(m,h,k) \in \mathcal{M} \times [H] \times [K]$ and for any $(s,a) \in \mathcal{S} \times \mathcal{A}$, with probability at least $1 - \delta^2$, we have

$$\left| \phi(s,a)^\top \mathbf{w}_{m,h}^{k,J_k,n} - \phi(s,a)^\top \widehat{\mathbf{w}}_{m,h}^k \right| \leq \left( 5 \sqrt{\frac{2d \log(1/\delta)}{3\beta_K}} + \frac{4}{3} \right) \| \phi(s,a) \|_{(\mathbf{\Lambda}_{m,h}^k)^{-1}}.$$

By taking union bound over $n$, we have for all $n \in [N]$, with probability $1 - \delta^2$, we have

$$\left| \phi(s,a)^\top \mathbf{w}_{m,h}^{k,J_k,n} - \phi(s,a)^\top \widehat{\mathbf{w}}_{m,h}^k \right| \leq \left( 5 \sqrt{\frac{2d \log\left(\sqrt{N}/\delta\right)}{3\beta_K}} + \frac{4}{3} \right) \| \phi(s,a) \|_{(\mathbf{\Lambda}_{m,h}^k)^{-1}}.$$

This indicates, for any $(m,h,k) \in \mathcal{M} \times [H] \times [K]$ and $(s,a) \in \mathcal{S} \times \mathcal{A}$, with probability at least $1 - \delta^2$, we have

**Term** $I_1 = \max_{n \in [N]} \left| \phi(s,a)^\top \mathbf{w}_{m,h}^{k,J_k,n} - \phi(s,a)^\top \widehat{\mathbf{w}}_{m,h}^k \right| \leq \left( 5 \sqrt{\frac{2d \log\left(\sqrt{N}/\delta\right)}{3\beta_K}} + \frac{4}{3} \right) \| \phi(s,a) \|_{(\mathbf{\Lambda}_{m,h}^k)^{-1}}.$

$$(\text{E.16})$$

**Bounding Term $I_2$:** based on Lemma D.8, with probability at least $1 - \delta$, for any $(m,h,k) \in \mathcal{M} \times [H] \times [K]$ and $(s,a) \in \mathcal{S} \times \mathcal{A}$, we have

$$\left| \phi(s,a)^\top \widehat{\mathbf{w}}_{m,h}^k - r_h^k(s,a) - \mathbb{P}_h V_{m,h+1}^k(s,a) \right| \leq 5H\sqrt{d} C_\delta \| \phi(s,a) \|_{(\mathbf{\Lambda}_{m,h}^k)^{-1}}.$$

Combine the two result above, by taking union bound, with probability at least $1 - \delta - \delta^2$, for any $(m,h,k) \in \mathcal{M} \times [H] \times [K]$ and $(s,a) \in \mathcal{S} \times \mathcal{A}$, we have

$$-l_{m,h}^k(s,a) \leq \left( 5H\sqrt{d} C_\delta + 5 \sqrt{\frac{2d \log\left(\sqrt{N}/\delta\right)}{3\beta_K}} + \frac{4}{3} \right) \| \phi(s,a) \|_{(\mathbf{\Lambda}_{m,h}^k)^{-1}}.$$

This completes the proof. $\qquad\square$

### E.8 Proof of Lemma D.10

*Proof.* Recall from Definition D.1,

$$l_{m,h}^k(s,a) = r_h(s,a) + \mathbb{P}_h V_{m,h+1}^k(s,a) - Q_{m,h}^k(s,a).$$

Note that

$$Q_{m,h}^k(s,a) = \min \left\{ \max_{n \in [N]} \phi(s,a)^\top \mathbf{w}_{m,h}^{k,J_k,n}, H - h + 1 \right\}^+ \leq \max_{n \in [N]} \phi(x,a)^\top \mathbf{w}_{m,h}^{k,J_k,n}.$$

Note that $\| \phi(s,a) \|_{(\mathbf{\Lambda}_{m,h}^k)^{-1}} \leq \sqrt{1/\lambda} \| \phi(s,a) \| \leq 1$ for all $\phi(s,a)$. Define $\mathcal{C}(\varepsilon)$ to be a $\varepsilon$-cover of $\left\{ \phi \mid \| \phi \|_{(\mathbf{\Lambda}_{m,h}^k)^{-1}} \leq 1 \right\}$. Based on Lemma J.8, we have $|\mathcal{C}(\varepsilon)| \leq (3/\varepsilon)^d$.

First, for any fixed $\phi(s,a) \in \mathcal{C}(\varepsilon)$, based on the results in Proposition D.3, we have that $\phi(s,a)^\top \mathbf{w}_{m,h}^{k,J_k,n} \sim \mathcal{N}\left( \phi(s,a)^\top \boldsymbol{\mu}_{m,h}^{k,J_k}, \phi(s,a)^\top \mathbf{\Sigma}_{m,h}^{k,J_k} \phi(s,a) \right)$ for any fixed $n \in [N]$. Now we define

$$Z_k = \frac{r_h(s,a) + \mathbb{P}_h V_{m,h+1}^k(s,a) - \phi(s,a)^\top \boldsymbol{\mu}_{m,h}^{k,J_k}}{\sqrt{\phi(s,a)^\top \mathbf{\Sigma}_{m,h}^{k,J_k} \phi(s,a)}}.$$

When $|Z_k| < 1$, by Gaussian concentration Lemma J.10, we have

$$\mathbb{P}\left( \phi(s,a)^\top \mathbf{w}_{m,h}^{k,J_k,n} \geq r_h(s,a) + \mathbb{P}_h V_{m,h+1}^k(s,a) \right)$$

$$= \mathbb{P}\left(\frac{\phi(s,a)^\top \mathbf{w}_{m,h}^{k,J_k,n} - \phi(s,a)^\top \boldsymbol{\mu}_{m,h}^{k,J_k}}{\sqrt{\phi(s,a)^\top \boldsymbol{\Sigma}_{m,h}^{k,J_k}\phi(s,a)}} \geq \frac{r_h(s,a) + \mathbb{P}_h V_{m,h+1}^k(s,a) - \phi(s,a)^\top \boldsymbol{\mu}_{m,h}^{k,J_k}}{\sqrt{\phi(s,a)^\top \boldsymbol{\Sigma}_{m,h}^{k,J_k}\phi(s,a)}}\right)$$

$$= \mathbb{P}\left(\frac{\phi(s,a)^\top \mathbf{w}_{m,h}^{k,J_k,n} - \phi(s,a)^\top \boldsymbol{\mu}_{m,h}^{k,J_k}}{\sqrt{\phi(s,a)^\top \boldsymbol{\Sigma}_{m,h}^{k,J_k}\phi(s,a)}} \geq Z_k\right)$$

$$\geq \frac{1}{2\sqrt{2\pi}}\exp(-Z_k^2/2)$$

$$\geq \frac{1}{2\sqrt{2e\pi}}.$$

**Consider the numerator of $Z_k$:**

$$\left| r_h(s,a) + \mathbb{P}_h V_{m,h+1}^k(s,a) - \phi(s,a)^\top \boldsymbol{\mu}_{m,h}^{k,J_k}\right|$$
$$\leq \left| r_h(s,a) + \mathbb{P}_h V_{m,h+1}^k(s,a) - \phi(s,a)^\top \widehat{\mathbf{w}}_{m,h}^k\right| + \left|\phi(s,a)^\top \widehat{\mathbf{w}}_{m,h}^k - \phi(s,a)^\top \boldsymbol{\mu}_{m,h}^{k,J_k}\right|.$$

Based on Lemma D.8, with probablity at least $1-\delta$, we have

$$\left| r_h(s,a) + \mathbb{P}_h V_{m,h+1}^k(s,a) - \phi(s,a)^\top \widehat{\mathbf{w}}_{m,h}^k\right| \leq 5H\sqrt{d}C_\delta \|\phi(s,a)\|_{(\boldsymbol{\Lambda}_{m,h}^k)^{-1}},$$

From (E.4), we have

$$\phi(s,a)^\top\left(\boldsymbol{\mu}_{m,h}^{k,J_k} - \widehat{\mathbf{w}}_{m,h}^k\right) \leq 4H\sqrt{MKd/\lambda}\sum_{i=1}^{k-1}\prod_{j=i+1}^k \left(1 - 2\eta_{m,j}\lambda_{\min}(\boldsymbol{\Lambda}_{m,h}^j)\right)^{J_j}\|\phi(s,a)\|.$$

Recall the proof of Lemma D.5, we set $\eta_{m,j} = 1/(4\lambda_{\max}(\boldsymbol{\Lambda}_{m,h}^j))$, $J_j \geq 2\kappa_j\log(1/\epsilon)$, then we have for all $j \in [K]$, $(1 - 2\eta_{m,j}\lambda_{\min}(\boldsymbol{\Lambda}_{m,h}^j))^{J_j} \leq \epsilon$, set $\epsilon = 1/4HMKd$ and $\lambda = 1$, we have

$$\left|\phi(s,a)^\top \widehat{\mathbf{w}}_{m,h}^k - \phi(s,a)^\top \boldsymbol{\mu}_{m,h}^{k,J_k}\right| \leq 4H\sqrt{MKd}\sum_{i=1}^{k-1}\epsilon^{k-i}\|\phi(s,a)\|$$

$$\leq \sum_{i=1}^{k-1}\epsilon^{k-i-1}\frac{1}{4MHKd}4H\sqrt{MKd}\sqrt{MK}\|\phi(s,a)\|_{(\boldsymbol{\Lambda}_{m,h}^k)^{-1}}$$

$$\leq \frac{4}{3}\|\phi(s,a)\|_{(\boldsymbol{\Lambda}_{m,h}^k)^{-1}}.$$

So, with probablity at least $1-\delta$, we have

$$\left| r_h(s,a) + \mathbb{P}_h V_{m,h+1}^k(s,a) - \phi(s,a)^\top \boldsymbol{\mu}_{m,h}^{k,J_k}\right| \leq \left(5H\sqrt{d}C_\delta + \frac{4}{3}\right)\|\phi(s,a)\|_{(\boldsymbol{\Lambda}_{m,h}^k)^{-1}}. \quad \text{(E.17)}$$

**Consider the denominator of $Z_k$:** recall from the definition of $\boldsymbol{\Sigma}_{m,h}^{k,J_k}$ from Proposition D.3, then we have

$$\phi(s,a)^\top \boldsymbol{\Sigma}_{m,h}^{k,J_k}\phi(s,a)$$

$$= \sum_{i=1}^k \frac{1}{\beta_{m,i}}\phi(s,a)^\top \mathbf{A}_k^{J_k}\dots\mathbf{A}_{i+1}^{J_{i+1}}\left(\mathbf{I} - \mathbf{A}^{2J_i}\right)\left(\boldsymbol{\Lambda}_{m,h}^i\right)^{-1}(\mathbf{I}+\mathbf{A}_i)^{-1}\mathbf{A}_{i+1}^{J_{i+1}}\dots\mathbf{A}_k^{J_k}\phi(s,a)$$

$$\geq \sum_{i=1}^k \frac{1}{2\beta_{m,i}}\phi(s,a)^\top \mathbf{A}_k^{J_k}\dots\mathbf{A}_{i+1}^{J_{i+1}}\left(\mathbf{I} - \mathbf{A}^{2J_i}\right)\left(\boldsymbol{\Lambda}_{m,h}^i\right)^{-1}\mathbf{A}_{i+1}^{J_{i+1}}\dots\mathbf{A}_k^{J_k}\phi(s,a),$$

where we used the fact that $\frac{1}{2}\mathbf{I} \preccurlyeq (\mathbf{I}+\mathbf{A}_k)^{-1}$. Then we have

$$\phi(s,a)^\top \boldsymbol{\Sigma}_{m,h}^{k,J_k}\phi(s,a)$$

$$\geq \sum_{i=1}^k \frac{1}{2\beta_{m,i}}\phi(s,a)^\top \mathbf{A}_k^{J_k}\dots\mathbf{A}_{i+1}^{J_{i+1}}\left(\left(\boldsymbol{\Lambda}_{m,h}^i\right)^{-1} - \mathbf{A}_i^{J_i}\left(\boldsymbol{\Lambda}_{m,h}^i\right)^{-1}\mathbf{A}_i^{J_i}\right)\mathbf{A}_{i+1}^{J_{i+1}}\dots\mathbf{A}_k^{J_k}\phi(s,a)$$

$$= \frac{1}{2\beta_K} \sum_{i=1}^{k-1} \phi(s,a)^\top \mathbf{A}_k^{J_k} \ldots \mathbf{A}_{i+1}^{J_{i+1}} \left( \left( \mathbf{\Lambda}_{m,h}^i \right)^{-1} - \left( \mathbf{\Lambda}_{m,h}^{i+1} \right)^{-1} \right) \mathbf{A}_{i+1}^{J_{i+1}} \ldots \mathbf{A}_k^{J_k} \phi(s,a)$$

$$- \frac{1}{2\beta_K} \phi(s,a)^\top \mathbf{A}_k^{J_k} \ldots \mathbf{A}_1^{J_1} \left( \mathbf{\Lambda}_{m,h}^1 \right)^{-1} \mathbf{A}_1^{J_1} \ldots \mathbf{A}_k^{J_k} \phi(s,a)$$

$$+ \frac{1}{2\beta_K} \phi(s,a)^\top \left( \mathbf{\Lambda}_{m,h}^k \right)^{-1} \phi(s,a).$$

By the definition of $\mathbf{\Lambda}_{m,h}^i$ and Woodbury formula, we have

$$\left( \mathbf{\Lambda}_{m,h}^i \right)^{-1} - \left( \mathbf{\Lambda}_{m,h}^{i+1} \right)^{-1} = \left( \mathbf{\Lambda}_{m,h}^i \right)^{-1} - \left( \mathbf{\Lambda}_{m,h}^i + \sum_{(s^l,a^l,s'^l) \in U_{m,h}(k)} \phi(s^l,a^l) \phi(s^l,a^l)^\top \right)^{-1}$$

$$= (\mathbf{\Lambda}_{m,h}^i)^{-1} \boldsymbol{\varphi} \left( \mathbf{I}_n + \boldsymbol{\varphi}^\top (\mathbf{\Lambda}_{m,h}^i)^{-1} \boldsymbol{\varphi} \right)^{-1} \boldsymbol{\varphi}^\top (\mathbf{\Lambda}_{m,h}^i)^{-1},$$

where $\boldsymbol{\varphi}$ is a matrix with the dimension of $d \times n$, $n$ is the number difference of $\phi(s^l,a^l)$ between $\left( \mathbf{\Lambda}_{m,h}^i \right)^{-1}$ and $\left( \mathbf{\Lambda}_{m,h}^{i+1} \right)^{-1}$ (i.e. we concatenate all $\phi(s^l,a^l)$ in to the matrix $\boldsymbol{\varphi}$). Note that $n \leq M$, we have

$$\phi(s,a)^\top \mathbf{A}_k^{J_k} \ldots \mathbf{A}_{i+1}^{J_{i+1}} \left( \left( \mathbf{\Lambda}_{m,h}^i \right)^{-1} - \left( \mathbf{\Lambda}_{m,h}^{i+1} \right)^{-1} \right) \mathbf{A}_{i+1}^{J_{i+1}} \ldots \mathbf{A}_k^{J_k} \phi(s,a)$$

$$= \phi(s,a)^\top \mathbf{A}_k^{J_k} \ldots \mathbf{A}_{i+1}^{J_{i+1}} \left( \left( \mathbf{\Lambda}_{m,h}^i \right)^{-1} \boldsymbol{\varphi} (\mathbf{I}_n + \boldsymbol{\varphi}^\top \left( \mathbf{\Lambda}_{m,h}^i \right)^{-1} \boldsymbol{\varphi})^{-1} \boldsymbol{\varphi}^\top \left( \mathbf{\Lambda}_{m,h}^i \right)^{-1} \right) \mathbf{A}_{i+1}^{J_{i+1}} \ldots \mathbf{A}_k^{J_k} \phi(s,a)$$

$$\leq \phi(s,a)^\top \mathbf{A}_k^{J_k} \ldots \mathbf{A}_{i+1}^{J_{i+1}} \left( \mathbf{\Lambda}_{m,h}^i \right)^{-1} \boldsymbol{\varphi} \boldsymbol{\varphi}^\top \left( \mathbf{\Lambda}_{m,h}^i \right)^{-1} \mathbf{A}_{i+1}^{J_{i+1}} \ldots \mathbf{A}_k^{J_k} \phi(s,a)$$

$$= \left\| \phi(s,a)^\top \mathbf{A}_k^{J_k} \ldots \mathbf{A}_{i+1}^{J_{i+1}} \left( \mathbf{\Lambda}_{m,h}^i \right)^{-1} \boldsymbol{\varphi} \right\|_2^2$$

$$\leq \left\| \mathbf{A}_k^{J_k} \ldots \mathbf{A}_{i+1}^{J_{i+1}} (\mathbf{\Lambda}_{m,h}^i)^{-1/2} \phi(s,a) \right\|_2^2 \cdot \left\| (\mathbf{\Lambda}_{m,h}^i)^{-1/2} \boldsymbol{\varphi} \right\|_F^2$$

$$\leq \prod_{j=i+1}^{k} \left( 1 - 2\eta_{m,j} \lambda_{\min} \left( \mathbf{\Lambda}_{m,h}^j \right) \right)^{2J_j} \operatorname{tr} \left( \boldsymbol{\varphi}^\top \left( \mathbf{\Lambda}_{m,h}^i \right)^{-1} \boldsymbol{\varphi} \right) \|\phi(s,a)\|_{(\mathbf{\Lambda}_{m,h}^i)^{-1}}^2,$$

where $\|\cdot\|_F$ is Frobenius norm and the last inequality is due to $\|\mathbf{\Lambda}^{-\frac{1}{2}} \mathbf{X}\|_F^2 = \operatorname{tr}(\mathbf{X}^\top \mathbf{\Lambda}^{-1} \mathbf{X})$ and (E.2). Therefore, we have

$$\phi(s,a)^\top \mathbf{\Sigma}_{m,h}^{k,J_k} \phi(s,a)$$

$$\geq \frac{1}{2\beta_K} \|\phi(s,a)\|_{(\mathbf{\Lambda}_{m,h}^k)^{-1}}^2 - \frac{1}{2\beta_K} \prod_{i=1}^{k} \left( 1 - 2\eta_{m,i} \lambda_{\min} \left( \mathbf{\Lambda}_{m,h}^i \right) \right)^{2J_i} \|\phi(s,a)\|_{(\mathbf{\Lambda}_{m,h}^1)^{-1}}^2$$

$$- \frac{1}{2\beta_K} \sum_{i=1}^{k-1} \prod_{j=i+1}^{k} \left( 1 - 2\eta_{m,j} \lambda_{\min} \left( \mathbf{\Lambda}_{m,h}^j \right) \right)^{2J_j} \operatorname{tr} \left( \boldsymbol{\varphi}^\top \left( \mathbf{\Lambda}_{m,h}^i \right)^{-1} \boldsymbol{\varphi} \right) \|\phi(s,a)\|_{(\mathbf{\Lambda}_{m,h}^i)^{-1}}^2.$$

Similar to the proof of Lemma D.5, note that $\operatorname{tr} \left( \boldsymbol{\varphi}^\top (\mathbf{\Lambda}_{m,h}^i)^{-1} \boldsymbol{\varphi} \right) \leq M$, when we choose $J_j \geq 2\kappa_j \log(3kM)$, we have

$$\|\phi(s,a)\|_{\mathbf{\Sigma}_{m,h}^{k,J_k}} \geq \frac{1}{2\sqrt{\beta_K}} \left( \|\phi(s,a)\|_{(\mathbf{\Lambda}_{m,h}^k)^{-1}} - \frac{\|\phi(s,a)\|}{(3KM)^k} - \sum_{i=1}^{k-1} \frac{\sqrt{M}}{(3kM)^{k-i}} \|\phi(s,a)\| \right)$$

$$\geq \frac{1}{2\sqrt{\beta_K}} \left( \|\phi(s,a)\|_{(\mathbf{\Lambda}_{m,h}^k)^{-1}} - \frac{1}{3\sqrt{kM}} \|\phi(s,a)\| - \frac{1}{6\sqrt{kM}} \|\phi(s,a)\| \right)$$

$$\geq \frac{1}{4\sqrt{\beta_K}} \|\phi(s,a)\|_{(\mathbf{\Lambda}_{m,h}^k)^{-1}}, \tag{E.18}$$

where we used the fact that $\lambda_{\min} \left( \left( \mathbf{\Lambda}_{m,h}^k \right)^{-1} \right) \geq 1/kM$ and $\|\phi(s,a)\|_{(\mathbf{\Lambda}_{m,h}^k)^{-1}} \geq 1/\sqrt{kM} \|\phi(s,a)\|$. Therefore, according to (E.17) and (E.18), with probablity at least $1 - \delta$, it holds that

$$|Z_k| = \left| \frac{r_h(s,a) + \mathbb{P}_h V_{m,h+1}^k(s,a) - \phi(s,a)^\top \boldsymbol{\mu}_{m,h}^{k,J_k}}{\sqrt{\phi(s,a)^\top \mathbf{\Sigma}_{m,h}^{k,J_k} \phi(s,a)}} \right|$$

$$\leq \frac{5H\sqrt{d}C_\delta + \frac{4}{3}}{\frac{1}{4\sqrt{\beta_K}}},$$

which implies $|Z_k| < 1$ when $\frac{1}{\sqrt{\beta_K}} = 20H\sqrt{d}C_\delta + \frac{16}{3}$.

Till now we have proved that for any fixed $\phi(s,a) \in \mathcal{C}(\varepsilon)$ and for all $(m,h,k) \in \mathcal{M} \times [H] \times [K]$, for any fixed $n \in [N]$, with probablity at least $1 - \delta$, we have

$$\mathbb{P}\Big(\phi(s,a)^\top \mathbf{w}_{m,h}^{k,J_k,n} - r_h(s,a) - \mathbb{P}_h V_{m,h+1}^k(s,a) \geq 0\Big) \geq \frac{1}{2\sqrt{2e\pi}}.$$

By taking union bound over $n \in [N]$, with probablity at least $1 - \delta$, we have

$$\mathbb{P}\Big(\max_{n \in [N]}\big\{\phi(s,a)^\top \mathbf{w}_{m,h}^{k,J_k,n} - r_h(s,a) - \mathbb{P}_h V_{m,h+1}^k(s,a)\big\} \geq 0\Big) \geq 1 - \Big(1 - \frac{1}{2\sqrt{2e\pi}}\Big)^N = 1 - {c_0'}^N,$$

where $c_0' = 1 - \frac{1}{2\sqrt{2e\pi}}$. Therefore, for any fixed $\phi(s,a) \in \mathcal{C}(\varepsilon)$ and for all $(m,h,k) \in \mathcal{M} \times [H] \times [K]$, with probability at least $(1 - \delta)\big(1 - {c_0'}^N\big) > 1 - \delta - {c_0'}^N$, we have

$$\max_{n \in [N]}\big\{\phi(s,a)^\top \mathbf{w}_{m,h}^{k,J_k,n} - r_h(s,a) - \mathbb{P}_h V_{m,h+1}^k(s,a)\big\} \geq 0. \tag{E.19}$$

Next for any $\phi = \phi(s,a)$, we can find $\phi' \in \mathcal{C}(\varepsilon)$ such that $\|\phi - \phi'\|_{(\mathbf{\Lambda}_{m,h}^k)^{-1}} \leq \varepsilon$. We define $\Delta\phi = \phi - \phi'$. Recall from Definition 4.1, we have

$$r_h(s,a) + \mathbb{P}_h V_{m,h+1}^k(s,a) = \phi(s,a)^\top \boldsymbol{\theta}_h + \phi(s,a)^\top \big\langle \boldsymbol{\mu}_h, V_{m,h+1}^k\big\rangle_{\mathcal{S}} \stackrel{\text{def}}{=} \phi(s,a)^\top \mathbf{w}_{m,h}^k,$$

where $\mathbf{w}_{m,h}^k = \boldsymbol{\theta}_h + \big\langle \boldsymbol{\mu}_h, V_{m,h+1}^k\big\rangle_{\mathcal{S}}$. Note that $\max\{\|\boldsymbol{\mu}_h(\mathcal{S})\|, \|\boldsymbol{\theta}_h\|\} \leq \sqrt{d}$ and $V_{m,h+1}^k \leq H - h \leq H$, thus we have

$$\big\|\mathbf{w}_{m,h}^k\big\| \leq \|\boldsymbol{\theta}_h\| + \big\|\big\langle \boldsymbol{\mu}_h, V_{m,h+1}^k\big\rangle_{\mathcal{S}}\big\| \leq \sqrt{d} + H\sqrt{d} \leq 2H\sqrt{d}.$$

Then we define the regression error $\Delta\mathbf{w}_{m,h}^k = \mathbf{w}_{m,h}^k - \mathbf{w}_{m,h}^{k,J_k,n}$. Thus we have

$$\max_{n \in [N]}\big\{\phi(s,a)^\top \mathbf{w}_{m,h}^{k,J_k,n} - r_h(s,a) - \mathbb{P}_h V_{m,h+1}^k(s,a)\big\} = \max_{n \in [N]}\big\{-\phi(s,a)^\top \Delta\mathbf{w}_{m,h}^k\big\}.$$

Then by Cauchy-Schwarz inequality, we have

$$\begin{aligned}
\phi^\top \Delta\mathbf{w}_{m,h}^k &= {\phi'}^\top \Delta\mathbf{w}_{m,h}^k + \Delta\phi^\top \Delta\mathbf{w}_{m,h}^k \\
&\geq {\phi'}^\top \Delta\mathbf{w}_{m,h}^k - \|\Delta\phi\| \cdot \big\|\Delta\mathbf{w}_{m,h}^k\big\| \\
&\geq {\phi'}^\top \Delta\mathbf{w}_{m,h}^k - \sqrt{MK}\varepsilon\big\|\Delta\mathbf{w}_{m,h}^k\big\|.
\end{aligned}$$

By triangle inequality, with probability at least $1 - \delta$, we have

$$\big\|\Delta\mathbf{w}_{m,h}^k\big\| \leq \big\|\mathbf{w}_{m,h}^k\big\| + \big\|\mathbf{w}_{m,h}^{k,J_k,n}\big\| \leq 2H\sqrt{d} + B_{\delta/NMHK}$$

Denote $\alpha_\delta = \sqrt{MK}\big(2H\sqrt{d} + B_{\delta/NMHK}\big)$. Then, for all $(m,h,k) \in \mathcal{M} \times [H] \times [K]$, with probability at least $1 - \delta$, we have

$$\max_{n \in [N]}\big\{\phi^\top \Delta\mathbf{w}_{m,h}^k\big\} \geq \max_{n \in [N]}\big\{{\phi'}^\top \Delta\mathbf{w}_{m,h}^k\big\} - \alpha_\delta\varepsilon.$$

Recall from (E.19), by taking union bound, with probability at least $1 - |\mathcal{C}(\varepsilon)|{c_0'}^N - 2\delta$, for all $(m,h,k) \in \mathcal{M} \times [H] \times [K]$ and for all $(s,a) \in \mathcal{S} \times \mathcal{A}$, we have

$$\max_{n \in [N]}\big\{\phi^\top \Delta\mathbf{w}_{m,h}^k\big\} \geq -\alpha_\delta\varepsilon.$$

Finally, with probability at least $1 - |\mathcal{C}(\varepsilon)|{c_0'}^N - 2\delta$, for all $(m,h,k) \in \mathcal{M} \times [H] \times [K]$ and for all $(s,a) \in \mathcal{S} \times \mathcal{A}$, we have

$$l_{m,h}^k(s,a) \leq \alpha_\delta\varepsilon.$$

This completes the proof. $\qquad\square$

## E.9 Proof of Lemma D.12

*Proof.* For simplicity, we denote $(s^k_{m,h}, a^k_{m,h})$ as $z^k_{m,h}$. Then we consider the following mappings $(\nu_M, \nu_K): [MK] \to [M] \times [K]$,

$$\nu_M(\tau) = \tau(\mathrm{mod}\, M), \qquad \nu_K = \left\lceil \frac{\tau}{M} \right\rceil,$$

where we set $\nu_M(\tau) = M$ if $M|\tau$. Next, for any $\tau \geq 0$, we define

$$\bar{\mathbf{\Lambda}}^\tau_h = \lambda \mathbf{I} + \sum_{u=1}^{\tau M} \phi\left(z^{\nu_K(u)}_{\nu_M(u),h}\right) \phi\left(z^{\nu_K(u)}_{\nu_M(u),h}\right)^\top, \quad \text{for } \tau > 0,$$

$$\bar{\mathbf{\Lambda}}^0_h = \lambda \mathbf{I}, \quad \text{for } \tau = 0.$$

We denote $\sigma = \{\sigma_1, \ldots, \sigma_n\}$ as the synchronization episodes, where $\sigma_i \in [K]$, we also denote $\sigma_0 = 0$. Then we separate the episodes $k = 1, \ldots, K$ into two groups based on the following condition,

$$1 \leq \frac{\det(\bar{\mathbf{\Lambda}}^{\sigma_i}_h)}{\det(\bar{\mathbf{\Lambda}}^{\sigma_{i-1}}_h)} \leq 3. \tag{E.20}$$

Note that the left inequality always holds due to $\bar{\mathbf{\Lambda}}^{\sigma_{i-1}}_h \preccurlyeq \bar{\mathbf{\Lambda}}^{\sigma_i}_h$ and the trivial fact that $\mathbf{A} \preccurlyeq \mathbf{B} \Rightarrow \det(\mathbf{A}) \leq \det(\mathbf{B})$. Then we define that $I_1 = \{k \in \mathbb{N}^+, k \in [\sigma_{i-1}, \sigma_i), \forall i \in [n] | (\text{E.20}) \text{ is true}\}$ and $I_2 = \{k \in \mathbb{N}^+, k \in [\sigma_{i-1}, \sigma_i), \forall i \in [n] | (\text{E.20}) \text{ is false}\}$, then $[K] = I_1 \cup I_2 \cup \{K\}$. For any $k \in [\sigma_{i-1}, \sigma_i)$ and $k \in I_1$, note that $\bar{\mathbf{\Lambda}}^{\sigma_{i-1}}_h \preccurlyeq \mathbf{\Lambda}^k_{m,h} \preccurlyeq \bar{\mathbf{\Lambda}}^k_h \preccurlyeq \bar{\mathbf{\Lambda}}^{\sigma_i}_h$, thus for any $m \in \mathcal{M}$, we have

$$\left\|\phi(z^k_{m,h})\right\|_{(\mathbf{\Lambda}^k_{m,h})^{-1}} \leq \left\|\phi(z^k_{m,h})\right\|_{(\bar{\mathbf{\Lambda}}^k_h)^{-1}} \sqrt{\frac{\det(\bar{\mathbf{\Lambda}}^k_h)}{\det(\mathbf{\Lambda}^k_{m,h})}}$$

$$\leq \left\|\phi(z^k_{m,h})\right\|_{(\bar{\mathbf{\Lambda}}^k_h)^{-1}} \sqrt{\frac{\det(\bar{\mathbf{\Lambda}}^{\sigma_i}_h)}{\det(\bar{\mathbf{\Lambda}}^{\sigma_{i-1}}_h)}}$$

$$\leq 2\left\|\phi(z^k_{m,h})\right\|_{(\bar{\mathbf{\Lambda}}^k_h)^{-1}}, \tag{E.21}$$

where the first inequality follows from Lemma J.12, the second inequality follows from the trivial fact that $\mathbf{A} \preccurlyeq \mathbf{B} \Rightarrow \det(\mathbf{A}) \leq \det(\mathbf{B})$, and the final inequality holds because $k \in I_1$. Then we will bound the summation for $k \in I_1$ and $k \in I_2$ respectively.

$$\sum_{k \in I_1 \cup \{K\}} \sum_{m \in \mathcal{M}} \|\phi(z^k_{m,h})\|_{(\mathbf{\Lambda}^k_{m,h})^{-1}} \leq \sqrt{MK \sum_{m \in \mathcal{M}} \sum_{k \in I_1 \cup \{K\}} \|\phi(z^k_{m,h})\|^2_{(\mathbf{\Lambda}^k_{m,h})^{-1}}}$$

$$\leq 2\sqrt{MK \sum_{m \in \mathcal{M}} \sum_{k \in I_1 \cup \{K\}} \|\phi(z^k_{m,h})\|^2_{(\bar{\mathbf{\Lambda}}^k_h)^{-1}}}$$

$$\leq 2\sqrt{MK \sum_{m \in \mathcal{M}} \sum_{k=1}^{K} \|\phi(z^k_{m,h})\|^2_{(\bar{\mathbf{\Lambda}}^k_h)^{-1}}}$$

$$\leq 2\sqrt{MK \log\left(\frac{\det(\mathbf{\Lambda}^K_h)}{\det(\lambda \mathbf{I})}\right)},$$

where the first inequality follows from Cauchy-Schwarz inequality, the second inequality holds due to (E.21), the final equality follows from Lemma J.1 and $\mathbf{\Lambda}^K_h = \sum_{m \in \mathcal{M}} \sum_{k=1}^{K} \phi(s^k_{m,h}, a^k_{m,h}) \phi(s^k_{m,h}, a^k_{m,h})^\top + \lambda \mathbf{I}$.

For any interval $[\sigma_{i-1}, \sigma_i)$, define $\Delta_i = \sigma_i - \sigma_{i-1} - 1$, we calculate that

$$\sum_{k=\sigma_{i-1}}^{\sigma_i - 1} \left\|\phi(z^k_{m,h})\right\|_{(\mathbf{\Lambda}^k_{m,h})^{-1}} \leq \sqrt{\Delta_i \sum_{k=\sigma_{i-1}}^{\sigma_i - 1} \left\|\phi(z^k_{m,h})\right\|^2_{(\mathbf{\Lambda}^k_{m,h})^{-1}}}$$

$$\leq \sqrt{\Delta_i \log\left(\frac{\det(\mathbf{\Lambda}_{m,h}^{\sigma_i - 1})}{\det(\mathbf{\Lambda}_{m,h}^{\sigma_{i-1}})}\right)}$$

$$\leq \sqrt{\gamma},$$

where the last inequality follows from the synchronization condition (3.3).

Define $R_h = \left\lceil \log\left(\frac{\det(\mathbf{\Lambda}_h^K)}{\det(\lambda \mathbf{I})}\right)\right\rceil$, note that $\sigma_n \leq K$, then we can find that

$$R_h \geq \log\left(\frac{\det(\bar{\mathbf{\Lambda}}_h^{\sigma_n})}{\det(\bar{\mathbf{\Lambda}}_h^{\sigma_0})}\right) = \sum_{i=1}^{n} \log\left(\frac{\det(\bar{\mathbf{\Lambda}}_h^{\sigma_i})}{\det(\bar{\mathbf{\Lambda}}_h^{\sigma_{i-1}})}\right).$$

We can claim that $I_2$ has at most $R_h$ synchronization episodes, otherwise

$$R_h \geq \sum_{i=1}^{n} \log\left(\frac{\det(\bar{\mathbf{\Lambda}}_h^{\sigma_i})}{\det(\bar{\mathbf{\Lambda}}_h^{\sigma_{i-1}})}\right) \geq \sum_{i \in \{i | \sigma_{i-1} \in I_2\}} \log\left(\frac{\det(\bar{\mathbf{\Lambda}}_h^{\sigma_i})}{\det(\bar{\mathbf{\Lambda}}_h^{\sigma_{i-1}})}\right) \geq R_h \log 3,$$

which causes the contradiction. Thus $I_2$ has at most $R_h$ intervals, then we get

$$\sum_{k \in I_2} \sum_{m \in \mathcal{M}} \left\|\phi(z_{m,h}^k)\right\|_{(\mathbf{\Lambda}_{m,h}^k)^{-1}} \leq R_h M \sqrt{\gamma} \leq \left(\log\left(\frac{\det(\mathbf{\Lambda}_h^K)}{\det(\lambda \mathbf{I})}\right) + 1\right) M \sqrt{\gamma}.$$

Finally, we can bound the total summation,

$$\sum_{m \in \mathcal{M}} \sum_{k=1}^{K} \left\|\phi(z_{m,h}^k)\right\|_{(\mathbf{\Lambda}_{m,h}^k)^{-1}} \leq \sum_{m \in \mathcal{M}} \sum_{k \in I_2} \left\|\phi(z_{m,h}^k)\right\|_{(\mathbf{\Lambda}_{m,h}^k)^{-1}} + \sum_{m \in \mathcal{M}} \sum_{k \in I_1 \cup \{K\}} \left\|\phi(z_{m,h}^k)\right\|_{(\mathbf{\Lambda}_{m,h}^k)^{-1}}$$

$$\leq \left(\log\left(\frac{\det(\mathbf{\Lambda}_h^K)}{\det(\lambda \mathbf{I})}\right) + 1\right) M \sqrt{\gamma} + 2\sqrt{MK \log\left(\frac{\det(\mathbf{\Lambda}_h^K)}{\det(\lambda \mathbf{I})}\right)}.$$

This completes the proof. $\qquad\square$

# F    Proof of the Regret Bound for CoopTS-LMC in Misspecified Setting

In this section, we prove the regret bound for CoopTS-LMC in the misspecified setting. The regret analysis, the essential supporting lemmas and their corresponding proofs are almost same as what we have presented in Appendix D and Appendix E. Here we mainly point out the differences of proof between these two settings.

## F.1    Supporting Lemmas

**Definition F.1** (Model prediction error). For any $(m, k, h) \in \mathcal{M} \times [K] \times [H]$, we define the model error associated with the reward $r_{m,h}$,

$$l_{m,h}^k(s, a) = r_{m,h}(s, a) + \mathbb{P}_{m,h} V_{m,h+1}^k(s, a) - Q_{m,h}^k(s, a).$$

**Lemma F.2.** Let $\lambda = 1$ in Algorithm 3. Under Definition 4.8, for any fixed $0 < \delta < 1$, with probability at least $1 - \delta$, for all $(m, k, h) \in \mathcal{M} \times [K] \times [H]$ and for any $(s, a) \in \mathcal{S} \times \mathcal{A}$, we have

$$\left|\phi(s, a)^\top \widehat{\mathbf{w}}_{m,h}^k - r_{m,h}(s, a) - \mathbb{P}_{m,h} V_{m,h+1}^k(s, a)\right|$$
$$\leq \left(5H\sqrt{d}C_\delta + 3H\zeta\sqrt{MKd}\right)\|\phi(s, a)\|_{(\mathbf{\Lambda}_{m,h}^k)^{-1}} + 3H\zeta,$$

where $C_\delta$ is defined in Lemma D.7.

*Proof of Lemma F.2.* Recall from Definition 4.8, we have

$$\left|\mathbb{P}_{m,h} V_{m,h+1}^k(s, a) - \phi(s, a)^\top \langle \boldsymbol{\mu}_h, V_{m,h+1}^k \rangle_\mathcal{S}\right| \leq \left\|\mathbb{P}_{m,h}(\cdot \mid s, a) - \langle \phi(s, a), \boldsymbol{\mu}_h(\cdot) \rangle\right\|_1 \|V_{m,h+1}^k\|_\infty$$
$$\leq 2H\left\|\mathbb{P}_{m,h}(\cdot \mid s, a) - \langle \phi(s, a), \boldsymbol{\mu}_h(\cdot) \rangle\right\|_{\mathrm{TV}}$$

$$\le 2H\zeta,$$

where the first inequality follows from Cauchy-Schwarz inequality, the second inequality follows from the fact that $\|V_{m,h+1}^k\|_\infty \le H$ and $P_2$, $\|P_1 - P_2\|_{\mathrm{TV}} = \frac{1}{2}\sum_{\mathbf{x}\in\boldsymbol{\omega}}|P_1(\mathbf{x}) - P_2(\mathbf{x})| = \frac{1}{2}\|P_1 - P_2\|_1$ for two distributions $P_1$ and $P_2$, note that here we regard distribution as infinite dimensional vector, the third inequality follows from Definition 4.8. Define $\Delta_{m,1} = \mathbb{P}_{m,h}V_{m,h+1}^k(s,a) - \phi(s,a)^\top\langle\boldsymbol{\mu}_h, V_{m,h+1}^k\rangle_{\mathcal{S}}$, thus $|\Delta_{m,1}| \le 2H\zeta$. Then we have

$$\mathbb{P}_{m,h}V_{m,h+1}^k(s,a) \tag{F.1}$$

$$= \phi(s,a)^\top\langle\boldsymbol{\mu}_h, V_{m,h+1}^k\rangle_{\mathcal{S}} + \Delta_{m,1}$$

$$= \phi(s,a)^\top\left(\boldsymbol{\Lambda}_{m,h}^k\right)^{-1}\left(\sum_{(s^l,a^l,s'^l)\in U_{m,h}(k)}\phi(s^l,a^l)\phi(s^l,a^l)^\top + \lambda\mathbf{I}\right)\langle\boldsymbol{\mu}_h, V_{m,h+1}^k\rangle_{\mathcal{S}} + \Delta_{m,1}$$

$$= \phi(s,a)^\top\left(\boldsymbol{\Lambda}_{m,h}^k\right)^{-1}\left(\sum_{(s^l,a^l,s'^l)\in U_{m,h}(k)}\phi(s^l,a^l)\phi(s^l,a^l)^\top\langle\boldsymbol{\mu}_h, V_{m,h+1}^k\rangle_{\mathcal{S}}\right)$$

$$+ \lambda\phi(s,a)^\top\left(\boldsymbol{\Lambda}_{m,h}^k\right)^{-1}\langle\boldsymbol{\mu}_h, V_{m,h+1}^k\rangle_{\mathcal{S}} + \Delta_{m,1}$$

$$= \phi(s,a)^\top\left(\boldsymbol{\Lambda}_{m,h}^k\right)^{-1}\left(\sum_{(s^l,a^l,s'^l)\in U_{m,h}(k)}\phi(s^l,a^l)\left(\mathbb{P}_{m,h}V_{m,h+1}^k\right)(s^l,a^l)\right)$$

$$- \phi(s,a)^\top\left(\boldsymbol{\Lambda}_{m,h}^k\right)^{-1}\left(\sum_{(s^l,a^l,s'^l)\in U_{m,h}(k)}\Delta_{m,1}\phi(s^l,a^l)\right)$$

$$+ \lambda\phi(s,a)^\top\left(\boldsymbol{\Lambda}_{m,h}^k\right)^{-1}\langle\boldsymbol{\mu}_h, V_{m,h+1}^k\rangle_{\mathcal{S}} + \Delta_{m,1}. \tag{F.2}$$

Based on (F.1), we can separate the following error into four parts,

$$\phi(s,a)^\top\widehat{\mathbf{w}}_{m,h}^k - r_{m,h}(s,a) - \mathbb{P}_{m,h}V_{m,h+1}^k(s,a)$$

$$= \phi(s,a)^\top\left(\boldsymbol{\Lambda}_{m,h}^k\right)^{-1}\sum_{(s^l,a^l,s'^l)\in U_{m,h}(k)}\left[r_{m,h}(s^l,a^l) + V_{m,h+1}^k(s'^l)\right]\phi(s^l,a^l) - r_{m,h}(s,a)$$

$$- \phi(s,a)^\top\left(\boldsymbol{\Lambda}_{m,h}^k\right)^{-1}\left(\sum_{(s^l,a^l,s'^l)\in U_{m,h}(k)}\phi(s^l,a^l)\left(\mathbb{P}_{m,h}V_{m,h+1}^k\right)(s^l,a^l)\right)$$

$$+ \Delta_{m,1}\phi(s,a)^\top\left(\boldsymbol{\Lambda}_{m,h}^k\right)^{-1}\left(\sum_{(s^l,a^l,s'^l)\in U_{m,h}(k)}\phi(s^l,a^l)\right)$$

$$- \lambda\phi(s,a)^\top\left(\boldsymbol{\Lambda}_{m,h}^k\right)^{-1}\langle\boldsymbol{\mu}_h, V_{m,h+1}^k\rangle_{\mathcal{S}} - \Delta_{m,1}$$

$$= \underbrace{\phi(s,a)^\top\left(\boldsymbol{\Lambda}_{m,h}^k\right)^{-1}\left(\sum_{(s^l,a^l,s'^l)\in U_{m,h}(k)}\phi(s^l,a^l)\left[\left(V_{m,h+1}^k - \mathbb{P}_{m,h}V_{m,h+1}^k\right)(s^l,a^l)\right]\right)}_{(i)}$$

$$+ \underbrace{\phi(s,a)^\top\left(\boldsymbol{\Lambda}_{m,h}^k\right)^{-1}\left(\sum_{(s^l,a^l,s'^l)\in U_{m,h}(k)}r_{m,h}(s^l,a^l)\phi(s^l,a^l)\right) - r_{m,h}(s,a)}_{(ii)}$$

$$\underbrace{- \lambda\phi(s,a)^\top\left(\boldsymbol{\Lambda}_{m,h}^k\right)^{-1}\langle\boldsymbol{\mu}_h, V_{m,h+1}^k\rangle_{\mathcal{S}}}_{(iii)}$$

$$\underbrace{+ \Delta_{m,1}\phi(s,a)^\top\left(\boldsymbol{\Lambda}_{m,h}^k\right)^{-1}\left(\sum_{(s^l,a^l,s'^l)\in U_{m,h}(k)}\phi(s^l,a^l)\right) - \Delta_{m,1}}_{(iv)}. \tag{F.3}$$

We now provide an upper bound for each of the terms in (F.3).

**Bounding Term (i) in** (F.3)**:** same as (E.11) in Appendix E.6, with probability at least $1 - \delta$, we have

$$|\textbf{Term (i)}| \le 3H\sqrt{d}C_\delta \|\phi(s,a)\|_{(\boldsymbol{\Lambda}_{m,h}^k)^{-1}}. \tag{F.4}$$

**Bounding Term (ii) + Term (iv) in** (F.3)**:** define $\Delta_{m,2} = r_{m,h}(s,a) - \phi(s,a)^\top \boldsymbol{\theta}_h$, then we have $|\Delta_{m,2}| \le \zeta$ due to Definition 4.8. Next we have

$$\phi(s,a)^\top (\boldsymbol{\Lambda}_{m,h}^k)^{-1} \Bigg( \sum_{(s^l,a^l,s'^l) \in U_{m,h}(k)} r_{m,h}(s^l,a^l)\phi(s^l,a^l) \Bigg) - r_{m,h}(s,a)$$

$$= \phi(s,a)^\top (\boldsymbol{\Lambda}_{m,h}^k)^{-1} \Bigg( \sum_{(s^l,a^l,s'^l) \in U_{m,h}(k)} r_{m,h}(s^l,a^l)\phi(s^l,a^l) \Bigg) - \phi(s,a)^\top \boldsymbol{\theta}_h - \Delta_{m,2}$$

$$= \phi(s,a)^\top (\boldsymbol{\Lambda}_{m,h}^k)^{-1} \Bigg( \sum_{(s^l,a^l,s'^l) \in U_{m,h}(k)} r_{m,h}(s^l,a^l)\phi(s^l,a^l) - \boldsymbol{\Lambda}_{m,h}^k \boldsymbol{\theta}_h \Bigg) - \Delta_{m,2}$$

$$= \phi(s,a)^\top (\boldsymbol{\Lambda}_{m,h}^k)^{-1} \Bigg( \sum_{(s^l,a^l,s'^l) \in U_{m,h}(k)} \phi(s^l,a^l)r_{m,h}(s^l,a^l)$$

$$- \sum_{(s^l,a^l,s'^l) \in U_{m,h}(k)} \phi(s^l,a^l)\phi(s^l,a^l)^\top \boldsymbol{\theta}_h - \lambda\mathbf{I}\boldsymbol{\theta}_h \Bigg) - \Delta_{m,2}$$

$$= \phi(s,a)^\top (\boldsymbol{\Lambda}_{m,h}^k)^{-1} \Bigg( \sum_{(s^l,a^l,s'^l) \in U_{m,h}(k)} \phi(s^l,a^l)\Delta_{m,2} - \lambda\mathbf{I}\boldsymbol{\theta}_h \Bigg) - \Delta_{m,2}$$

$$= -\lambda\phi(s,a)^\top (\boldsymbol{\Lambda}_{m,h}^k)^{-1} \boldsymbol{\theta}_h + \underbrace{\Delta_{m,2}\phi(s,a)^\top (\boldsymbol{\Lambda}_{m,h}^k)^{-1} \Bigg( \sum_{(s^l,a^l,s'^l) \in U_{m,h}(k)} \phi(s^l,a^l) \Bigg) - \Delta_{m,2}}_{(v)}, \tag{F.5}$$

where the third equality uses the definition of $\boldsymbol{\Lambda}_{m,h}^k$. By Combining (F.5) and (E.13) in Appendix E.6, we obtain

$$|\textbf{Term (ii)} + \textbf{Term (iv)}| \le \sqrt{\lambda d}\|\phi(s,a)\|_{(\boldsymbol{\Lambda}_{m,h}^k)^{-1}} + |\textbf{Term (iv)} + \textbf{Term (v)}|. \tag{F.6}$$

Then we calculate that

$$|\textbf{Term (iv)} + \textbf{Term (v)}|$$

$$= \Bigg| (\Delta_{m,1} + \Delta_{m,2})\phi(s,a)^\top (\boldsymbol{\Lambda}_{m,h}^k)^{-1} \Bigg( \sum_{(s^l,a^l,s'^l) \in U_{m,h}(k)} \phi(s^l,a^l) \Bigg) - (\Delta_{m,1} + \Delta_{m,2}) \Bigg|$$

$$\le |\Delta_{m,1} + \Delta_{m,2}| \cdot \Bigg| \phi(s,a)^\top (\boldsymbol{\Lambda}_{m,h}^k)^{-1} \Bigg( \sum_{(s^l,a^l,s'^l) \in U_{m,h}(k)} \phi(s^l,a^l) \Bigg) \Bigg| + |\Delta_{m,1} + \Delta_{m,2}|$$

$$\le 3H\zeta\|\phi(s,a)\|_{(\boldsymbol{\Lambda}_{m,h}^k)^{-1}} \sum_{(s^l,a^l,s'^l) \in U_{m,h}(k)} \|\phi(s^l,a^l)\|_{(\boldsymbol{\Lambda}_{m,h}^k)^{-1}} + 3H\zeta$$

$$\le 3H\zeta\|\phi(s,a)\|_{(\boldsymbol{\Lambda}_{m,h}^k)^{-1}} \Bigg( \mathcal{K}(k) \sum_{(s^l,a^l,s'^l) \in U_{m,h}(k)} \|\phi(s^l,a^l)\|_{(\boldsymbol{\Lambda}_{m,h}^k)^{-1}}^2 \Bigg)^{\frac{1}{2}} + 3H\zeta$$

$$\le 3H\zeta\sqrt{MKd}\|\phi(s,a)\|_{(\boldsymbol{\Lambda}_{m,h}^k)^{-1}} + 3H\zeta, \tag{F.7}$$

where the second inequality follows from Cauchy-Schwarz inequality and the fact that $|\Delta_{m,1} + \Delta_{m,2}| \le |\Delta_{m,1}| + |\Delta_{m,2}| \le 2H\zeta + \zeta \le 3H\zeta$, the third inequality holds because of Cauchy-Schwarz inequality, and the last inequality holds because $\mathcal{K}(k) \le MK$ and Lemma J.4. Substitute (F.7) into (F.6), we have

$$|\textbf{Term (ii)} + \textbf{Term (iv)}| \le \big(3H\zeta\sqrt{MKd} + \sqrt{\lambda d}\big)\|\phi(s,a)\|_{(\boldsymbol{\Lambda}_{m,h}^k)^{-1}} + 3H\zeta. \tag{F.8}$$

**Bounding Term (iii) in** (F.3)**:** same as (E.15) in Appendix E.6, we have

$$|\textbf{Term (iii)}| \le H\sqrt{\lambda d}\|\phi(s,a)\|_{(\Lambda_{m,h}^k)^{-1}}. \tag{F.9}$$

**Combine all the terms in** (F.3) **together:** by using triangle inequality in (F.3), we combine (F.4), (F.8) and (F.9), then set $\lambda = 1$, with probability at least $1 - \delta$, we get

$$\left|\phi(s,a)^\top \widehat{\mathbf{w}}_{m,h}^k - r_{m,h}(s,a) - \mathbb{P}_{m,h}V_{m,h+1}^k(s,a)\right|$$
$$\le \left(3H\sqrt{d}C_\delta + \sqrt{d} + H\sqrt{d} + 3H\zeta\sqrt{MKd}\right)\|\phi(s,a)\|_{(\Lambda_{m,h}^k)^{-1}} + 3H\zeta$$
$$\le \left(5H\sqrt{d}C_\delta + 3H\zeta\sqrt{MKd}\right)\|\phi(s,a)\|_{(\Lambda_{m,h}^k)^{-1}} + 3H\zeta.$$

This completes the proof. $\qquad\square$

**Lemma F.3** (Error bound). *Let $\lambda = 1$ in Algorithm 3. Under Definition 4.8, for any fixed $0 < \delta < 1$, with probability at least $1 - \delta - \delta^2$, for any $(m,k,h) \in \mathcal{M} \times [K] \times [H]$ and for any $(s,a) \in \mathcal{S} \times \mathcal{A}$, we have*

$$-l_{m,h}^k(s,a) \le \left(5H\sqrt{d}C_\delta + 3H\zeta\sqrt{MKd} + 5\sqrt{\frac{2d\log\left(\sqrt{N}/\delta\right)}{3\beta_K}} + \frac{4}{3}\right)\|\phi(s,a)\|_{(\Lambda_{m,h}^k)^{-1}} + 3H\zeta,$$

*where $C_\delta$ is defined in Lemma D.7.*

*Proof of Lemma F.3.* We do the same process as that in Appendix E.7, and we have

$$-l_{m,h}^k(s,a) \le \underbrace{\max_{n\in[N]}\left|\phi(s,a)^\top \mathbf{w}_{m,h}^{k,J_k,n} - \phi(s,a)^\top \widehat{\mathbf{w}}_{m,h}^k\right|}_{(i)}$$
$$+ \underbrace{\left|\phi(s,a)^\top \widehat{\mathbf{w}}_{m,h}^k - r_{m,h}(s,a) - \mathbb{P}_{m,h}V_{m,h+1}^k(s,a)\right|}_{(ii)}.$$

**Bounding Term (i):** based on (E.16), for any $(m,h,k) \in \mathcal{M} \times [H] \times [K]$ and $(s,a) \in \mathcal{S} \times \mathcal{A}$, with probability at least $1 - \delta^2$, we have

$$\max_{n\in[N]}\left|\phi(s,a)^\top \mathbf{w}_{m,h}^{k,J_k,n} - \phi(s,a)^\top \widehat{\mathbf{w}}_{m,h}^k\right| \le \left(5\sqrt{\frac{2d\log\left(\sqrt{N}/\delta\right)}{3\beta_K}} + \frac{4}{3}\right)\|\phi(s,a)\|_{(\Lambda_{m,h}^k)^{-1}}.$$

**Bounding Term (ii):** based on Lemma F.2, for all $(m,h,k) \in \mathcal{M} \times [H] \times [K]$ and $(s,a) \in \mathcal{S} \times \mathcal{A}$, we have

$$\left|\phi(s,a)^\top \widehat{\mathbf{w}}_{m,h}^k - r_h^k(s,a) - \mathbb{P}_h V_{m,h+1}^k(s,a)\right| \le \left(5H\sqrt{d}C_\delta + 3H\zeta\sqrt{MKd}\right)\|\phi(s,a)\|_{(\Lambda_{m,h}^k)^{-1}} + 3H\zeta.$$

Combine the two result above, by taking union bound, with probability at least $1 - \delta - \delta^2$, we have

$$-l_{m,h}^k(s,a) \le \left(5H\sqrt{d}C_\delta + 3H\zeta\sqrt{MKd} + 5\sqrt{\frac{2d\log\left(\sqrt{N}/\delta\right)}{3\beta_K}} + \frac{4}{3}\right)\|\phi(s,a)\|_{(\Lambda_{m,h}^k)^{-1}} + 3H\zeta.$$

This completes the proof. $\qquad\square$

**Lemma F.4** (Optimism). *Let $\lambda = 1$ in Algorithm 3 and $c_0' = 1 - \frac{1}{2\sqrt{2e\pi}}$. Under Definition 4.8, for any fixed $0 < \delta < 1$, with probability at least $1 - |\mathcal{C}(\varepsilon)|c_0'^N - 2\delta$ where $|\mathcal{C}(\varepsilon)| \le (3/\varepsilon)^d$, for all $(m,h,k) \in \mathcal{M} \times [H] \times [K]$ and for all $(s,a) \in \mathcal{S} \times \mathcal{A}$, we have*

$$l_{m,h}^k(s,a) \le \alpha_\delta \varepsilon + 3H\zeta,$$

*where $\alpha_\delta = \sqrt{MK}\left(2H\sqrt{d} + B_{\delta/NMHK}\right)$.*

*Proof of Lemma F.4.* This proof is similar to the proof in Appendix E.8, we just prove the part that for fixed $\phi \in \mathcal{C}(\varepsilon)$. Recall from Definition F.1,

$$l_{m,h}^k(s,a) = r_{m,h}(s,a) + \mathbb{P}_{m,h}V_{m,h+1}^k(s,a) - Q_{m,h}^k(s,a).$$

Note that

$$Q_{m,h}^k(s,a) = \min\left\{\max_{n\in[N]}\phi(s,a)^\top \mathbf{w}_{m,h}^{k,J_k,n}, H-h+1\right\}^+ \leq \max_{n\in[N]}\phi(x,a)^\top \mathbf{w}_{m,h}^{k,J_k,n}.$$

Here we define

$$Z_k = \frac{r_{m,h}(s,a) + \mathbb{P}_{m,h}V_{m,h+1}^k(s,a) - \phi(s,a)^\top \boldsymbol{\mu}_{m,h}^{k,J_k} - (\Delta_{m,1} + \Delta_{m,2})}{\sqrt{\phi(s,a)^\top \boldsymbol{\Sigma}_{m,h}^{k,J_k}\phi(s,a)}},$$

where $\Delta_{m,1} = \mathbb{P}_{m,h}V_{m,h+1}^k(s,a) - \phi(s,a)^\top \langle \boldsymbol{\mu}_h, V_{m,h+1}^k \rangle_{\mathcal{S}}$, $\Delta_{m,2} = r_{m,h}(s,a) - \phi(s,a)^\top \boldsymbol{\theta}_h$. Based on the results in Proposition D.3, we have that $\phi(s,a)^\top \mathbf{w}_{m,h}^{k,J_k,n} \sim \mathcal{N}\left(\phi(s,a)^\top \boldsymbol{\mu}_{m,h}^{k,J_k}, \phi(s,a)^\top \boldsymbol{\Sigma}_{m,h}^{k,J_k}\phi(s,a)\right)$, for any fixed $n \in [N]$. When $|Z_k| < 1$, by Gaussian concentration Lemma J.10, we have

$$\mathbb{P}\left(r_{m,h}(s,a) + \mathbb{P}_{m,h}V_{m,h+1}^k(s,a) - \phi(s,a)^\top \mathbf{w}_{m,h}^{k,J_k,n} \leq (\Delta_{m,1} + \Delta_{m,2})\right)$$

$$= \mathbb{P}\left(\phi(s,a)^\top \mathbf{w}_{m,h}^{k,J_k,n} \geq r_{m,h}(s,a) + \mathbb{P}_{m,h}V_{m,h+1}^k(s,a) - (\Delta_{m,1} + \Delta_{m,2})\right)$$

$$= \mathbb{P}\left(\frac{\phi(s,a)^\top \mathbf{w}_{m,h}^{k,J_k,n} - \phi(s,a)^\top \boldsymbol{\mu}_{m,h}^{k,J_k}}{\sqrt{\phi(s,a)^\top \boldsymbol{\Sigma}_{m,h}^{k,J_k}\phi(s,a)}} \geq \frac{r_{m,h}(s,a) + \mathbb{P}_{m,h}V_{m,h+1}^k(s,a) - (\Delta_{m,1} + \Delta_{m,2}) - \phi(s,a)^\top \boldsymbol{\mu}_{m,h}^{k,J_k}}{\sqrt{\phi(s,a)^\top \boldsymbol{\Sigma}_{m,h}^{k,J_k}\phi(s,a)}}\right)$$

$$= \mathbb{P}\left(\frac{\phi(s,a)^\top \mathbf{w}_{m,h}^{k,J_k,n} - \phi(s,a)^\top \boldsymbol{\mu}_{m,h}^{k,J_k}}{\sqrt{\phi(s,a)^\top \boldsymbol{\Sigma}_{m,h}^{k,J_k}\phi(s,a)}} \geq Z_k\right)$$

$$\geq \frac{1}{2\sqrt{2\pi}}\exp(-Z_k^2/2)$$

$$\geq \frac{1}{2\sqrt{2e\pi}}.$$

**Consider the numerator of $Z_k$:**

$$\left| r_{m,h}(s,a) + \mathbb{P}_{m,h}V_{m,h+1}^k(s,a) - \phi(s,a)^\top \boldsymbol{\mu}_{m,h}^{k,J_k} - (\Delta_{m,1} + \Delta_{m,2})\right|$$

$$\leq \underbrace{\left| r_{m,h}(s,a) + \mathbb{P}_{m,h}V_{m,h+1}^k(s,a) - \phi(s,a)^\top \widehat{\mathbf{w}}_{m,h}^k - (\Delta_{m,1} + \Delta_{m,2})\right|}_{I_1}$$

$$+ \underbrace{\left|\phi(s,a)^\top \widehat{\mathbf{w}}_{m,h}^k - \phi(s,a)^\top \boldsymbol{\mu}_{m,h}^{k,J_k}\right|}_{I_2}. \tag{F.10}$$

**Bounding Term $I_1$ in (F.10):** recall the proof of Lemma F.2, we do the almost same error decomposition as (F.3) with the only difference of adding term $(\Delta_{m,1} + \Delta_{m,2})$

$$\phi(s,a)^\top \widehat{\mathbf{w}}_{m,h}^k - r_{m,h}(s,a) - \mathbb{P}_{m,h}V_{m,h+1}^k(s,a) + (\Delta_{m,1} + \Delta_{m,2})$$

$$= \underbrace{\phi(s,a)^\top \left(\boldsymbol{\Lambda}_{m,h}^k\right)^{-1}\left(\sum_{(s^l,a^l,s'^l)\in U_{m,h}(k)}\phi(s^l,a^l)\left[\left(V_{m,h+1}^k - \mathbb{P}_{m,h}V_{m,h+1}^k\right)(s^l,a^l)\right]\right)}_{(i)}$$

$$+ \underbrace{\phi(s,a)^\top \left(\boldsymbol{\Lambda}_{m,h}^k\right)^{-1}\left(\sum_{(s^l,a^l,s'^l)\in U_{m,h}(k)}r_{m,h}(s^l,a^l)\phi(s^l,a^l)\right) - r_{m,h}(s,a)}_{(ii)}$$

$$- \underbrace{\lambda\phi(s,a)^\top \left(\boldsymbol{\Lambda}_{m,h}^k\right)^{-1}\langle \boldsymbol{\mu}_h, V_{m,h+1}^k \rangle_{\mathcal{S}}}_{(iii)}$$

$$+ \Delta_{m,1} \phi(s,a)^\top \left(\mathbf{\Lambda}_{m,h}^k\right)^{-1} \underbrace{\left( \sum_{(s^l,a^l,s'^l)\in U_{m,h}(k)} \phi(s^l,a^l) \right)}_{} + \Delta_{m,2} \, . \tag{F.11}$$
$$\underbrace{\phantom{+ \Delta_{m,1} \phi(s,a)^\top \left(\mathbf{\Lambda}_{m,h}^k\right)^{-1} \left( \sum_{(s^l,a^l,s'^l)\in U_{m,h}(k)} \phi(s^l,a^l) \right)}}_{\text{(iv)}}$$

We now provide an upper bound for each of the terms in (F.11).

**Bounding Term (i) in** (F.11): almost same as (E.11) in Appendix E.6 with the only difference between $\mathbb{P}_h$ and $\mathbb{P}_{m,h}$, with probability at least $1-\delta$, we have

$$|\textbf{Term (i)}| \leq 3H\sqrt{d}C_\delta \|\phi(s,a)\|_{(\mathbf{\Lambda}_{m,h}^k)^{-1}} . \tag{F.12}$$

**Bounding Term (ii) + Term (iv) in** (F.11): we do the same calculation as that in the proof of Lemma F.2, based on (F.5), we have

$$\textbf{Term (ii)} = \phi(s,a)^\top \left(\mathbf{\Lambda}_{m,h}^k\right)^{-1} \left( \sum_{(s^l,a^l,s'^l)\in U_{m,h}(k)} r_{m,h}(s^l,a^l)\phi(s^l,a^l) \right) - r_{m,h}(s,a)$$

$$= -\lambda \phi(s,a)^\top \left(\mathbf{\Lambda}_{m,h}^k\right)^{-1} \boldsymbol{\theta}_h + \Delta_{m,2} \phi(s,a)^\top \left(\mathbf{\Lambda}_{m,h}^k\right)^{-1} \underbrace{\left( \sum_{(s^l,a^l,s'^l)\in U_{m,h}(k)} \phi(s^l,a^l) \right) - \Delta_{m,2}}_{\text{(v)}} \, .$$
$$\tag{F.13}$$

By Combining (F.13) and (E.13) in Appendix E.6, we obtain

$$|\textbf{Term (ii)} + \textbf{Term (iv)}| \leq \sqrt{\lambda d}\|\phi(s,a)\|_{(\mathbf{\Lambda}_{m,h}^k)^{-1}} + |\textbf{Term (iv)} + \textbf{Term (v)}| . \tag{F.14}$$

Then we calculate that

$$|\textbf{Term (iv)} + \textbf{Term (v)}| = |\Delta_{m,1} + \Delta_{m,2}| \cdot \left| \phi(s,a)^\top \left(\mathbf{\Lambda}_{m,h}^k\right)^{-1} \left( \sum_{(s^l,a^l,s'^l)\in U_{m,h}(k)} \phi(s^l,a^l) \right) \right|$$

$$\leq 3H\zeta \|\phi(s,a)\|_{(\mathbf{\Lambda}_{m,h}^k)^{-1}} \sum_{(s^l,a^l,s'^l)\in U_{m,h}(k)} \left\| \phi(s^l,a^l) \right\|_{(\mathbf{\Lambda}_{m,h}^k)^{-1}}$$

$$\leq 3H\zeta \|\phi(s,a)\|_{(\mathbf{\Lambda}_{m,h}^k)^{-1}} \left( \mathcal{K}(k) \sum_{(s^l,a^l,s'^l)\in U_{m,h}(k)} \left\| \phi(s^l,a^l) \right\|_{(\mathbf{\Lambda}_{m,h}^k)^{-1}}^2 \right)^{\frac{1}{2}}$$

$$\leq 3H\zeta \sqrt{MKd} \|\phi(s,a)\|_{(\mathbf{\Lambda}_{m,h}^k)^{-1}} , \tag{F.15}$$

where the first inequality follows from Cauchy-Schwarz inequality and the fact that $|\Delta_{m,1} + \Delta_{m,2}| \leq 3H\zeta$, the second inequality holds because of Cauchy-Schwarz inequality, and the last inequality holds because $\mathcal{K}(k) \leq MK$ and Lemma J.4. Substitute (F.15) into (F.14), we have

$$|\textbf{Term (ii)} + \textbf{Term (iv)}| \leq \left(3H\zeta\sqrt{MKd} + \sqrt{\lambda d}\right)\|\phi(s,a)\|_{(\mathbf{\Lambda}_{m,h}^k)^{-1}} . \tag{F.16}$$

**Bounding Term (iii) in** (F.11): same as (E.15) in Appendix E.6, we have

$$|\textbf{Term (iii)}| \leq H\sqrt{\lambda d}\|\phi(s,a)\|_{(\mathbf{\Lambda}_{m,h}^k)^{-1}} . \tag{F.17}$$

**Combine all the terms in** (F.11) **together**: by using triangle inequality in (F.11), we combine (F.12), (F.16) and (F.17), then set $\lambda = 1$, with probability at least $1-\delta$, we get

$$\left| \phi(s,a)^\top \widehat{\mathbf{w}}_{m,h}^k - r_{m,h}(s,a) - \mathbb{P}_{m,h} V_{m,h+1}^k(s,a) + (\Delta_{m,1} + \Delta_{m,2}) \right|$$
$$\leq \left(5H\sqrt{d}C_\delta + 3H\zeta\sqrt{MKd}\right)\|\phi(s,a)\|_{(\mathbf{\Lambda}_{m,h}^k)^{-1}} .$$

**Bounding Term $I_2$ in** (F.10): same as the proof in Appendix E.8, we have

$$\left| \phi(s,a)^\top \widehat{\mathbf{w}}_{m,h}^k - \phi(s,a)^\top \boldsymbol{\mu}_{m,h}^{k,J_k} \right| \leq \frac{4}{3} \|\phi(s,a)\|_{(\mathbf{\Lambda}_{m,h}^k)^{-1}} .$$

So, with probability at least $1 - \delta$, we have

$$\left| r_{m,h}(s,a) + \mathbb{P}_{m,h} V_{m,h+1}^k(s,a) - \phi(s,a)^\top \boldsymbol{\mu}_{m,h}^{k,J_k} \right| \leq \left( 5H\sqrt{d}C_\delta + 3H\zeta\sqrt{MKd} + \frac{4}{3} \right) \|\phi(s,a)\|_{(\boldsymbol{\Lambda}_{m,h}^k)^{-1}}.$$
(F.18)

**Consider the denominator of $Z_k$:** same as the proof in Appendix E.8, with (E.18), we have

$$\|\phi(s,a)\|_{\boldsymbol{\Sigma}_{m,h}^{k,J_k}} \geq \frac{1}{4\sqrt{\beta_K}} \|\phi(s,a)\|_{(\boldsymbol{\Lambda}_{m,h}^k)^{-1}},$$
(F.19)

where we used the fact that $\lambda_{\min}\left(\left(\boldsymbol{\Lambda}_{m,h}^k\right)^{-1}\right) \geq 1/k$ and $\|\phi(s,a)\|_{(\boldsymbol{\Lambda}_{m,h}^k)^{-1}} \geq 1/\sqrt{k}\|\phi(s,a)\|_2$. Therefore, according to (F.18) and (F.19), with probability at least $1 - \delta$, it holds that

$$
\begin{aligned}
|Z_k| &= \left| \frac{r_{m,h}(s,a) + \mathbb{P}_{m,h}V_{m,h+1}^k(s,a) - \phi(s,a)^\top \boldsymbol{\mu}_{m,h}^{k,J_k}}{\sqrt{\phi(s,a)^\top \boldsymbol{\Sigma}_{m,h}^{k,J_k} \phi(s,a)}} \right| \\
&\leq \frac{\left( 5H\sqrt{d}C_\delta + 3H\zeta\sqrt{MKd} + \frac{4}{3} \right) \|\phi(s,a)\|_{(\boldsymbol{\Lambda}_{m,h}^k)^{-1}}}{\frac{1}{4\sqrt{\beta_K}} \|\phi(s,a)\|_{(\boldsymbol{\Lambda}_{m,h}^k)^{-1}}} \\
&= \frac{5H\sqrt{d}C_\delta + 3H\zeta\sqrt{MKd} + \frac{4}{3}}{\frac{1}{4\sqrt{\beta_K}}},
\end{aligned}
$$

which implies $|Z_k| < 1$ when $\frac{1}{\sqrt{\beta_K}} = 20H\sqrt{d}C_\delta + 12H\zeta\sqrt{MKd} + \frac{16}{3}$.

Now we have already proved that, for any fixed $n \in [N]$, with probability at least $1 - \delta$, we have

$$\mathbb{P}\left( r_{m,h}(s,a) + \mathbb{P}_{m,h}V_{m,h+1}^k(s,a) - \phi(s,a)^\top \mathbf{w}_{m,h}^{k,J_k,n} \leq (\Delta_{m,1} + \Delta_{m,2}) \right) \geq \frac{1}{2\sqrt{2e\pi}}.$$

By taking union bound over $n \in [N]$, with probablity at least $1 - \delta$, we have

$$
\begin{aligned}
&\mathbb{P}\left( \max_{n \in [N]} \left\{ \phi(s,a)^\top \mathbf{w}_{m,h}^{k,J_k,n} - r_{m,h}(s,a) - \mathbb{P}_{m,h}V_{m,h+1}^k(s,a) \right\} \geq -(\Delta_{m,1} + \Delta_{m,2}) \right) \\
&\geq 1 - \left( 1 - \frac{1}{2\sqrt{2e\pi}} \right)^N \\
&= 1 - c_0'^N,
\end{aligned}
$$

where $c_0' = 1 - \frac{1}{2\sqrt{2e\pi}}$. Finally, with probability at least $(1 - \delta)\left(1 - c_0'^N\right)$, for all $(s,a) \in \mathcal{S} \times \mathcal{A}$, we have

$$l_{m,h}^k(s,a) \leq 3H\zeta.$$

Till now we have completed the proof of fixed $\phi \in \mathcal{C}(\varepsilon)$. Follow the proof in Appendix E.8, we can get the final result. $\square$

## F.2 Regret Analysis

In this part, we give out the proof of Theorem 4.12, the regret bound for CoopTS-LMC in the misspecified setting.

*Proof of Theorem 4.12.* This proof is almost same as the proof in Appendix D.2. We do the same regret decomposition (D.2) and obtain the same bound for **Term(i)** (D.3) and **Term(ii)** (D.4). Next we bound **Term (iii)** with new lemmas in the misspecified setting.

**Bounding Term (iii) in** (D.2): based on Lemma F.3 and Lemma F.4, by taking union bound, with probability at least $1 - |\mathcal{C}(\varepsilon)|c_0'^N - 2\delta' - MHK(\delta' + \delta'^2)$, we have

$$\sum_{m \in \mathcal{M}} \sum_{k=1}^K \sum_{h=1}^H \left( \mathbb{E}_{\pi^*}\left[ l_{m,h}^k(s_{m,h}, a_{m,h}) | s_{m,1} = s_{m,1}^k \right] - l_{m,h}^k\left( s_{m,h}^k, a_{m,h}^k \right) \right)$$

$$\leq \sum_{m\in\mathcal{M}}\sum_{k=1}^{K}\sum_{h=1}^{H}\Big(-l_{m,h}^{k}\big(s_{m,h}^{k},a_{m,h}^{k}\big)+\alpha_{\delta'}\varepsilon+3H\zeta\Big)$$

$$\leq \sum_{m\in\mathcal{M}}\sum_{k=1}^{K}\sum_{h=1}^{H}\Bigg(\bigg(5H\sqrt{d}C_{\delta'}+3H\zeta\sqrt{MKd}+5\sqrt{\frac{2d\log\big(\sqrt{N}/\delta'\big)}{3\beta_{K}}}+\frac{4}{3}\bigg)\big\|\phi(s_{m,h}^{k},a_{m,h}^{k})\big\|_{(\mathbf{\Lambda}_{m,h}^{k})^{-1}}$$

$$+\alpha_{\delta'}\varepsilon+6H\zeta\Bigg)$$

$$= HMK\alpha_{\delta'}\varepsilon+6H^{2}MK\zeta+\bigg(5H\sqrt{d}C_{\delta'}+3H\zeta\sqrt{MKd}+5\sqrt{\frac{2d\log\big(\sqrt{N}/\delta'\big)}{3\beta_{K}}}+\frac{4}{3}\bigg)$$

$$\times\sum_{h=1}^{H}\sum_{m\in\mathcal{M}}\sum_{k=1}^{K}\big\|\phi(s_{m,h}^{k},a_{m,h}^{k})\big\|_{(\mathbf{\Lambda}_{m,h}^{k})^{-1}}$$

$$\leq HMK\alpha_{\delta'}\varepsilon+6H^{2}MK\zeta+\bigg(5H\sqrt{d}C_{\delta'}+3H\zeta\sqrt{MKd}+5\sqrt{\frac{2d\log\big(\sqrt{N}/\delta'\big)}{3\beta_{K}}}+\frac{4}{3}\bigg)$$

$$\times\sum_{h=1}^{H}\bigg(\log\bigg(\frac{\det(\mathbf{\Lambda}_{h}^{K})}{\det(\lambda\mathbf{I})}\bigg)+1\bigg)M\sqrt{\gamma}+2\sqrt{MK\log\bigg(\frac{\det(\mathbf{\Lambda}_{h}^{K})}{\det(\lambda\mathbf{I})}\bigg)}$$

$$\leq HMK\alpha_{\delta'}\varepsilon+6H^{2}MK\zeta+\bigg(5H\sqrt{d}C_{\delta'}+3H\zeta\sqrt{MKd}+5\sqrt{\frac{2d\log\big(\sqrt{N}/\delta'\big)}{3\beta_{K}}}+\frac{4}{3}\bigg)$$

$$\times H\Big(d(\log(1+MK/d)+1)M\sqrt{\gamma}+2\sqrt{MKd\log(1+MK/d)}\Big)$$

$$= \widetilde{\mathcal{O}}\Big(d^{\frac{3}{2}}H^{2}\sqrt{M}\big(\sqrt{dM\gamma}+\sqrt{K}\big)+d^{\frac{3}{2}}H^{2}M\sqrt{K}\big(\sqrt{dM\gamma}+\sqrt{K}\big)\zeta\Big). \tag{F.20}$$

The first inequality follows from Lemma F.4, the second inequality follows from Lemma F.3, the third inequality follows from Lemma D.12, the last inequality holds due to Lemma J.2 and the fact that $\|\phi(\cdot)\|_{2}\leq 1$, the last equality follows from $\frac{1}{\sqrt{\beta_{K}}}=20H\sqrt{d}C_{\delta'}+12H\zeta\sqrt{MKd}+\frac{16}{3}$, which we define in Lemma F.4.

The probability calculation is same as that in Appendix D.2. By combining **Terms (i)(ii)(iii)** together, we get that the final regret bound for CoopTS-LMC in misspecified setting is

$$\widetilde{\mathcal{O}}\Big(d^{\frac{3}{2}}H^{2}\sqrt{M}\big(\sqrt{dM\gamma}+\sqrt{K}\big)+d^{\frac{3}{2}}H^{2}M\sqrt{K}\big(\sqrt{dM\gamma}+\sqrt{K}\big)\zeta\Big),$$

with probability at least $1-\delta$. Here we finish the proof. $\qquad\square$

# G  Proof of the Regret Bound for CoopTS-PHE

Before getting the regret bound for CoopTS-PHE, we first present some essential technical lemmas required for our analysis.

## G.1  Supporting Lemmas

**Proposition G.1.** The difference between the perturbed estimated parameter $\widetilde{\mathbf{w}}_{m,h}^{k,n}$ and unperturbed estimated parameter $\widehat{\mathbf{w}}_{m,h}^{k}$ satisfies the Gaussian distribution,

$$\zeta_{m,h}^{k,n}=\widetilde{\mathbf{w}}_{m,h}^{k,n}-\widehat{\mathbf{w}}_{m,h}^{k}\sim\mathcal{N}\Big(\mathbf{0},\sigma^{2}\big(\mathbf{\Lambda}_{m,h}^{k}\big)^{-1}\Big),$$

where $\widehat{\mathbf{w}}_{m,h}^{k}=\big(\mathbf{\Lambda}_{m,h}^{k}\big)^{-1}\Big(\sum_{(s^{l},a^{l},s'^{l})\in U_{m,h}(k)}\big[r_{h}+V_{m,h+1}^{k}\big(s'^{l}\big)\big]\phi\big(s^{l},a^{l}\big)\Big)$ is the unperturbed estimated parameter.

Next we will define some good events that hold with high probability to help prove the critical lemmas in this section.

**Lemma G.2** (Good events). For any fixed $0 < \delta < 1$, with some constant $c > 0$, we define the following random events

$$\mathcal{G}_{m,h}^k(\boldsymbol{\zeta}, \delta) \stackrel{\text{def}}{=} \Big\{ \max_{n \in [N]} \big\|\boldsymbol{\zeta}_{m,h}^{k,n}\big\|_{\boldsymbol{\Lambda}_{m,h}^k} \leq c_1 \sigma \sqrt{d} \Big\},$$

$$\mathcal{G}(M, K, H, \delta) \stackrel{\text{def}}{=} \bigcap_{m \in \mathcal{M}} \bigcap_{k \leq K} \bigcap_{h \leq H} \mathcal{G}_{m,h}^k(\boldsymbol{\zeta}, \delta),$$

where $c_1 = c\sqrt{\log(dNMKH/\delta)}$. Then the event $\mathcal{G}(M, K, H, \delta)$ occurs with probability at least $1 - \delta$.

**Lemma G.3.** Let $\lambda = 1$ in Algorithm 2. For any fixed $0 < \delta < 1$, conditioned on the event $\mathcal{G}(M, K, H, \delta)$, with probability $1 - \delta$, for all $(m, k, h) \in \mathcal{M} \times [K] \times [H]$, we have

$$\left\| \sum_{(s^l, a^l, s'^l) \in U_{m,h}(k)} \phi(s^l, a^l) \big[\big(V_{m,h+1}^k - \mathbb{P}_h V_{m,h+1}^k\big)(s^l, a^l)\big] \right\|_{(\boldsymbol{\Lambda}_{m,h}^k)^{-1}} \leq 3H\sqrt{d}D_\delta,$$

where we define $D_\delta = \Big[\frac{1}{2}\log(K+1) + \log\Big(\frac{6\sqrt{2}K(2H\sqrt{MKd}+c_1\sigma\sqrt{d})}{H}\Big) + \log\frac{1}{\delta}\Big]^{1/2}$.

**Lemma G.4.** Let $\lambda = 1$ in Algorithm 2. Under Definition 4.1, for any fixed $0 < \delta < 1$, conditioned on the event $\mathcal{G}(M, K, H, \delta)$, with probability $1 - \delta$, for all $(m, k, h) \in \mathcal{M} \times [K] \times [H]$ and for any $(s, a) \in \mathcal{S} \times \mathcal{A}$, we have

$$\left| \phi(s,a)^\top \widehat{\mathbf{w}}_{m,h}^k - r_h(s,a) - \mathbb{P}_h V_{m,h+1}^k(s,a) \right| \leq 5H\sqrt{d}D_\delta \|\phi(s,a)\|_{(\boldsymbol{\Lambda}_{m,h}^k)^{-1}}. \tag{G.1}$$

**Lemma G.5** (Optimism). Let $\lambda = 1$ in Algorithm 2 and set $c_0 = \Phi(1)$. Under Definition 4.1, conditioned on the event $\mathcal{G}(M, K, H, \delta)$, with probability at least $1 - |\mathcal{C}(\varepsilon)|c_0^N - \delta$ where $|\mathcal{C}(\varepsilon)| \leq (3/\varepsilon)^d$, for all $(m, k, h) \in \mathcal{M} \times [K] \times [H]$ and for all $(s, a) \in \mathcal{S} \times \mathcal{A}$, we have

$$l_{m,h}^k(s,a) \leq A_\delta \varepsilon,$$

where $A_\delta = c_1 \sigma \sqrt{d} + 5H\sqrt{d}D_\delta = \widetilde{\mathcal{O}}(Hd)$.

**Lemma G.6** (Error bound). Let $\lambda = 1$ in Algorithm 2. Under Definition 4.1, for any fixed $0 < \delta < 1$, conditioned on the event $\mathcal{G}(M, K, H, \delta)$, with probability $1 - \delta$, for all $(m, k, h) \in \mathcal{M} \times [K] \times [H]$ and for any $(s, a) \in \mathcal{S} \times \mathcal{A}$, we have

$$-l_{m,h}^k(s,a) \leq c_2 Hd\|\phi(s,a)\|_{(\boldsymbol{\Lambda}_{m,h}^k)^{-1}},$$

where $c_2 = \widetilde{\mathcal{O}}(1)$.

## G.2 Regret Analysis

In this part, we give out the proof of Theorem 4.2, the regret bound for CoopTS-PHE.

*Proof of Theorem 4.2.* Based on the result from Lemma D.13, we do the regret decomposition first

$$\text{Regret}(K) = \sum_{m \in \mathcal{M}} \sum_{k=1}^K V_{m,1}^*\big(s_{m,1}^k\big) - V_{m,1}^{\pi_m^k}\big(s_{m,1}^k\big)$$

$$= \underbrace{\sum_{m \in \mathcal{M}} \sum_{k=1}^K \sum_{h=1}^H \mathbb{E}_{\pi^*}\big[\big\langle Q_{m,h}^k(s_{m,h}, \cdot), \pi_{m,h}^*(\cdot, |s_{m,h}) - \pi_{m,h}^k(\cdot|s_{m,h})\big\rangle|s_{m,1} = s_{m,1}^k\big]}_{(i)}$$

$$+ \underbrace{\sum_{m \in \mathcal{M}} \sum_{k=1}^K \sum_{h=1}^H (D_{m,k,h,1} + D_{m,k,h,2})}_{(ii)}$$

$$+ \sum_{m \in \mathcal{M}} \sum_{k=1}^{K} \sum_{h=1}^{H} \underbrace{\left( \mathbb{E}_{\pi^*} \left[ l_{m,h}^k(s_{m,h}, a_{m,h}) | s_{m,1} = s_{m,1}^k \right] - l_{m,h}^k \left( s_{m,h}^k, a_{m,h}^k \right) \right)}_{(iii)}.$$

(G.2)

Next, we will bound the above three terms respectively.

**Bounding Term (i) in** (G.2)**:** for the policy $\pi_{m,h}^k$, we have

$$\sum_{m \in \mathcal{M}} \sum_{k=1}^{K} \sum_{h=1}^{H} \mathbb{E}_{\pi^*} \left[ \langle Q_{m,h}^k(s_{m,h}, \cdot), \pi_{m,h}^*(\cdot | s_{m,h}) - \pi_{m,h}^k(\cdot | s_{m,h}) \rangle | s_{m,1} = s_{m,1}^k \right] \le 0. \quad \text{(G.3)}$$

This is because by definition $\pi_{m,h}^k$ is the greedy policy for $Q_{m,h}^k$.

**Bounding Term (ii) in** (G.2)**:** note that $0 \le Q_{m,h}^k \le H - h + 1 \le H$, based on (D.1), for any $(m,k,h) \in \mathcal{M} \times [K] \times [H]$, we have $|D_{m,k,h,1}| \le 2H$ and $|D_{m,k,h,2}| \le 2H$. Note that $D_{m,k,h,1}$ is a martingale difference sequence $\mathbb{E}[D_{m,k,h,1} | \mathcal{F}_{m,k,h}] = 0$. By applying Azuma-Hoeffding inequality, with probability at least $1 - \delta/3$, we have

$$\sum_{m \in \mathcal{M}} \sum_{k=1}^{K} \sum_{h=1}^{H} D_{m,k,h,1} \le 2\sqrt{2MH^3K \log(6/\delta)}.$$

Note that $D_{m,k,h,2}$ is also a martingale difference sequence. By applying Azuma-Hoeffding inequality, with probability at least $1 - \delta/3$, we have

$$\sum_{m \in \mathcal{M}} \sum_{k=1}^{K} \sum_{h=1}^{H} D_{m,k,h,2} \le 2\sqrt{2MH^3K \log(6/\delta)}.$$

By taking union bound, with probability at least $1 - 2\delta/3$, we have

$$\sum_{m \in \mathcal{M}} \sum_{k=1}^{K} \sum_{h=1}^{H} D_{m,k,h,1} + \sum_{m \in \mathcal{M}} \sum_{k=1}^{K} \sum_{h=1}^{H} D_{m,k,h,2} \le 4\sqrt{2MH^3K \log(6/\delta)}. \quad \text{(G.4)}$$

**Bounding Term (iii) in** (G.2)**:** conditioned on the event $\mathcal{G}(M, K, H, \delta')$, based on Lemma G.6 and Lemma G.5, by taking union bound, with probability at least $1 - |\mathcal{C}(\varepsilon)|c_0^N - \delta' - MHK\delta'$, we have

$$\sum_{m \in \mathcal{M}} \sum_{k=1}^{K} \sum_{h=1}^{H} \left( \mathbb{E}_{\pi^*} \left[ l_{m,h}^k(s_{m,h}, a_{m,h}) | s_{m,1} = s_{m,1}^k \right] - l_{m,h}^k \left( s_{m,h}^k, a_{m,h}^k \right) \right)$$

$$\le \sum_{m \in \mathcal{M}} \sum_{k=1}^{K} \sum_{h=1}^{H} \left( A_{\delta'} \varepsilon - l_{m,h}^k \left( s_{m,h}^k, a_{m,h}^k \right) \right)$$

$$\le HMKA_{\delta'}\varepsilon + \sum_{m \in \mathcal{M}} \sum_{k=1}^{K} \sum_{h=1}^{H} c_2 dH \left\| \phi(s_{m,h}^k, a_{m,h}^k) \right\|_{(\Lambda_{m,h}^k)^{-1}}$$

$$\le HMKA_{\delta'}\varepsilon + c_2 dH \sum_{h=1}^{H} \left( \log\left( \frac{\det(\Lambda_h^K)}{\det(\lambda \mathbf{I})} \right) + 1 \right) M\sqrt{\gamma} + 2\sqrt{MK \log\left( \frac{\det(\Lambda_h^K)}{\det(\lambda \mathbf{I})} \right)}$$

$$\le HMKA_{\delta'}\varepsilon + c_2 dH \cdot H \left( d(\log(1 + MK/d) + 1)M\sqrt{\gamma} + 2\sqrt{MKd \log(1 + MK/d)} \right).$$

The first inequality follows from Lemma G.5, the second inequality holds due to Lemma G.6, the third inequality follows from Lemma D.12, the last inequality holds due to Lemma J.2 and the fact that $\|\phi(\cdot)\|_2 \le 1$.

Here we choose $\varepsilon = dH\sqrt{d/MK}/A_{\delta'} = \widetilde{\mathcal{O}}(\sqrt{d/MK})$. Conditioned on the event $\mathcal{G}(M, K, H, \delta')$, we have

$$\sum_{m \in \mathcal{M}} \sum_{k=1}^{K} \sum_{h=1}^{H} \left( \mathbb{E}_{\pi^*} \left[ l_{m,h}^k(s_{m,h}, a_{m,h}) | s_{m,1} = s_{m,1}^k \right] - l_{m,h}^k \left( s_{m,h}^k, a_{m,h}^k \right) \right) \le \widetilde{\mathcal{O}} \left( dH^2 \left( dM\sqrt{\gamma} + \sqrt{dMK} \right) \right),$$

(G.5)

with probability at least $1 - |\mathcal{C}(\varepsilon)|c_0^N - \delta' - MHK\delta'$. Based on Lemma G.2, the event $\mathcal{G}(M, K, H, \delta')$ occurs with probability at least $1 - \delta'$. Therefore, (G.5) occurs with probability at least

$$\left(1 - \delta'\right)\left(1 - |\mathcal{C}(\varepsilon)|c_0^N - \delta' - MHK\delta'\right).$$

We set $\delta' = \delta/6(MHK + 2)$ and choose $N = \widetilde{C}\log(\delta)/\log(c_0)$ where $\widetilde{C} = \widetilde{\mathcal{O}}(d)$, then we have

$$\left(1 - \delta'\right)\left(1 - |\mathcal{C}(\varepsilon)|c_0^N - \delta' - MHK\delta'\right) \geq 1 - \delta/3.$$

**Combining Terms (i)(ii)(iii) together:** Based on (G.3), (G.4) and (G.5). By taking union bound, we get that the final regret bound for CoopTS-PHE is $\widetilde{\mathcal{O}}\left(dH^2\left(dM\sqrt{\gamma} + \sqrt{dMK}\right)\right)$ with probability at least $1 - \delta$. $\qquad\square$

# H  Proof of Supporting Lemmas in Appendix G

## H.1  Proof of Proposition G.1

*Proof.* Based on (3.5), we can calculate that

$$\widetilde{\mathbf{w}}_{m,h}^{k,n} = \left(\boldsymbol{\Lambda}_{m,h}^k\right)^{-1}\left(\sum_{(s^l, a^l, s'^l) \in U_{m,h}(k)} \left[\left(r_h(s^l, a^l) + \epsilon_h^{k,l,n}\right) + V_{m,h+1}^k(s'^l)\right]\boldsymbol{\phi}(s^l, a^l) - \lambda\boldsymbol{\xi}_h^{k,n}\right)$$

$$= \widehat{\mathbf{w}}_{m,h}^k + \left(\boldsymbol{\Lambda}_{m,h}^k\right)^{-1}\left(\sum_{(s^l, a^l, s'^l) \in U_{m,h}(k)} \epsilon_h^{k,l,n}\boldsymbol{\phi}(s^l, a^l) - \lambda\boldsymbol{\xi}_h^{k,n}\right), \qquad (\text{H.1})$$

where $\widehat{\mathbf{w}}_{m,h}^k = \left(\boldsymbol{\Lambda}_{m,h}^k\right)^{-1}\left(\sum_{(s^l, a^l, s'^l) \in U_{m,h}(k)} \left[r_h + V_{m,h+1}^k(s'^l)\right]\boldsymbol{\phi}(s^l, a^l)\right)$ is the unperturbed estimated parameter. Since $\epsilon_h^{k,l,n} \sim \mathcal{N}(0, \sigma^2)$, for $l \in [\mathcal{K}(k)]$, based on the property of Gaussian distribution, we have

$$\epsilon_h^{k,l,n}\boldsymbol{\phi}(s^l, a^l) \sim \mathcal{N}\left(0, \sigma^2\boldsymbol{\phi}(s^l, a^l)\boldsymbol{\phi}(s^l, a^l)^\top\right),$$

Since $\boldsymbol{\xi}_h^{k,n} \sim \mathcal{N}(\mathbf{0}, \sigma^2\mathbf{I})$, we can calculate the covariance matrix of the second term in (H.1),

$$\left(\boldsymbol{\Lambda}_{m,h}^k\right)^{-1}\text{Cov}\left(\sum_{(s^l, a^l, s'^l) \in U_{m,h}(k)} \epsilon_h^{k,l,n}\boldsymbol{\phi}(s^l, a^l) - \lambda\boldsymbol{\xi}_h^{k,n}\right)\left(\boldsymbol{\Lambda}_{m,h}^k\right)^{-1}$$

$$= \left(\boldsymbol{\Lambda}_{m,h}^k\right)^{-1}\sigma^2\left(\sum_{(s^l, a^l, s'^l) \in U_{m,h}(k)} \boldsymbol{\phi}(s^l, a^l)\boldsymbol{\phi}(s^l, a^l)^\top + \lambda\mathbf{I}\right)\left(\boldsymbol{\Lambda}_{m,h}^k\right)^{-1}$$

$$= \sigma^2\left(\boldsymbol{\Lambda}_{m,h}^k\right)^{-1}\boldsymbol{\Lambda}_{m,h}^k\left(\boldsymbol{\Lambda}_{m,h}^k\right)^{-1}$$

$$= \sigma^2\left(\boldsymbol{\Lambda}_{m,h}^k\right)^{-1}.$$

It is obvious that the mean of the second term in (H.1) is $\mathbf{0}$. Thus, we have

$$\boldsymbol{\zeta}_{m,h}^{k,n} = \widetilde{\mathbf{w}}_{m,h}^{k,n} - \widehat{\mathbf{w}}_{m,h}^k \sim \mathcal{N}\left(0, \sigma^2\left(\boldsymbol{\Lambda}_{m,h}^k\right)^{-1}\right).$$

This completes the proof. $\qquad\square$

## H.2  Proof of Lemma G.2

*Proof.* Recall that in Proposition G.1, we have

$$\left\{\boldsymbol{\zeta}_{m,h}^{k,n}\right\} \sim \mathcal{N}\left(\mathbf{0}, \sigma^2\left(\boldsymbol{\Lambda}_{m,h}^k\right)^{-1}\right).$$

By Lemma J.10, for fixed $n \in [N]$, with probability at least $1 - \delta$, we have

$$\left\|\boldsymbol{\zeta}_{m,h}^{k,n}\right\|_{\boldsymbol{\Lambda}_{m,h}^k} \leq c\sqrt{d\sigma^2\log(d/\delta)}.$$

By applying union bound over $N$ samples, we have

$$\mathbb{P}\left( \max_{n \in [N]} \left\| \boldsymbol{\zeta}_{m,h}^{k,n} \right\|_{\boldsymbol{\Lambda}_{m,h}^k} \le c\sqrt{d\sigma^2 \log(d/\delta)} \right) \ge 1 - N\delta.$$

Now we define $c_1 = c\sqrt{\log(dNMKH/\delta)}$, and we define the event

$$\mathcal{G}_{m,h}^k(\boldsymbol{\zeta}, \delta) \stackrel{\text{def}}{=} \left\{ \max_{n \in [N]} \left\| \boldsymbol{\zeta}_{m,h}^{k,n} \right\|_{\boldsymbol{\Lambda}_{m,h}^k} \le c_1\sigma\sqrt{d} \right\}.$$

Thus for any fixed $m$, $h$ and $k$, the event $\mathcal{G}_{m,h}^k(\boldsymbol{\zeta}, \delta)$ occurs with a probability of at least $1 - \delta/MKH$. By taking union bound over all $(m, h, k) \in \mathcal{M} \times [H] \times [K]$, we have

$$\mathbb{P}\big(\mathcal{G}(M, K, H, \delta)\big) = \mathbb{P}\left( \bigcap_{m \in \mathcal{M}} \bigcap_{k \le K} \bigcap_{h \le H} \mathcal{G}_{m,h}^k(\boldsymbol{\zeta}, \delta) \right) \ge 1 - \delta.$$

This completes the proof. $\qquad\square$

### H.3 Proof of Lemma G.3

*Proof.* Based on the result in Lemma D.4, for any $(m, h, k) \in \mathcal{M} \times [H] \times [K]$, we have

$$\left\| \widehat{\mathbf{w}}_{m,h}^k \right\| \le 2H\sqrt{Mkd/\lambda}.$$

By recalling the construction of $\boldsymbol{\Lambda}_{m,h}^k$, it is trivial to find that $\lambda_{\min}\big(\boldsymbol{\Lambda}_{m,h}^k\big) \ge \lambda$. Conditioned on the event $\mathcal{G}(M, K, H, \delta)$, we have

$$\sqrt{\lambda}\left\| \boldsymbol{\zeta}_{m,h}^{k,n} \right\| \le \left\| \boldsymbol{\zeta}_{m,h}^{k,n} \right\|_{\boldsymbol{\Lambda}_{m,h}^k} \le c_1\sigma\sqrt{d}.$$

Then by triangle inequality, for all $n \in [N]$, we obtain the upper bound

$$\left\| \widetilde{\mathbf{w}}_{m,h}^{k,n} \right\| = \left\| \widehat{\mathbf{w}}_{m,h}^k + \boldsymbol{\zeta}_{m,h}^{k,n} \right\| \le 2H\sqrt{Mkd/\lambda} + c_1\sigma\sqrt{d/\lambda}.$$

Based on the result from Lemma J.7 and Lemma J.9, we have that, for any $\varepsilon > 0$, and for all $(m, k, h) \in \mathcal{M} \times [K] \times [H]$, with probability at least $1 - \delta$, we have

$$\left\| \sum_{(s^l, a^l, s'^l) \in U_{m,h}(k)} \phi\big(s^l, a^l\big) \left[ \big(V_{m,h+1}^k - \mathbb{P}_h V_{m,h+1}^k\big)\big(s^l, a^l\big) \right] \right\|_{(\boldsymbol{\Lambda}_{m,h}^k)^{-1}}$$

$$\le \left( 4H^2 \left[ \frac{d}{2}\log\left(\frac{k+\lambda}{\lambda}\right) + d\log\left(\frac{3\big(2H\sqrt{Mkd/\lambda} + c_1\sigma\sqrt{d/\lambda}\big)}{\varepsilon}\right) + \log\frac{1}{\delta} \right] + \frac{8k^2\varepsilon^2}{\lambda} \right)^{1/2}$$

$$\le 2H \left[ \frac{d}{2}\log\left(\frac{k+\lambda}{\lambda}\right) + d\log\left(\frac{3\big(2H\sqrt{Mkd/\lambda} + c_1\sigma\sqrt{d/\lambda}\big)}{\varepsilon}\right) + \log\frac{1}{\delta} \right]^{1/2} + \frac{2\sqrt{2}k\varepsilon}{\sqrt{\lambda}}.$$

Here we set $\lambda = 1, \varepsilon = \frac{H}{2\sqrt{2}k}$, with probability at least $1 - \delta$, we have

$$\left\| \sum_{(s^l, a^l, s'^l) \in U_{m,h}(k)} \phi\big(s^l, a^l\big) \left[ \big(V_{m,h+1}^k - \mathbb{P}_h V_{m,h+1}^k\big)\big(s^l, a^l\big) \right] \right\|_{(\boldsymbol{\Lambda}_{m,h}^k)^{-1}}$$

$$\le 2H\sqrt{d} \left[ \frac{1}{2}\log(K+1) + \log\left(\frac{6\sqrt{2}K\big(2H\sqrt{MKd} + c_1\sigma\sqrt{d}\big)}{H}\right) + \log\frac{1}{\delta} \right]^{1/2} + H$$

$$\le 3H\sqrt{d}D_\delta,$$

where we define $D_\delta = \left[ \frac{1}{2}\log(K+1) + \log\left(\frac{6\sqrt{2}K(2H\sqrt{MKd} + c_1\sigma\sqrt{d})}{H}\right) + \log\frac{1}{\delta} \right]^{1/2}$. Here we finish the proof. $\qquad\square$

## H.4 Proof of Lemma G.4

*Proof.* This proof is almost same as the proof of Lemma D.8 in Appendix E.6. The only difference is the **Term (i)** in (E.10). Here based on Lemma G.6, conditioned on the event $\mathcal{G}(M, K, H, \delta)$, with probability $1 - \delta$, we have

$$\textbf{Term (i)} \leq 3H\sqrt{d}D_\delta \|\phi(s,a)\|_{(\mathbf{\Lambda}_{m,h}^k)^{-1}}.$$

Finally, conditioned on the event $\mathcal{G}(M, K, H, \delta)$, with probability $1 - \delta$, for all $(m, k, h) \in \mathcal{M} \times [K] \times [H]$ and for any $(s,a) \in \mathcal{S} \times \mathcal{A}$, we have

$$\left|\phi(s,a)^\top \widehat{\mathbf{w}}_{m,h}^k - r_h(s,a) - \mathbb{P}_h V_{m,h+1}^k(s,a)\right| \leq \left(3H\sqrt{d}D_\delta + H\sqrt{d} + \sqrt{d}\right)\|\phi(s,a)\|_{(\mathbf{\Lambda}_{m,h}^k)^{-1}}$$
$$\leq 5H\sqrt{d}D_\delta \|\phi(s,a)\|_{(\mathbf{\Lambda}_{m,k}^k)^{-1}}.$$

Here we finish the proof. $\qquad\square$

## H.5 Proof of Lemma G.5

*Proof.* Recall from Definition 4.1, we have

$$r_h(s,a) + \mathbb{P}_h V_{m,h+1}^k(s,a) = \phi(s,a)^\top \boldsymbol{\theta}_h + \phi(s,a)^\top \left\langle \boldsymbol{\mu}_h, V_{m,h+1}^k \right\rangle_\mathcal{S} \stackrel{\text{def}}{=} \phi(s,a)^\top \mathbf{w}_{m,h}^k,$$

where $\mathbf{w}_{m,h}^k = \boldsymbol{\theta}_h + \left\langle \boldsymbol{\mu}_h, V_{m,h+1}^k \right\rangle_\mathcal{S}$. Note that $\max\{\|\boldsymbol{\mu}_h(\mathcal{S})\|, \|\boldsymbol{\theta}_h\|\} \leq \sqrt{d}$ and $V_{m,h+1}^k \leq H - h \leq H$, thus we have

$$\|\mathbf{w}_{m,h}^k\| \leq \|\boldsymbol{\theta}_h\| + \|\left\langle \boldsymbol{\mu}_h, V_{m,h+1}^k \right\rangle_\mathcal{S}\|$$
$$\leq \sqrt{d} + H\sqrt{d}$$
$$\leq 2H\sqrt{d}.$$

Then we define the regression error $\Delta\mathbf{w}_{m,h}^k = \mathbf{w}_{m,h}^k - \widehat{\mathbf{w}}_{m,h}^k$. For any $(m, h, k) \in \mathcal{M} \times [H] \times [K]$ and any $(s,a) \in \mathcal{S} \times \mathcal{A}$, we have

$$l_{m,h}^k(s,a) = r_h(s,a) + \mathbb{P}_h V_{m,h+1}^k(s,a) - Q_{m,h}^k(s,a)$$
$$= r_h(s,a) + \mathbb{P}_h V_{m,h+1}^k(s,a) - \min\left\{H - h + 1, \max_{n\in[N]} \phi(s,a)^\top \left(\widehat{\mathbf{w}}_{m,h}^k + \boldsymbol{\zeta}_{m,h}^{k,n}\right)\right\}^+$$
$$\leq \max\left\{\phi(s,a)^\top \mathbf{w}_{m,h}^k - (H - h + 1), \phi(s,a)^\top \mathbf{w}_{m,h}^k - \max_{n\in[N]} \phi(s,a)^\top \left(\widehat{\mathbf{w}}_{m,h}^k + \boldsymbol{\zeta}_{m,h}^{k,n}\right)\right\}$$
$$\leq \max\left\{0, \phi(s,a)^\top \Delta\mathbf{w}_{m,h}^k - \max_{n\in[N]} \phi(s,a)^\top \boldsymbol{\zeta}_{m,h}^{k,n}\right\}, \tag{H.2}$$

where the last inequality holds because $|r_h| \leq 1$ and $V_{m,h+1}^k \leq H - h$, this indicates $r_h(s,a) + \mathbb{P}_h V_{m,h+1}^k(s,a) = \phi(s,a)^\top \mathbf{w}_{m,h}^k \leq H - h + 1$. Note that $\|\phi(s,a)\|_{(\mathbf{\Lambda}_{m,h}^k)^{-1}} \leq \sqrt{1/\lambda}\|\phi(s,a)\| \leq 1$ for all $\phi(s,a)$. Define $\mathcal{C}(\varepsilon)$ to be a $\varepsilon$-cover of $\{\phi \mid \|\phi\|_{(\mathbf{\Lambda}_{m,h}^k)^{-1}} \leq 1\}$. Based on Lemma J.8, we have $|\mathcal{C}(\varepsilon)| \leq (3/\varepsilon)^d$.

First, for any fixed $\phi(s,a) \in \mathcal{C}(\varepsilon)$, we have

$$\{\phi^\top \boldsymbol{\zeta}_{m,h}^{k,n}\} \sim \mathcal{N}\left(\mathbf{0}, \sigma^2 \|\phi\|_{(\mathbf{\Lambda}_{m,h}^k)^{-1}}^2\right).$$

Use the property of Gaussian distribution, we obtain

$$\mathbb{P}\left(\phi^\top \boldsymbol{\zeta}_{m,h}^{k,n} - \sigma\|\phi\|_{(\mathbf{\Lambda}_{m,h}^k)^{-1}} \geq 0\right) = \Phi(-1).$$

By taking union bound over $n \in [N]$, we obtain

$$\mathbb{P}\left(\max_{n\in[N]}\left\{\phi^\top \boldsymbol{\zeta}_{m,h}^{k,n} - \sigma\|\phi\|_{(\mathbf{\Lambda}_{m,h}^k)^{-1}}\right\} \geq 0\right) \geq 1 - (1 - \Phi(-1))^N = 1 - \Phi(1)^N = 1 - c_0^N.$$

By applying union bound over $\mathcal{C}(\varepsilon)$, with probability $1 - |\mathcal{C}(\varepsilon)|c_0^N$, for all $\phi \in \mathcal{C}(\varepsilon)$, we have

$$\max_{n \in [N]} \left\{ \phi^\top \zeta_{m,h}^{k,n} - \sigma \|\phi\|_{(\Lambda_{m,h}^k)^{-1}} \right\} \geq 0. \tag{H.3}$$

Then, for any $\phi = \phi(s,a)$, we can find $\phi' \in \mathcal{C}(\varepsilon)$ such that $\|\phi - \phi'\|_{(\Lambda_{m,h}^k)^{-1}} \leq \varepsilon$. Define $\Delta\phi = \phi - \phi'$, we have

$$
\begin{aligned}
\phi^\top \zeta_{m,h}^{k,n} - \phi^\top \Delta\mathbf{w}_{m,h}^k &= \phi'^\top \zeta_{m,h}^{k,n} - \phi'^\top \Delta\mathbf{w}_{m,h}^k + \Delta\phi^\top \zeta_{m,h}^{k,n} - \Delta\phi^\top \Delta\mathbf{w}_{m,h}^k \\
&\geq \phi'^\top \zeta_{m,h}^{k,n} - \|\phi'\|_{(\Lambda_{m,h}^k)^{-1}} \|\Delta\mathbf{w}_{m,h}^k\|_{\Lambda_{m,h}^k} - \|\Delta\phi\|_{(\Lambda_{m,h}^k)^{-1}} \|\zeta_{m,h}^{k,n}\|_{\Lambda_{m,h}^k} \\
&\quad - \|\Delta\phi\|_{(\Lambda_{m,h}^k)^{-1}} \|\Delta\mathbf{w}_{m,h}^k\|_{\Lambda_{m,h}^k} \\
&\geq \phi'^\top \zeta_{m,h}^{k,n} - \|\phi'\|_{(\Lambda_{m,h}^k)^{-1}} \|\Delta\mathbf{w}_{m,h}^k\|_{\Lambda_{m,h}^k} \\
&\quad - \varepsilon\big(\|\zeta_{m,h}^{k,n}\|_{\Lambda_{m,h}^k} + \|\Delta\mathbf{w}_{m,h}^k\|_{\Lambda_{m,h}^k}\big). \tag{H.4}
\end{aligned}
$$

Conditioned on the event $\mathcal{G}(M, K, H, \delta)$, we have

$$\|\zeta_{m,h}^{k,n}\|_{\Lambda_{m,h}^k} \leq c_1 \sigma \sqrt{d}.$$

For any vector $\mathbf{x} \in \mathbb{R}^d$, we have

$$
\begin{aligned}
\mathbf{x}^\top \Delta\mathbf{w}_{m,h}^k &= \mathbf{x}^\top \big(\mathbf{w}_{m,h}^k - \widehat{\mathbf{w}}_{m,h}^k\big) \\
&= \mathbf{x}^\top (\Lambda_{m,h}^k)^{-1} \left( \Lambda_{m,h}^k \mathbf{w}_{m,h}^k - \left( \sum_{(s^l, a^l, s'^l) \in U_{m,h}(k)} [r_h + V_{m,h+1}^k(s'^l)]\phi(s^l, a^l) \right) \right) \\
&= \mathbf{x}^\top (\Lambda_{m,h}^k)^{-1} \left( \sum_{l=1}^{\mathcal{K}(k)} \phi(s^l, a^l)\phi(s^l, a^l)^\top \mathbf{w}_{m,h}^k + \lambda \mathbf{w}_{m,h}^k \right. \\
&\quad \left. - \left( \sum_{l=1}^{\mathcal{K}(k)} [r_h + V_{m,h+1}^k(s'^l)]\phi(s^l, a^l) \right) \right) \\
&= \mathbf{x}^\top (\Lambda_{m,h}^k)^{-1} \left( \mathbf{w}_{m,h}^k + \left( \sum_{l=1}^{\mathcal{K}(k)} [\mathbb{P}_h V_{m,h+1}^k - V_{m,h+1}^k(s'^l)]\phi(s^l, a^l) \right) \right),
\end{aligned}
$$

where the third equality holds due to the definition of $\Lambda_{m,h}^k$. We set $\mathbf{x} = \Lambda_{m,h}^k \Delta\mathbf{w}_{m,h}^k$. By using Cauchy-Schwarz inequality, we have

$$
\begin{aligned}
\|\Delta\mathbf{w}_{m,h}^k\|_{\Lambda_{m,h}^k}^2 &= \Delta\mathbf{w}_{m,h}^{k\top} \left( \mathbf{w}_{m,h}^k + \left( \sum_{l=1}^{\mathcal{K}(k)} [\mathbb{P}_h V_{m,h+1}^k - V_{m,h+1}^k(s'^l)]\phi(s^l, a^l) \right) \right) \\
&\leq \|\Delta\mathbf{w}_{m,h}^k\|_{\Lambda_{m,h}^k} \cdot \left\| \mathbf{w}_{m,h}^k + \left( \sum_{l=1}^{\mathcal{K}(k)} [\mathbb{P}_h V_{m,h+1}^k - V_{m,h+1}^k(s'^l)]\phi(s^l, a^l) \right) \right\|_{(\Lambda_{m,h}^k)^{-1}}.
\end{aligned}
$$

This indicates that with probability at least $1 - \delta$, for all $(m, h, k) \in \mathcal{M} \times [H] \times [K]$, we have

$$
\begin{aligned}
\|\Delta\mathbf{w}_{m,h}^k\|_{\Lambda_{m,h}^k} &\leq \|\mathbf{w}_{m,h}^k\|_{(\Lambda_{m,h}^k)^{-1}} + \left\| \sum_{l=1}^{\mathcal{K}(k)} [\mathbb{P}_h V_{m,h+1}^k - V_{m,h+1}^k(s'^l)]\phi(s^l, a^l) \right\|_{(\Lambda_{m,h}^k)^{-1}} \\
&\leq \|\mathbf{w}_{m,h}^k\| + 3H\sqrt{d}D_\delta \\
&\leq 5H\sqrt{d}D_\delta,
\end{aligned}
$$

where the second inequality holds because of Lemma G.3. Then for all $(m, h, k) \in \mathcal{M} \times [H] \times [K]$, with probability at least $1 - \delta$, (H.4) becomes

$$\max_{n \in [N]} \left\{ \phi^\top \zeta_{m,h}^{k,n} - \phi^\top \Delta\mathbf{w}_{m,h}^k \right\}$$

$$\geq \max_{n \in [N]} \left\{ \boldsymbol{\phi'}^\top \boldsymbol{\zeta}_{m,h}^{k,n} - \|\boldsymbol{\phi'}\|_{(\boldsymbol{\Lambda}_{m,h}^k)^{-1}} \|\Delta \mathbf{w}_{m,h}^k\|_{\boldsymbol{\Lambda}_{m,h}^k} \right\} - \varepsilon \big(c_1 \sigma \sqrt{d} + 5H\sqrt{d}D_\delta\big).$$

Now we choose $\sigma = \widetilde{\mathcal{O}}(H\sqrt{d})$ and guarantee that $\sigma > 5H\sqrt{d}D_\delta \geq \|\Delta \mathbf{w}_{m,h}^k\|_{\boldsymbol{\Lambda}_{m,h}^k}$, this is achievable through calculation. Define $A_\delta = c_1\sigma\sqrt{d} + 5H\sqrt{d}D_\delta = \widetilde{\mathcal{O}}(Hd)$. Then, for all $(m, h, k) \in \mathcal{M} \times [H] \times [K]$, with probability at least $1 - \delta$, we have

$$\max_{n \in [N]} \left\{ \boldsymbol{\phi}^\top \boldsymbol{\zeta}_{m,h}^{k,n} - \boldsymbol{\phi}^\top \Delta \mathbf{w}_{m,h}^k \right\} \geq \max_{n \in [N]} \left\{ \boldsymbol{\phi'}^\top \boldsymbol{\zeta}_{m,h}^{k,n} - \sigma\|\boldsymbol{\phi'}\|_{(\boldsymbol{\Lambda}_{m,h}^k)^{-1}} \right\} - A_\delta \varepsilon.$$

Recall from (H.3), by taking union bound, with probability at least $1 - |\mathcal{C}(\varepsilon)|c_0^N - \delta$, for all $(m, h, k) \in \mathcal{M} \times [H] \times [K]$ and for all $(s, a) \in \mathcal{S} \times \mathcal{A}$, we have

$$\max_{n \in [N]} \left\{ \boldsymbol{\phi}^\top \boldsymbol{\zeta}_{m,h}^{k,n} - \boldsymbol{\phi}^\top \Delta \mathbf{w}_{m,h}^k \right\} \geq -A_\delta \varepsilon.$$

Finally, recall from (H.2), we have, with probability at least $1 - |\mathcal{C}(\varepsilon)|c_0^N - \delta$, for all $(m, h, k) \in \mathcal{M} \times [H] \times [K]$ and for all $(s, a) \in \mathcal{S} \times \mathcal{A}$, we have

$$l_{m,h}^k(s, a) \leq A_\delta \varepsilon.$$

This completes the proof. $\qquad\square$

## H.6    Proof of Lemma G.6

*Proof.* Recall the definition of model prediction error in Definition D.1, we get

$$-l_{m,h}^k(s,a) = Q_{m,h}^k(s,a) - r_h(s,a) - \mathbb{P}_h V_{m,h+1}^k(s,a)$$

$$= \min \left\{ \max_{n \in [N]} \boldsymbol{\phi}(s,a)^\top \big(\widehat{\mathbf{w}}_{m,h}^k + \boldsymbol{\zeta}_{m,h}^{k,n}\big), H - h + 1 \right\}^+ - r_h(s,a) - \mathbb{P}_h V_{m,h+1}^k(s,a)$$

$$\leq \max_{n \in [N]} \boldsymbol{\phi}(s,a)^\top \big(\widehat{\mathbf{w}}_{m,h}^k + \boldsymbol{\zeta}_{m,h}^{k,n}\big) - r_h(s,a) - \mathbb{P}_h V_{m,h+1}^k(s,a)$$

$$= \max_{n \in [N]} \boldsymbol{\phi}(s,a)^\top \boldsymbol{\zeta}_{m,h}^{k,n} - \big(r_h(s,a) + \mathbb{P}_h V_{m,h+1}^k(s,a) - \boldsymbol{\phi}(s,a)^\top \widehat{\mathbf{w}}_{m,h}^k\big)$$

$$\leq \left| r_h(s,a) + \mathbb{P}_h V_{m,h+1}^k(s,a) - \boldsymbol{\phi}(s,a)^\top \widehat{\mathbf{w}}_{m,h}^k \right| + \max_{n \in [N]} \left| \boldsymbol{\phi}(s,a)^\top \boldsymbol{\zeta}_{m,h}^{k,n} \right|.$$

Based on Lemma G.4, conditioned on the event $\mathcal{G}(M, K, H, \delta)$, with probability $1 - \delta$, for all $(m, k, h) \in \mathcal{M} \times [K] \times [H]$ and for any $(s, a) \in \mathcal{S} \times \mathcal{A}$, we have

$$\left| \boldsymbol{\phi}(s,a)^\top \widehat{\mathbf{w}}_{m,h}^k - r_h(s,a) - \mathbb{P}_h V_{m,h+1}^k(s,a) \right| \leq 5H\sqrt{d}D_\delta \|\boldsymbol{\phi}(s,a)\|_{(\boldsymbol{\Lambda}_{m,h}^k)^{-1}} \qquad \text{(H.5)}$$

Conditioned on the event $\mathcal{G}(M, K, H, \delta)$, for all $(m, h, k) \in \mathcal{M} \times [H] \times [K]$ and for any $(s, a) \in \mathcal{S} \times \mathcal{A}$, we have

$$\max_{n \in [N]} \left| \boldsymbol{\phi}(s,a)^\top \boldsymbol{\zeta}_{m,h}^{k,n} \right| \leq c_1 \sigma \sqrt{d} \|\boldsymbol{\phi}(s,a)\|_{(\boldsymbol{\Lambda}_{m,h}^k)^{-1}}. \qquad \text{(H.6)}$$

Combine (H.5) and (H.6), then use $\sigma$ defined in Lemma G.5. Conditioned on the event $\mathcal{G}(M, K, H, \delta)$, with probability $1 - \delta$, for all $(m, h, k) \in \mathcal{M} \times [H] \times [K]$ and for any $(s, a) \in \mathcal{S} \times \mathcal{A}$, we get

$$-l_{m,h}^k(s,a) \leq \big(5H\sqrt{d}D_\delta + c_1\sigma\sqrt{d}\big)\|\boldsymbol{\phi}(s,a)\|_{(\boldsymbol{\Lambda}_{m,h}^k)^{-1}}$$

$$\leq c_2 Hd \|\boldsymbol{\phi}(s,a)\|_{(\boldsymbol{\Lambda}_{m,h}^k)^{-1}},$$

where $c_2 = \widetilde{\mathcal{O}}(1)$. Here we completes the proof. $\qquad\square$

# I    Proof of the Regret Bound for CoopTS-PHE in Misspecified Setting

In this section, we prove the regret bound for CoopTS-PHE in the misspecified setting. The regret analysis, the essential supporting lemmas and their corresponding proofs are very similar to what we have presented in Appendix G and Appendix H. Here we mainly point out the differences of proof between these two settings.

## I.1 Supporting Lemmas

**Lemma I.1.** Let $\lambda = 1$ in Algorithm 2. Under Definition 4.8, for any fixed $0 < \delta < 1$, conditioned on the event $\mathcal{G}(M, K, H, \delta)$, with probability $1 - \delta$, for all $(m, k, h) \in \mathcal{M} \times [K] \times [H]$ and for any $(s, a) \in \mathcal{S} \times \mathcal{A}$, we have

$$\left| \phi(s, a)^\top \widehat{\mathbf{w}}_{m,h}^k - r_h(s, a) - \mathbb{P}_h V_{m,h+1}^k(s, a) \right| \leq \left( 5H\sqrt{d}D_\delta + 3H\zeta\sqrt{MKd} \right) \|\phi(s, a)\|_{(\mathbf{\Lambda}_{m,h}^k)^{-1}} + 3H\zeta,$$

(I.1)

where $D_\delta$ is defined in Lemma G.3.

*Proof of Lemma I.1.* This proof is almost same as the proof of Lemma F.2, with the only difference in bounding **Term(i)** in (F.3). Here (F.4) becomes

$$|\textbf{Term(i)}| \leq 3H\sqrt{d}D_\delta \|\phi(s, a)\|_{(\mathbf{\Lambda}_{m,h}^k)^{-1}}.$$

Finally we can get the desired result. $\qquad\square$

**Lemma I.2** (Optimism). Let $\lambda = 1$ in Algorithm 2 and set $c_0 = \Phi(1)$. Under Definition 4.8, conditioned on the event $\mathcal{G}(M, K, H, \delta)$, with probability at least $1 - |\mathcal{C}(\varepsilon)|c_0^N - \delta$ where $|\mathcal{C}(\varepsilon)| \leq (3/\varepsilon)^d$, for all $(m, k, h) \in \mathcal{M} \times [K] \times [H]$ and for all $(s, a) \in \mathcal{S} \times \mathcal{A}$, we have

$$l_{m,h}^k \leq A_\delta \varepsilon + 3H\zeta,$$

where $A_\delta = c_1 \sigma \sqrt{d} + 5H\sqrt{d}D_\delta = \widetilde{\mathcal{O}}(Hd)$.

*Proof of Lemma I.2.* This proof is similar to the proof in Appendix H.5. In the previous part, we have defined

$$\Delta_{m,1} = \mathbb{P}_{m,h} V_{m,h+1}^k(s, a) - \phi(s, a)^\top \langle \boldsymbol{\mu}_h, V_{m,h+1}^k \rangle_\mathcal{S},$$

$$\Delta_{m,2} = r_{m,h}(s, a) - \phi(s, a)^\top \boldsymbol{\theta}_h,$$

where $|\Delta_{m,1}| \leq 2H\zeta$ and $|\Delta_{m,2}| \leq \zeta$. Thus we have

$$r_{m,h}(s, a) + \mathbb{P}_{m,h} V_{m,h+1}^k(s, a) = \phi(s, a)^\top \mathbf{w}_{m,h}^k + \Delta_{m,1} + \Delta_{m,2},$$

where $\mathbf{w}_{m,h}^k = \langle \boldsymbol{\mu}_h, V_{m,h+1}^k \rangle_\mathcal{S} + \boldsymbol{\theta}_h$. Then we define $\Delta \mathbf{w}_{m,h}^k = \mathbf{w}_{m,h}^k - \widehat{\mathbf{w}}_{m,h}^k$. For any $(m, h, k) \in \mathcal{M} \times [H] \times [K]$ and any $(s, a) \in \mathcal{S} \times \mathcal{A}$, we have

$$l_{m,h}^k(s, a) = r_{m,h}(s, a) + \mathbb{P}_{m,h} V_{m,h+1}^k(s, a) - Q_{m,h}^k(s, a)$$

$$= r_{m,h}(s, a) + \mathbb{P}_{m,h} V_{m,h+1}^k(s, a) - \min \left\{ H - h + 1, \max_{n \in [N]} \phi(s, a)^\top \left( \widehat{\mathbf{w}}_{m,h}^k + \boldsymbol{\zeta}_{m,h}^{k,n} \right) \right\}^+$$

$$\leq \max \left\{ \phi(s, a)^\top \mathbf{w}_{m,h}^k - (H - h + 1), \phi(s, a)^\top \mathbf{w}_{m,h}^k - \max_{n \in [N]} \phi(s, a)^\top \left( \widehat{\mathbf{w}}_{m,h}^k + \boldsymbol{\zeta}_{m,h}^{k,n} \right) \right\}$$

$$\quad + \Delta_{m,1} + \Delta_{m,2}$$

$$\leq \max \left\{ 0, \phi(s, a)^\top \Delta \mathbf{w}_{m,h}^k - \max_{n \in [N]} \phi(s, a)^\top \boldsymbol{\zeta}_{m,h}^{k,n} \right\} + 3H\zeta.$$

(I.2)

In Appendix H.5, we have proved that with probability at least $1 - |\mathcal{C}(\varepsilon)|c_0^N - \delta$, for all $(m, h, k) \in \mathcal{M} \times [H] \times [K]$ and for all $(s, a) \in \mathcal{S} \times \mathcal{A}$, we have

$$\max_{n \in [N]} \left\{ \phi^\top \boldsymbol{\zeta}_{m,h}^{k,n} - \phi^\top \Delta \mathbf{w}_{m,h}^k \right\} \geq -A_\delta \varepsilon.$$

Substitute it into (I.2), we can get the final result. $\qquad\square$

**Lemma I.3** (Error bound). Let $\lambda = 1$ in Algorithm 2. Under Definition 4.8, for any fixed $0 < \delta < 1$, conditioned on the event $\mathcal{G}(M, K, H, \delta)$, with probability $1 - \delta$, for all $(m, k, h) \in \mathcal{M} \times [K] \times [H]$ and for any $(s, a) \in \mathcal{S} \times \mathcal{A}$, we have

$$-l_{m,h}^k(s, a) \leq \left( c_2 H d + 3H\zeta\sqrt{MKd} \right) \|\phi(s, a)\|_{(\mathbf{\Lambda}_h^k)^{-1}} + 3H\zeta,$$

where $c_2 = \widetilde{\mathcal{O}}(1)$ is same as that in Lemma G.6.

*Proof of Lemma I.3.* Similar to the proof in Appendix H.6, using (H.6) in Appendix H.6 and (I.1), we have

$$-l_{m,h}^k(s,a) \leq \left| r_h(s,a) + \mathbb{P}_h V_{m,h+1}^k(s,a) - \phi(s,a)^\top \widehat{\mathbf{w}}_{m,h}^k \right| + \max_{n \in [N]} \left| \phi(s,a)^\top \zeta_{m,h}^{k,n} \right|$$

$$\leq \left( 5H\sqrt{d}D_\delta + 3H\zeta\sqrt{MKd} + c_1\sigma\sqrt{d} \right) \|\phi(s,a)\|_{(\mathbf{\Lambda}_h^k)^{-1}} + 3H\zeta$$

$$\leq \left( c_2 Hd + 3H\zeta\sqrt{MKd} \right) \|\phi(s,a)\|_{(\mathbf{\Lambda}_h^k)^{-1}} + 3H\zeta,$$

where $c_2 = \widetilde{\mathcal{O}}(1)$ is same as that in Lemma G.6. Here we completes the proof. $\qquad\square$

## I.2   Regret Analysis

In this part, we give out the proof of Theorem 4.10, the regret bound for CoopTS-PHE in the misspecified setting.

*Proof of Theorem 4.10.* This proof is almost same as the proof in Appendix G.2. We do the same regret decomposition (G.2) and obtain the same bound for **Term (i)** (G.3) and **Term (ii)** (G.4). Next we bound **Term (iii)** with new lemmas in misspecified setting.

**Bounding Term (iii) in** (G.2)**:** conditioned on the event $\mathcal{G}(M, K, H, \delta')$, based on Lemma I.3 and Lemma I.2, by taking union bound, with probability at least $1 - |\mathcal{C}(\varepsilon)|c_0'^N - \delta' - MHK\delta'$, we have

$$\sum_{m \in \mathcal{M}} \sum_{k=1}^K \sum_{h=1}^H \left( \mathbb{E}_{\pi^*} \left[ l_{m,h}^k(s_{m,h}, a_{m,h}) | s_{m,1} = s_{m,1}^k \right] - l_{m,h}^k(s_{m,h}^k, a_{m,h}^k) \right)$$

$$\leq \sum_{m \in \mathcal{M}} \sum_{k=1}^K \sum_{h=1}^H \left( -l_{m,h}^k(s_{m,h}^k, a_{m,h}^k) + A_{\delta'}\varepsilon + 3H\zeta \right)$$

$$\leq \sum_{m \in \mathcal{M}} \sum_{k=1}^K \sum_{h=1}^H \left( \left( c_2 dH + 3H\zeta\sqrt{MKd} \right) \|\phi(s,a)\|_{(\mathbf{\Lambda}_h^k)^{-1}} + 3H\zeta + A_{\delta'}\varepsilon + 3H\zeta \right)$$

$$= HMKA_{\delta'}\varepsilon + 6H^2MK\zeta + \left( c_2 dH + 3H\zeta\sqrt{MKd} \right) \sum_{h=1}^H \sum_{m \in \mathcal{M}} \sum_{k=1}^K \left\| \phi(s_{m,h}^k, a_{m,h}^k) \right\|_{(\mathbf{\Lambda}_{m,h}^k)^{-1}}$$

$$\leq HMKA_{\delta'}\varepsilon + 6H^2MK\zeta + \left( c_2 dH + 3H\zeta\sqrt{MKd} \right)$$

$$\times \sum_{h=1}^H \left( \log\left( \frac{\det(\mathbf{\Lambda}_h^K)}{\det(\lambda \mathbf{I})} \right) + 1 \right) M\sqrt{\overline{\gamma}} + 2\sqrt{MK \log\left( \frac{\det(\mathbf{\Lambda}_h^K)}{\det(\lambda \mathbf{I})} \right)}$$

$$\leq HMKA_{\delta'}\varepsilon + 6H^2MK\zeta + \left( c_2 dH + 3H\zeta\sqrt{MKd} \right)$$

$$\times H\left( d(\log(1 + MK/d) + 1)M\sqrt{\overline{\gamma}} + 2\sqrt{MKd \log(1 + MK/d)} \right)$$

$$= \widetilde{\mathcal{O}}\left( d^{\frac{3}{2}} H^2 \sqrt{M}\left( \sqrt{dM\gamma} + \sqrt{K} \right) + dH^2 M\sqrt{K}\left( \sqrt{dM\gamma} + \sqrt{K} \right)\zeta \right).$$

The first inequality follows from Lemma I.2, the second inequality holds due to Lemma I.3, the third inequality follows from Lemma D.12, the last inequality holds due to Lemma J.2 and the fact that $\|\phi(\cdot)\|_2 \leq 1$, and again we choose $\varepsilon = dH\sqrt{d/MK}/A_{\delta'} = \widetilde{\mathcal{O}}(\sqrt{d/MK})$.

The probability calculation is same as that in Appendix G.2. By combining **Terms (i)(ii)(iii)** together, we get that the final regret bound for CoopTS-PHE in misspecified setting is

$$\text{Regret}(K) = \widetilde{\mathcal{O}}\left( d^{\frac{3}{2}} H^2 \sqrt{M}\left( \sqrt{dM\gamma} + \sqrt{K} \right) + dH^2 M\sqrt{K}\left( \sqrt{dM\gamma} + \sqrt{K} \right)\zeta \right),$$

with probability at least $1 - \delta$. Here we finish the proof. $\qquad\square$

# J Auxiliary Lemmas

**Lemma J.1.** [1, Lemma 11] Let $\{\mathbf{X}_t\}_{t=1}^{\infty}$ be a sequence in $\mathbb{R}^d$, $\mathbf{V}$ is $d \times d$ positive definite matrix and define $\bar{\mathbf{V}}_t = \mathbf{V} + \sum_{s=1}^{t} \mathbf{X}_s \mathbf{X}_s^{\top}$. Then, we have that

$$\log\left(\frac{\det(\bar{\mathbf{V}}_n)}{\det(\mathbf{V})}\right) \leq \sum_{t=1}^{n} \|\mathbf{X}_t\|_{\bar{\mathbf{V}}_{t-1}^{-1}}^2.$$

Further, if $\|\mathbf{X}_t\|_2 \leq L$ for all $t$, then

$$\sum_{t=1}^{n} \min\left\{1, \|\mathbf{X}_t\|_{\bar{\mathbf{V}}_{t-1}^{-1}}^2\right\} \leq 2\big(\log \det(\bar{\mathbf{V}}_n) - \log \det \mathbf{V}\big)$$

$$\leq 2\big(d\log\big(\big(\operatorname{trace}(\mathbf{V}) + nL^2\big)/d\big) - \log \det \mathbf{V}\big),$$

and finally, if $\lambda_{\min}(\mathbf{V}) \geq \max\big(1, L^2\big)$ then

$$\sum_{t=1}^{n} \|\mathbf{X}_t\|_{\bar{\mathbf{V}}_{t-1}^{-1}}^2 \leq 2\log\frac{\det(\bar{\mathbf{V}}_n)}{\det(\mathbf{V})}.$$

**Lemma J.2.** [1, Lemma 10] Suppose $\mathbf{X}_1, \mathbf{X}_2, \ldots, \mathbf{X}_t \in \mathbb{R}^d$ and for any $1 \leq s \leq t$, $\|\mathbf{X}_s\|_2 \leq L$. Let $\bar{\mathbf{V}}_t = \lambda\mathbf{I} + \sum_{s=1}^{t} \mathbf{X}_s \mathbf{X}_s^{\top}$ for some $\lambda > 0$. Then,

$$\det\big(\bar{\mathbf{V}}_t\big) \leq \big(\lambda + tL^2/d\big)^d.$$

**Lemma J.3.** [32, Lemma D.5] Let $\mathbf{A} \in \mathbb{R}^{d \times d}$ be a positive definite matrix where its largest eigenvalue $\lambda_{\max}(\mathbf{A}) \leq \lambda$. Let $\mathbf{x}_1, \ldots, \mathbf{x}_k$ be $k$ vectors in $\mathbb{R}^d$. Then it holds that

$$\left\|\mathbf{A}\sum_{i=1}^{k} \mathbf{x}_i\right\| \leq \sqrt{\lambda k}\left(\sum_{i=1}^{k} \|\mathbf{x}_i\|_{\mathbf{A}}^2\right)^{1/2}.$$

**Lemma J.4.** [36, Lemma D.1] Let $\mathbf{\Lambda}_t = \lambda\mathbf{I} + \sum_{i=1}^{t} \boldsymbol{\phi}_i \boldsymbol{\phi}_i^{\top}$, where $\boldsymbol{\phi}_i \in \mathbb{R}^d$ and $\lambda > 0$. Then it holds that

$$\sum_{i=1}^{t} \boldsymbol{\phi}_i^{\top}(\mathbf{\Lambda}_t)^{-1}\boldsymbol{\phi}_i \leq d.$$

**Lemma J.5.** [33, Lemma D.1] Given a multivariate normal distribution $\mathbf{X} \sim \mathcal{N}(\mathbf{0}, \mathbf{\Sigma})$, we have,

$$\mathbb{P}\left(\|\mathbf{X}\| \leq \sqrt{\frac{1}{\delta}\operatorname{tr}(\mathbf{\Sigma})}\right) \geq 1 - \delta.$$

**Lemma J.6.** [30] If $\mathbf{A}$ and $\mathbf{B}$ are positive semi-definite square matrices of the same size, then

$$0 \leq [\operatorname{tr}(\mathbf{AB})]^2 \leq \operatorname{tr}\big(\mathbf{A}^2\big)\operatorname{tr}\big(\mathbf{B}^2\big) \leq [\operatorname{tr}(\mathbf{A})]^2[\operatorname{tr}(\mathbf{B})]^2.$$

**Lemma J.7.** [36, Lemma D.4] Let $\{s_i\}_{i=1}^{\infty}$ be a stochastic process on state space $\mathcal{S}$ with corresponding filtration $\{\mathcal{F}_i\}_{i=1}^{\infty}$. Let $\{\boldsymbol{\phi}_i\}_{i=1}^{\infty}$ be an $\mathbb{R}^d$-valued stochastic process where $\boldsymbol{\phi}_i \in \mathcal{F}_{i-1}$, and $\|\boldsymbol{\phi}_i\| \leq 1$. Let $\mathbf{\Lambda}_k = \lambda\mathbf{I} + \sum_{i=1}^{k} \boldsymbol{\phi}_i \boldsymbol{\phi}_i^{\top}$. Then for any $\delta > 0$, with probability at least $1 - \delta$, for all $k \geq 0$, and any $V \in \mathcal{V}$ with $\sup_{s \in \mathcal{S}} |V(s)| \leq H$, we have

$$\left\|\sum_{i=1}^{k} \boldsymbol{\phi}_i\{V(s_i) - \mathbb{E}[V(s_i) \mid \mathcal{F}_{i-1}]\}\right\|_{\mathbf{\Lambda}_k^{-1}}^2 \leq 4H^2\left[\frac{d}{2}\log\left(\frac{k+\lambda}{\lambda}\right) + \log\frac{\mathcal{N}_\varepsilon}{\delta}\right] + \frac{8k^2\varepsilon^2}{\lambda},$$

where $\mathcal{N}_\varepsilon$ is the $\varepsilon$-covering number of $\mathcal{V}$ with respect to the distance $\operatorname{dist}(V, V') = \sup_{s \in \mathcal{S}} |V(s) - V'(s)|$.

**Lemma J.8.** [75, Covering number of Euclidean ball] For any $\varepsilon > 0$, $\mathcal{N}_\varepsilon$, the $\varepsilon$-covering number of the Euclidean ball of radius $B > 0$ in $\mathbb{R}^d$ satisfies

$$\mathcal{N}_\varepsilon \leq \left(1 + \frac{2B}{\varepsilon}\right)^d \leq \left(\frac{3B}{\varepsilon}\right)^d.$$

**Lemma J.9.** Let $\mathcal{V}$ denote a class of functions mapping from $\mathcal{S}$ to $\mathbb{R}$ with the following parametric form

$$V(\cdot) = \max_{a \in \mathcal{A}} \left\{ \min \left\{ \max_{n \in [N]} \boldsymbol{\phi}(\cdot, a)^\top \mathbf{w}^n, H - h + 1 \right\}^+ \right\},$$

where the parameter $\mathbf{w}^n$ satisifies $\|\mathbf{w}^n\| \le B$ for all $n \in [N]$ and for all $(x, a) \in \mathcal{S} \times \mathcal{A}$, we have $\|\boldsymbol{\phi}(x, a)\| \le 1$. Let $N_{\mathcal{V}, \varepsilon}$ be the $\varepsilon$-covering number of $\mathcal{V}$ with respect to the distance dist $(V, V') = \sup_{s \in \mathcal{S}} |V(s) - V'(s)|$. Then

$$\mathcal{N}_{\mathcal{V}, \varepsilon} \le \left( \frac{3B}{\varepsilon} \right)^d.$$

*Proof.* Consider any two functions $V_1, V_2 \in \mathcal{V}$ with parameters $\{\mathbf{w}_1^n\}_{n \in [N]}$ and $\{\mathbf{w}_2^n\}_{n \in [N]}$ respectively. Then we have

$$\text{dist}(V_1, V_2) \le \sup_{s, a} \left| \max_{n \in [N]} \boldsymbol{\phi}(s, a)^\top \mathbf{w}_1^n - \max_{n \in [N]} \boldsymbol{\phi}(s, a)^\top \mathbf{w}_2^n \right|$$

$$\le \sup_{s, a} \left| \max_{n \in [N]} \left( \boldsymbol{\phi}(s, a)^\top \mathbf{w}_1^n - \boldsymbol{\phi}(s, a)^\top \mathbf{w}_2^n \right) \right|$$

$$\le \sup_{\|\boldsymbol{\phi}\| \le 1} \max_{n \in [N]} \left| \boldsymbol{\phi}^\top \mathbf{w}_1^n - \boldsymbol{\phi}^\top \mathbf{w}_2^n \right|$$

$$= \max_{n \in [N]} \sup_{\|\boldsymbol{\phi}\| \le 1} \left| \boldsymbol{\phi}^\top (\mathbf{w}_1^n - \mathbf{w}_2^n) \right|$$

$$\le \max_{n \in [N]} \sup_{\|\boldsymbol{\phi}\| \le 1} \|\boldsymbol{\phi}\| \|\mathbf{w}_1^n - \mathbf{w}_2^n\|$$

$$\le \max_{n \in [N]} \|\mathbf{w}_1^n - \mathbf{w}_2^n\|.$$

Let $\mathcal{N}_{\mathbf{w}, \varepsilon}$ denote the $\varepsilon$-covering number of $\{\mathbf{w} \in \mathbb{R}^d \mid \|\mathbf{w}\| \le B\}$. Then, Lemma J.8 implies

$$\mathcal{N}_{\mathbf{w}, \varepsilon} \le \left( 1 + \frac{2B}{\varepsilon} \right)^d \le \left( \frac{3B}{\varepsilon} \right)^d.$$

For any $V_1 \in \mathcal{V}$, we consider its corresponding parameters $\{\mathbf{w}_1^n\}_{n \in [N]}$. For any $n \in [N]$, we can find $\mathbf{w}_2^n$ such that $\|\mathbf{w}_1^n - \mathbf{w}_2^n\| \le \varepsilon$, then we can get $V_2 \in \mathcal{V}$ with parameters $\{\mathbf{w}_2^n\}_{n \in [N]}$. Then we have $\text{dist}(V_1, V_2) \le \max_{n \in [N]} \|\mathbf{w}_1^n - \mathbf{w}_2^n\| \le \varepsilon$. Thus, we have,

$$\mathcal{N}_{\mathcal{V}, \varepsilon} \le \mathcal{N}_{\mathbf{w}, \varepsilon} \le \left( 1 + \frac{2B}{\varepsilon} \right)^d \le \left( \frac{3B}{\varepsilon} \right)^d.$$

This completes the proof. $\qquad \square$

**Lemma J.10.** [3] Suppose $Z$ is a Gaussian random variable $Z \sim \mathcal{N}(\mu, \sigma^2)$, where $\sigma > 0$. For $0 \le z \le 1$, we have

$$\mathbb{P}(Z > \mu + z\sigma) \ge \frac{1}{\sqrt{8\pi}} e^{\frac{-z^2}{2}}, \quad \mathbb{P}(Z < \mu - z\sigma) \ge \frac{1}{\sqrt{8\pi}} e^{\frac{-z^2}{2}}.$$

And for $z \ge 1$, we have

$$\frac{e^{-z^2/2}}{2z\sqrt{\pi}} \le \mathbb{P}(|Z - \mu| > z\sigma) \le \frac{e^{-z^2/2}}{z\sqrt{\pi}}.$$

**Lemma J.11.** [32, Lemma D.2] Consider a $d$-dimensional multivariate normal distribution $\mathcal{N}(\mathbf{0}, A\boldsymbol{\Lambda}^{-1})$ where $A$ is a scalar. Let $\boldsymbol{\eta}_1, \boldsymbol{\eta}_2, \ldots, \boldsymbol{\eta}_N$ be $N$ independent samples from the distribution. Then for any $\delta > 0$

$$\mathbb{P} \left( \max_{j \in [M]} \|\boldsymbol{\eta}_j\|_{\boldsymbol{\Lambda}} \le c\sqrt{dA \log(d/\delta)} \right) \ge 1 - M\delta,$$

where $c$ is some absolute constant.

**Lemma J.12.** [1, Lemma 12] Let $\mathbf{A}, \mathbf{B}$ and $\mathbf{C}$ be positive semi-definite matrices such that $\mathbf{A} = \mathbf{B} + \mathbf{C}$. Then we have that

$$\sup_{\mathbf{x} \ne 0} \frac{\mathbf{x}^\top \mathbf{A} \mathbf{x}}{\mathbf{x}^\top \mathbf{B} \mathbf{x}} \le \frac{\det(\mathbf{A})}{\det(\mathbf{B})}.$$

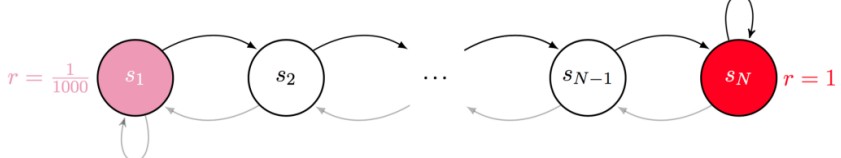

Figure 3: The N-Chain environment [62].

## K    Additional Experimental Details

We conduct comprehensive experiments investigating the exploration strategies for DQN under a multi-agent setting. For all the $Q$ networks in our experiments, we use ReLU as our activation function. Given that all experiments are conducted under multi-agent settings unless explicitly specified as a single-agent or centralized scenario, we denote our methods: CoopTS-PHE as "PHE" and CoopTS-LMC as "LMC" in experimental contexts and figures. In addition to our methods, the baselines we selected are either commonly used (DQN [57], DDQN [28]) or with competitive empirical performance (Bootstrapped DQN [62], NoisyNet DQN [26]). Both Bootstrapped DQN and NoisyNet DQN are randomized exploration methods. Bootstrapped DQN uses finite ensembles to generate the randomized value functions and views them as approximate posterior samples of $Q$-value functions. NoisyNet DQN injects noise into the parameters of neural networks to aid efficient exploration. For those figures which aim to compare among different $m$ agents within a single plot, we use **Total Episodes** to indicate the total number of training samples for a direct comparison. Note that the shaded areas on all figures represent the standard deviation.

Table 2: The swept hyper-parameters in N-Chain for PHE

| Hyper-parameter | Values |
| --- | --- |
| Learning Rate $\eta_k$ | $\{10^{-1}, 3 \times 10^{-2}, 10^{-2}, 3 \times 10^{-3}, 10^{-3}, 3 \times 10^{-4}, 10^{-4}\}$ |
| No Target Networks | $\{1, 2, 4, 8\}$ |
| Reward Noise | $\{0, 10^{-4}10^{-3}, 10^{-2}, 10^{-1}, 1.0\}$ |
| Regularization Noise | $\{0, 10^{-4}10^{-3}, 10^{-2}, 10^{-1}, 1.0\}$ |

Table 3: The swept hyper-parameters in N-Chain for LMC

| Hyper-parameter | Values |
| --- | --- |
| Learning Rate $\eta_k$ | $\{10^{-1}, 3 \times 10^{-2}, 10^{-2}, 3 \times 10^{-3}, 10^{-3}, 3 \times 10^{-4}, 10^{-4}\}$ |
| Bias Factor $\alpha$ | $\{1.0, 0.1, 0.01\}$ |
| Inverse Temperature $\beta_{m,k}$ | $\{10^0, 10^2, 10^4, 10^6, 10^8\}$ |
| No Update $J_k$ | $\{1, 4, 16, 32\}$ |

### K.1    $N$-chain

We commence by presenting the comprehensive results for $N = 25$ in Figure 4, illustrating that our randomized exploration methods exhibit greater suitability in realistic scenarios characterized by an increasing number of agents. This superiority is particularly evident under two potential circumstances: (1) where there are more limitations on computation or data access from each source in the real world, and (2) when parallel learning from multiple sources can significantly enhance runtime efficiency.

Subsequently, we provide a more comprehensive study to investigate the exploration capabilities facilitated by parallel training. Preliminary experiments are conducted with a reduced state space, specifically considering $N = 10$. The study aims to investigate exploration capabilities across varying agent counts, specifically within the set $m \in \{1, 2, 3, 4\}$.

We list the details of all swept hyper-parameters in $N$-chain for PHE and LMC in Table 2 and Table 3 respectively. Specifically, PHE is trained with reward noise $\epsilon_h^{k,l,n} = 10^{-2}$ and regularizer noise

Table 4: Hyper-parameters used in the N-chain

| Hyper-parameter | PHE | LMC | DQN | Bootstrapped DQN | Noisy DQN | DDQN |
|---|---|---|---|---|---|---|
| Discount Factor $\lambda$ | 0.99 | 0.99 | 0.99 | 0.99 | 0.99 | 0.99 |
| Learning Rate $\eta_k$ | $3 \times 10^{-2}$ | $10^{-4}$ | $3 \times 10^{-2}$ | $3 \times 10^{-2}$ | $3 \times 10^{-2}$ | $3 \times 10^{-2}$ |
| Hidden Activation | Relu | Relu | Relu | Relu | Relu | Relu |
| Output Activation | Linear | Linear | Linear | Linear | Linear | Linear |
| No Update $J_k$ | 1 | 4 | 1 | 1 | 1 | 1 |
| No Target Networks | 2 | 1 | 1 | 4 | 1 | 1 |
| Batch Size | 32 | 32 | 32 | 32 | 32 | 32 |
| NN size | $32 \times 32$ | $32 \times 32$ | $32 \times 32$ | $32 \times 32$ | $32 \times 32$ | $32 \times 32$ |

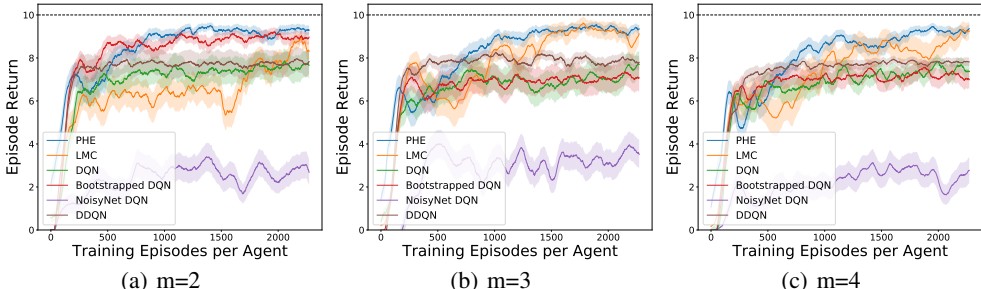

| (a) m=2 | (b) m=3 | (c) m=4 |
|---|---|---|

Figure 4: Comparison among different exploration strategies in $N$-chain with $N = 25$. All results are averaged over 10 runs.

$\boldsymbol{\xi}_h^{k,n} = 10^{-3}$ in (3.5) and LMC is trained with $\beta_{m,k} = 10^2$ and in (3.7) and optimized by Adam SGLD [33] with $\alpha_1 = 0.9$, $\alpha_2 = 0.999$ and bias factor $\alpha = 0.1$. The final hyper-parameters used in $N$-chain are presented in Table 4.

**Performance Consistency with Varying** $m$   In the investigation detailed in Figure 5, we explore parallel learning without inter-agent communication. Note that the $x$-axis implies the total training episodes from $m$ agents. Consequently, while multiple agents engage in simultaneous policy learning, each agent independently formulates its policies without the exchange of transition information. The discernible trend in this scenario is that an increase in the number of agents sharing the total episodes results in a slower rate of policy learning. Notably, despite this temporal discrepancy, all learning trajectories eventually approximate convergence towards the optimal dashed line.

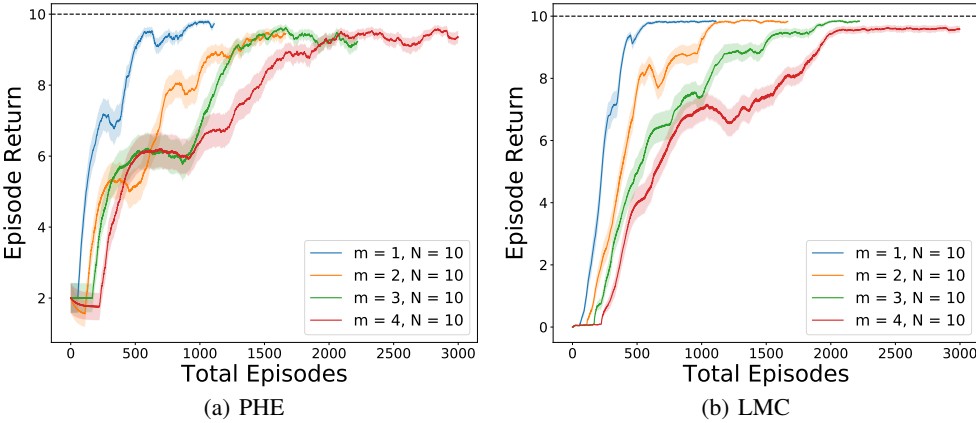

| (a) PHE | (b) LMC |
|---|---|

Figure 5: Rewards with averaged over 10 independent runs for different numbers of agents among algorithms without communication. Note that when $m = 1$, one agent indicates a centralized setting.

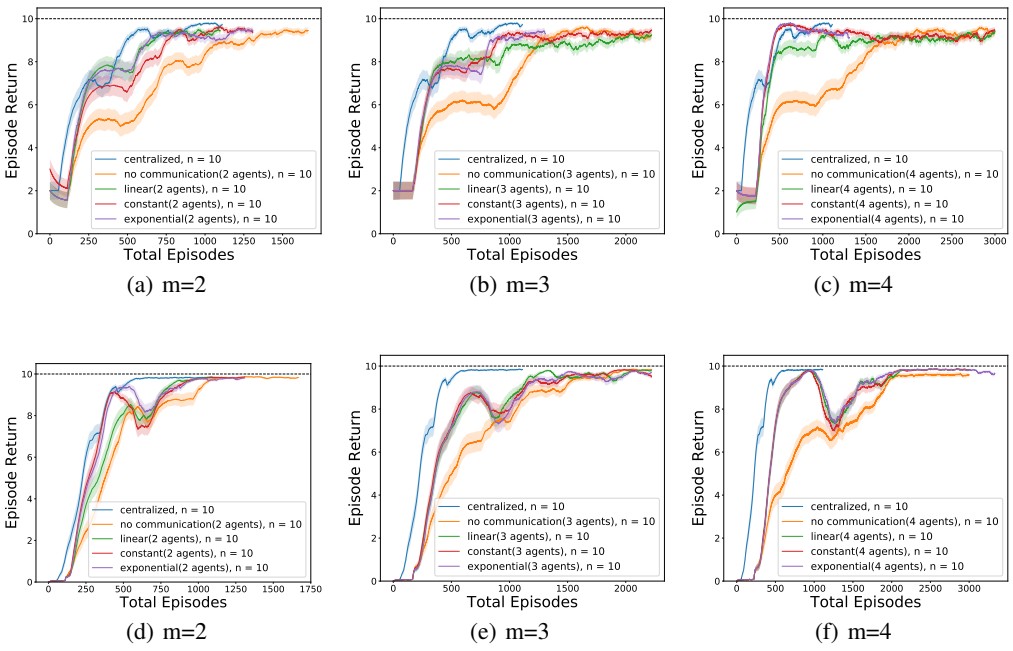

Figure 6: Different number of agents $m$ with different synchronization strategies as well as the single-agent and no communication settings in $N = 10$. **Top:** PHE, **Bottom:** LMC

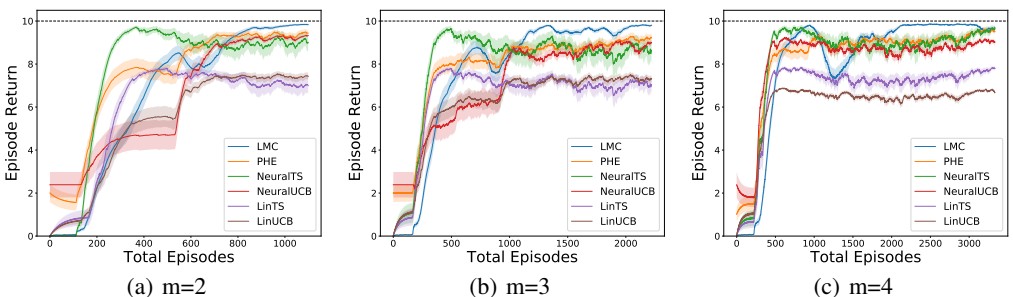

Figure 7: Performance with different number of agents $m$ compared with bandit-inspired exploration in $N = 10$.

**Different Synchronization Conditions**    To further demonstrate the efficiency of parallel learning with communication, we compare different synchronization conditions in Section 3.1. Specifically, we denote synchronization (1) in every constant step as *constant*, (2) following exponential function as *exponential*, and (3) based on (3.3) as *linear*. To have a fair comparison among different synchronization conditions, we firstly record the empirical number of synchronization via *linear* condition in average, and then we consider constant value for *constant* condition and select proper base $b$ for *exponential* condition with a similar number of synchronization. Figure 6 illustrates that any synchronization condition can improve learning efficiency but still with centralized learning as an upper bound. Note that the $x$-axis implies the total training episodes from $m$ agents.

**Performance Compared with Bandit-inspired Methods**    Since one of our proposed random exploration strategies, PHE is a variant of approximation TS, it is fair for us to investigate the performance of other exploration methods from bandit algorithms with the integration of DQN. We mainly compare both TS and UCB under neural network (*i.e.,* NeuralTS [90] and NeuralUCB [94]) and linear (*i.e.,* LinTS [5] and LinUCB [49]) settings. We show that a performance gap exists between linear approaches and other neural-based methods even in a small-scale exploration problem with $N = 10$ in Figure 7. Note that the $x$-axis implies the total training episodes from $m$ agents.

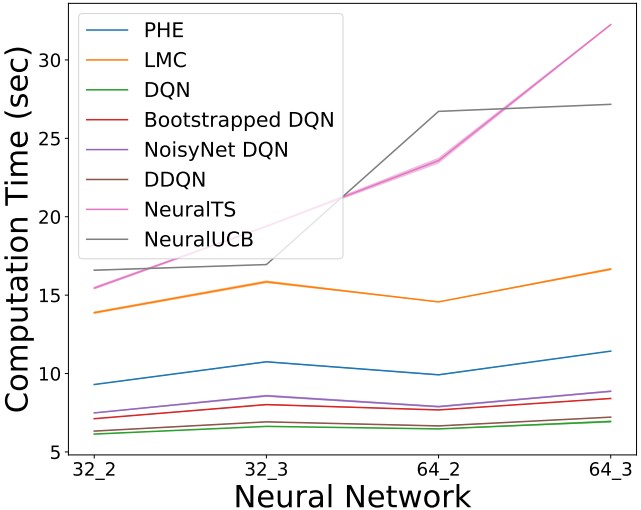

Figure 8: Computation time with different exploration strategies.

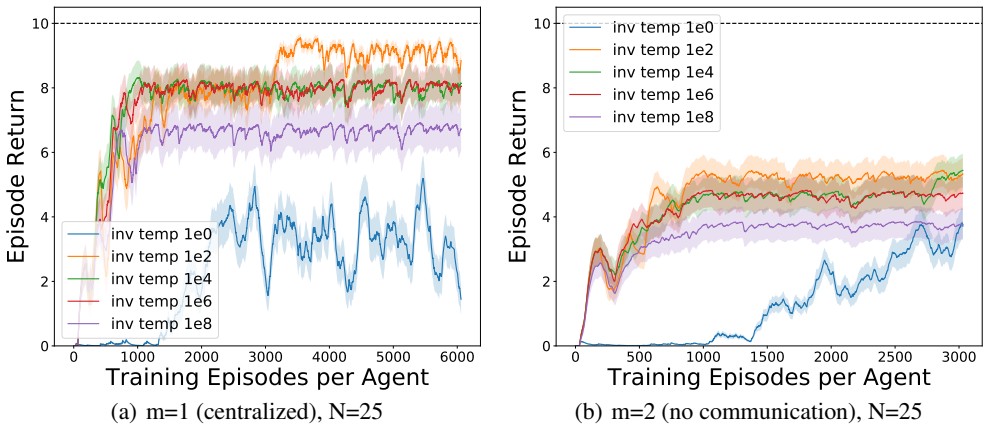

(a) m=1 (centralized), N=25

(b) m=2 (no communication), N=25

Figure 9: Hyper-parameter tuning of inverse temperature (inv temp) $\beta_{m,k}$ for LMC with $N = 25$: (a) centralized setting $m = 1$ (b) 2 agents without communication $m = 2$.

**Computational Time**   We have demonstrated that both NeuralTS and NeuralUCB exhibit convergence to performance levels comparable to our proposed randomized exploration strategies (*i.e.,* PHE and LMC) when considering the case of $N = 10$ with $m = 4$ under the synchronization condition (*linear*), as outlined in (3.3). However, we argue that the scalability of both methods is limited due to their associated computational costs. To substantiate this assertion, we conduct experiments across all methods including DQN baselines with $N = 10$ and $m = 4$ over $10^4$ steps with varying neural network sizes, such as $[32, 32, 32]$, which signifies three layers with 32 neurons in each layer. Importantly, the length of the chain $N$ has no bearing on the running time.

In Figure 8, we show the computational time of all methods under different neural network sizes. The solid lines represent the average computational time over 10 random seeds and the shaded area represents the standard deviation. We observe that NeuralTS and NeuralUCB have heavy running time consistently with varying network sizes. Although the computation time of LMC is still higher than other remaining approaches, we observe that it maintains a similar computation time with different neural network sizes, which can still be scaled up to more complex problems with larger neural networks.

**Hyper-parameter Tuning of Inverse Temperature** $\beta_{m,k}$   Subsequently, we scale the problem to $N = 25$. Given the extended horizon, the demand for exploration intensifies, leading us to conduct

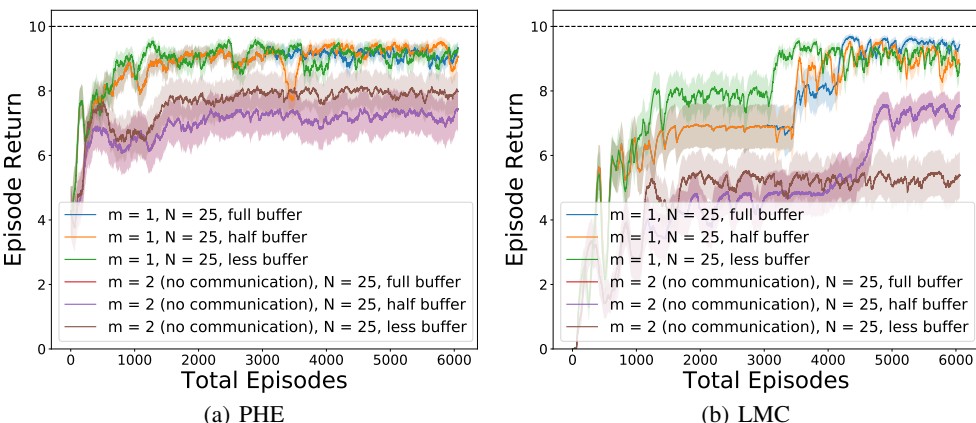

(a) PHE

(b) LMC

Figure 10: Different buffer size with $N = 25$ between single agent (centralized) and 2 agents (no communication). Note that the full buffer indicates the size of the total episodes. Each agent in no communication setting only occupies half of the total episodes. Therefore, two curves (full buffer, half buffer) in no communication are consistent.

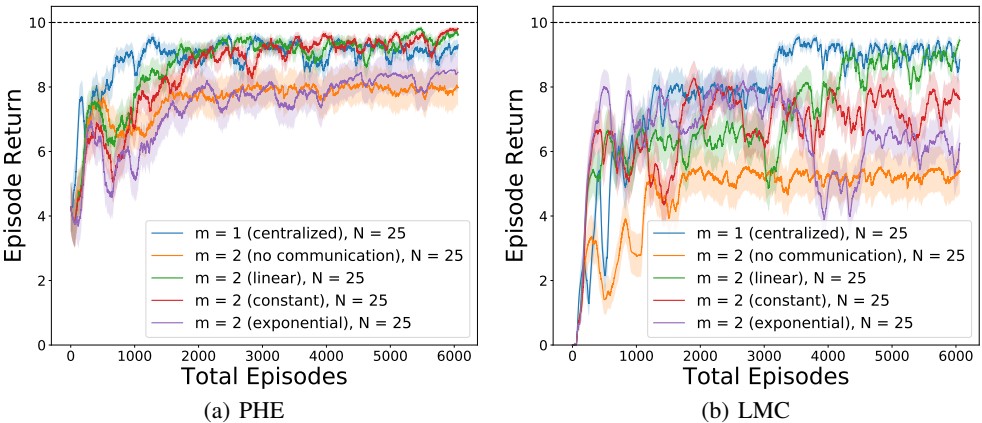

(a) PHE

(b) LMC

Figure 11: Different synchronization strategies as well as the single-agent and no communication settings in $N = 25$.

hyper-parameter tuning for the inverse temperature parameter $\beta_{m,k}$ in LMC, as illustrated in Figure 9. It is crucial to note that the efficacy of learning is significantly influenced by the exploration capacity in both centralized learning and parallel learning without communication. Our observations reveal a discernible gap between centralized and parallel learning, a departure from the pattern observed in Figure 5. We posit that the disparity may stem from issues associated with the replay buffer size in off-policy RL algorithms. Specifically, when the replay buffer exhausts its capacity for new transitions, the incoming transition replaces the oldest one.

**Hyper-parameter Tuning of Buffer Size** Therefore, we present a performance comparison between a solitary agent ($m = 1$) and a scenario involving two agents ($m = 2$) in Figure 10 with different buffer sizes. Full buffer and half buffer indicate the replay buffer's capacity to store the complete set and half of the transitions during training, respectively. We observe that the learning process is more efficient with less buffer size in a centralized setting because having an excessively large replay buffer may potentially impede the efficiency of the learning process. Furthermore, the gap between centralized setting and paralleling learning still exists among different buffer sizes. Therefore, we focus on the setting of less buffer size with different synchronization conditions in Figure 11. We conclude that *linear* condition results in competitive performance in both PHE and LMC in the $N$-chain problem and we report all exploration strategies with *linear* condition in

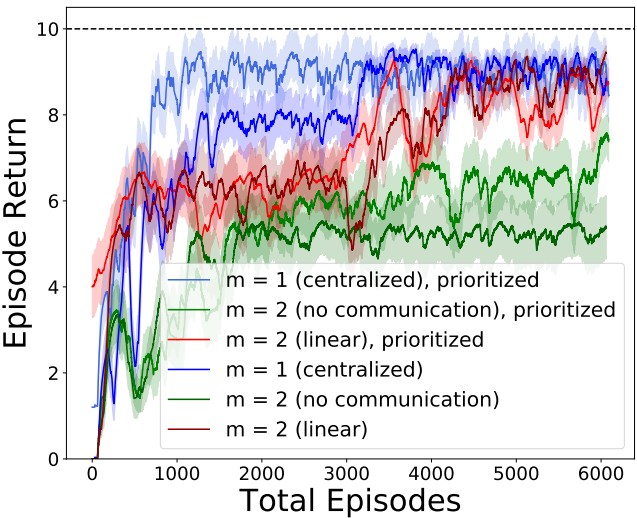

Figure 12: Gap reduction improvement with prioritized experience replay for parallel learning without communication. Note that the same settings with standard and prioritized experience replay are in the same-ish color.

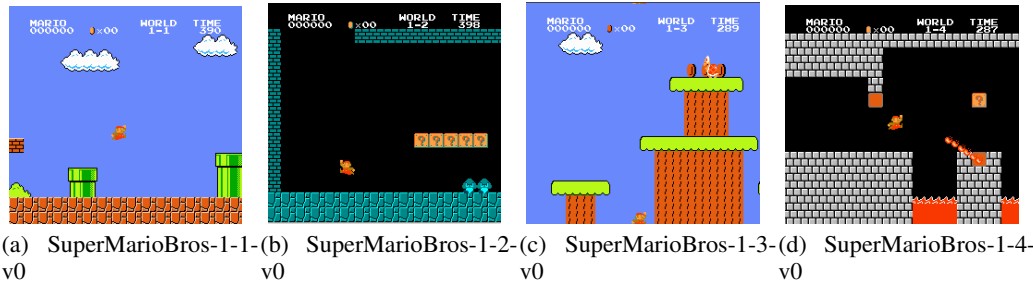

(a) SuperMarioBros-1-1-v0    (b) SuperMarioBros-1-2-v0    (c) SuperMarioBros-1-3-v0    (d) SuperMarioBros-1-4-v0

Figure 13: Illustrations of $4$ different environments in Super Mario Bros task.

Section 5.1. Note that the $x$-axis in Figure 10 and Figure 11 represent the total training episodes from $m$ agents.

**Ablation Study of Sampling Mechanism**   To reduce the reward gap, we adopt a better sampling mechanism in the replay buffer with prioritized experience replay (PER). In Figure 12, parallel learning without inter-agent communication can increase reward with PER, where the $x$-axis represents the total training episodes from $m$ agents. However, centralized learning with PER improves faster convergence with similar performance and the trends for *linear* condition curves are similar. Therefore, the gap between centralized and parallel learning without communication is reduced with PER. Note that the main experimental results in Figure 1 are based on standard experience replay because standard sampling in *linear* condition has similar performance against PER with faster training time.

### K.2 Super Mario Bros

While cooperative parallel learning enhances training efficiency through data sharing, challenges emerge when handling data from devices capturing images or audio due to privacy concerns in real-world applications. In response, our approach extends randomized exploration strategies to a federated reinforcement learning framework as shown in Algorithm 4, from Algorithm 1, which incorporates parameter synchronization among $Q$ neural networks rather than relying on the conventional practice of sharing agents' transitions in Line $14$ in Algorithm 4. Note that the synchronization follows the format as in Algorithm 1 to update Q functions with horizon $h \in H$. However, in practice, we can directly update the weight of the neural network to reduce the communication cost.

Table 5: Hyper-parameters used in the Super Mario Bros

| Hyper-parameter | PHE | LMC | DQN | Bootstrapped DQN | Noisy DQN | DDQN |
|---|---|---|---|---|---|---|
| Discount Factor $\lambda$ | 0.9 | 0.9 | 0.9 | 0.9 | 0.9 | 0.9 |
| Learning Rate $\eta_k$ | $10^{-2}$ | $3 \times 10^{-4}$ | $10^{-2}$ | $10^{-2}$ | $10^{-2}$ | $10^{-2}$ |
| Hidden Activation | Relu | Relu | Relu | Relu | Relu | Relu |
| Output Activation | Linear | Linear | Linear | Linear | Linear | Linear |
| No Update $J_k$ | 1 | 4 | 1 | 1 | 1 | 1 |
| No Target Networks | 2 | 1 | 1 | 4 | 1 | 1 |
| Batch Size | 32 | 32 | 32 | 32 | 32 | 32 |

The training process unfolds within a federated reinforcement learning framework, wherein local updates and global aggregations are iteratively executed [37]. Specifically, each agent iterates through multiple local updates of its value function, followed by server-mediated averaging of these functions across all agents, constituting a form of parameter sharing. Note that the transitions are not accessible among agents, leading us to directly synchronize all agents with parameter sharing every constant local iteration instead of synchronization condition in (3.3). We use the same architecture for all the experiments in the Super Mario Bros task with the preprocessed images as the input states and 7 discrete actions in action space.

Particularly, we construct 3 convolutional neural network layers with width [32, 64, 32], followed by 2 fully connected layers with the output of action space in the $Q$ network. The detailed hyper-parameters for Super Mario Bros task are presented in Table 5.

---

**Algorithm 4** Unified Algorithm Framework for Randomized Exploration in Federated Learning

---

1: **for** episode $k = 1, ..., K$ **do**
2:    **for** agent $m \in \mathcal{M}$ **do**
3:       Receive initial state $s_{m,1}^k$.
4:       $V_{m,H+1}^k(\cdot) \leftarrow 0$.
5:       $\{Q_{m,h}^k(\cdot, \cdot)\}_{h=1}^H \leftarrow$ **Randomized Exploration**         $\triangleleft$ Algorithm 2 or Algorithm 3
6:       **for** step $h = 1, ..., H$ **do**
7:          $a_{m,h}^k \leftarrow \mathrm{argmax}_{a \in \mathcal{A}} Q_{m,h}^k(s_{m,h}^k, a)$.
8:          Receive $s_{m,h+1}^k$ and $r_h$.
9:          **if** Condition **then**
10:             SYNCHRONIZE $\leftarrow$ True.
11:          **end if**
12:       **end for**
13:    **end for**
14:    **if** SYNCHRONIZE **then**
15:       **for** step $h = H, ..., 1$ **do**
16:          $\bar{Q}_m^k \leftarrow \frac{1}{M} \sum_{m=1}^M Q_{m,h}^k$
17:          $Q_{m,h}^k \leftarrow \bar{Q}_{m,h}^k, \forall m$
18:       **end for**
19:    **end if**
20: **end for**

---

### K.3 Thermal Control of Building Energy Systems

BuildingEnv encompasses the regulation of heat flow in a multi-zone building to sustain a desired temperature setpoint. We focus on one pre-defined building called "office small" in different cities with varying weather types, *i.e.,* Tampa (Hot Humid), Tucson (Hot Dry), Rochester (Cold Humid), and Great Falls (Cold Dry). Each episode is designed to span a single day, comprising 5-minute time intervals (H = 288, $\tau$ = 5/60 hours).

Table 6: Hyper-parameters used in the building energy systems

| Hyper-parameter | PHE | LMC | DQN | Bootstrapped DQN | Noisy DQN | DDQN |
|---|---|---|---|---|---|---|
| Discount Factor $\lambda$ | 0.99 | 0.99 | 0.99 | 0.99 | 0.99 | 0.99 |
| Learning Rate $\eta_k$ | $3 \times 10^{-3}$ | $3 \times 10^{-3}$ | $3 \times 10^{-3}$ | $3 \times 10^{-3}$ | $3 \times 10^{-3}$ | $3 \times 10^{-3}$ |
| Hidden Activation | Relu | Relu | Relu | Relu | Relu | Relu |
| Output Activation | Linear | Linear | Linear | Linear | Linear | Linear |
| No Update $J_k$ | 1 | 8 | 1 | 1 | 1 | 1 |
| No Target Networks | 2 | 1 | 1 | 4 | 1 | 1 |
| Batch Size | 32 | 32 | 32 | 32 | 32 | 32 |
| NN size | $64 \times 64$ | $64 \times 64$ | $64 \times 64$ | $64 \times 64$ | $64 \times 64$ | $64 \times 64$ |

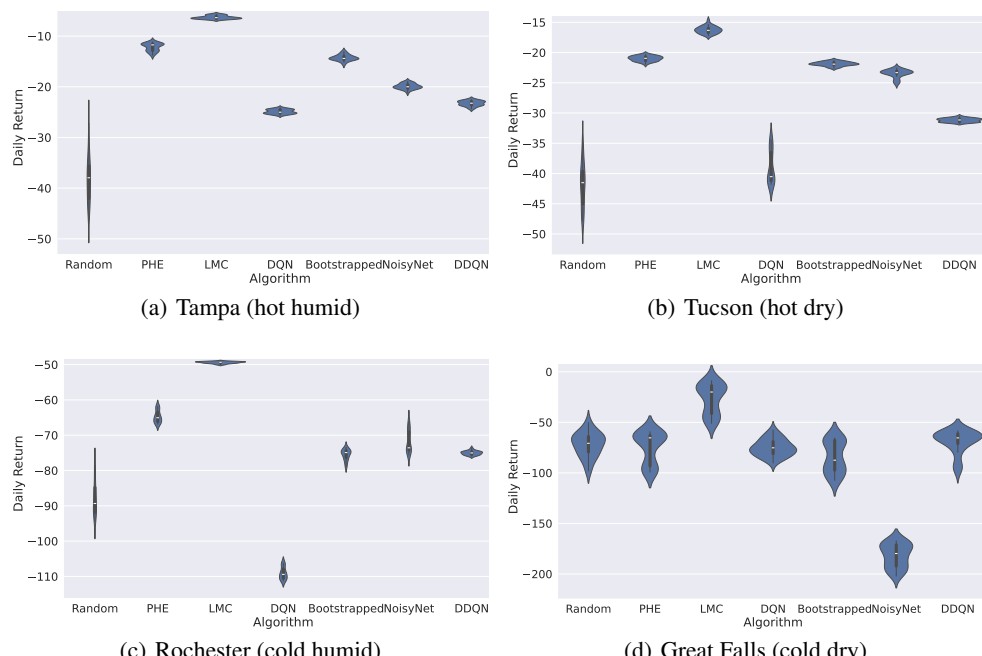

(a) Tampa (hot humid)      (b) Tucson (hot dry)

(c) Rochester (cold humid)      (d) Great Falls (cold dry)

Figure 14: Evaluation performance at different cities in building energy systems

**Observation Space** The state at time step $t$, denoted as $s(t) \in \mathbb{R}^{M+4}$, encompasses the temperatures $T_i(t)$ of each zone, where $i \in M$, along with four additional properties: $Q^{GHI}(t)$, $\bar{Q}^p(t)$, $T_G(t)$, and $T_E(t)$. Specifically, $Q^{GHI}(t)$ represents the heat gain from solar irradiance, $\bar{Q}^p(t)$ denotes the heat acquired from occupant activities, while $T_G(t)$ and $T_E(t)$ signify the ground and outdoor environment temperatures, respectively.

**Action Space** The continuous version of the action $a(t) \in [-1, 1]^M$ controls the heating of $M$ zones. However, since our randomized exploration strategies use DQN [57] as the backbone, we adopt the multi-discrete action space defined in [85], which is a vector of action spaces. Then we convert the multi-discrete action space to a single discrete action space with action mapping.

**Reward Function** The primary objective is to minimize energy consumption while ensuring the maintenance of temperature within a specified comfort range. Therefore, the reward is penalized with both temperature deviations and HVAC energy consumption as follows:

$$r(t) = -(1 - \beta)\|a(t)\|_2 - \beta\|T^{\text{target}}(t) - T(t)\|_2,$$

where $T^{\text{target}}(t) = [T_1^{\text{target}}(t), T_2^{\text{target}}(t), ..., T_M^{\text{target}}(t)]$ are the target temperatures and $T^{(}t) = [T_1(t), T_2(t), ..., T_M(t)]$ are the actual zonal temperatures. The parameter $\beta$ is the trade-off between the energy consumption and temperature deviation penalties.

We execute experiments following the united framework in Algorithm 1, synchronizing every constant number of steps across diverse weather conditions in varying cities. The hyper-parameters we used are in Table 6. Subsequently, we evaluate the performance of all methods in distinct cities respectively, as illustrated in Figure 14. Notably, our proposed random exploration strategies demonstrate a consistently higher mean return across all cities. However, it is worth highlighting that DQN in Figure 14(c) and Noisy-Net in Figure 14(d) exhibit lower returns compared to random actions. This outcome can be attributed to the discrete action space configuration [85]. In addition, we observe that maintaining thermal control of buildings is more challenging in cold weather conditions compared to hot weather conditions.

