# OpenReview forum: "Randomized Exploration in Cooperative Multi-Agent Reinforcement Learning"
_NeurIPS.cc/2024/Conference — NeurIPS 2024 poster_

### Official Review · Reviewer_WAEL · 2024-06-23

**Soundness:** 4
**Presentation:** 4
**Contribution:** 3
**Rating:** 7
**Confidence:** 4

**Summary:**

This paper presents the first result on provably efficient randomized exploration in cooperative multi-agent RL. This paper focuses on parallel MDP where the transition kernel assumes an approximately linear structure. To this end, two Thompson-sampling-type algorithms are propose which leverages the perturbed-history exploration and Langevin Monte Carlo exploration strategies, respectively. When applied to linear MDPs, both algorithms provaly achieve $\tilde{\mathcal{O}}(d^{3/2} H^2 \sqrt{MK})$ regret bound with $\tilde{\mathcal{O}} (d H M^2)$ communication complexity, where $H$ is the horizon length, $M$ is the number of agents and $d$ is the parameter dimension, marking the first-ever non-trivial theoretical results for randomized exploration in cooperative multi-agent RL. To evaluate the proposed methods, experiments on multiple parallel environments, including $N$-chain, a video game, and an energy system control problem, are conducted. The results show the effectiveness of the proposed algorithms, even under certain misspecified transitions.

**Strengths:**

**Significance**:

**1.** The paper proposed the first meaningful theoretical result for randomized exploration in multi-agent cooperative RL.

**2.** The algorithms are not only theoretically meaningful, but also easy to implement and has various advantages such as computational efficiency and avoidance of sampling bias.

**Quality**: this paper is high-quality.

**1.** The theory part of this paper is very solid.
First a unified framework is introduced which has wide applicability and can incorporate various specific settings. Then detailed application to linear MDP with theoretical guarantees is given. The paper further considers the misspecified setting where the transition kernel is slightly misspecified in a certain way, which can be common in practice. This helps extend the applicability of the proposed method.

**2.** The experimental result of this paper is extensive. Results on both video games and realistic energy control are provided.

**Clearity**: This paper is clearly written.

**1.** The theory of this paper is very clear, with the help of well-chosen terms and notations, unambiguous definitions and clear description of theorems.

**2.** A detailed table comparing all necessary related works is provided, helping readers quickly grasp the pros and cons of the proposed methods.

**3.** In section 3.1 and 3.2, a thorough description of the interpretation behind the PHE and LMC strategy is given. The rationale behind the synchronization rules is explained.

**4.** For all the experiments in section 5, the necessary implementation detail is given.

**Weaknesses:**

I do not detect any technical or major weakness in the paper. I think this paper does make a novel theoretical contribution to a meaningful problem in RL. It is also well-written.

**Questions:**

I do not have specific confusion regarding the major content of this paper since it is clearly written. Still I am interested in the following and any thoughts from the authors would be great.

This paper reports theoretical guarantees on linear MDPs. I am just wondering if randomized exploration in cooperative RL can be extended to richer MDP classes. It seems that the current algorithm design is very suitable for linear MDPs. However, the possibility and path of extending to other MDP classes is less clear.

**Limitations:**

Yes.

---

> ### Author Rebuttal · Authors · 2024-08-07
>
> Thank you for your valuable time and providing positive feedback on our work. We hope our response addresses your question.
>
> ---
> ### Q1. Extension of the algorithm beyond linear MDP setting
>
> Empirically, we would like to clarify that our algorithm is designed for general function approximation. For general MDPs, we could simply utilize a more powerful and expressive function class to approximate the value function and then directly apply our proposed algorithms since their update rules ((3.5) and (3.7) in the paper) can work with any function classes. This is in contrast with UCB and vanilla TS based algorithms [1, 2] which need to precisely compute the exploration bonus term based on the linear structure of the reward or value function.
>
> Theoretically, when the transition is a linear MDP, it is equivalent to assuming the value function is linear and thus we can apply linear function classes with our algorithms. However, if we want to extend the theoretical results to richer MDPs beyond linear MDPs, it would require us to be able to analyze the convergence of the randomized strategies to the true posterior distribution which might be non-log-concave. We suspect this could be done by following some rigorous analyses in the approximate sampling literature for non-log-concave distributions [3]. Nevertheless, these results tend to have exponential dependency on the dimension or depend on specific assumptions on the properties of the posterior distributions, which could complicate the analysis and make the regret analysis vacuous without developing dedicated techniques. Thus we leave these interesting and challenging topics for future study.
>
> ---
> We hope we have addressed all of your questions/concerns. If you have any further questions, we would be more than happy to answer them.
>
>
> ### References:
>
> [1] Chu, Wei, et al. "Contextual bandits with linear payoff functions." Proceedings of the Fourteenth International Conference on Artificial Intelligence and Statistics. JMLR Workshop and Conference Proceedings, 2011.
>
> [2] Jin, Chi, et al. "Provably efficient reinforcement learning with linear function approximation." Conference on learning theory. PMLR, 2020.
>
> [3] Dalalyan, Arnak S. "Theoretical guarantees for approximate sampling from smooth and log-concave densities." Journal of the Royal Statistical Society Series B: Statistical Methodology 79.3 (2017): 651-676.

---

> > ### Comment · Reviewer_WAEL · 2024-08-12
> >
> > Dear authors,
> >
> > Thank you for the response and insights on the extension beyond linear MDP! I have read all the reviews, and I will take all the reviews and responses into consideration during the discussion session among reviewers. I maintain my positive rating for the manuscript. I would suggest add the discussion on the extension beyond linear MDP and the technical novelty during the paper revision.
> >
> > Best,
> > Reviewer

---

> > > ### Author Response · Authors · 2024-08-12
> > >
> > > Thank you very much for your further positive feedback! We will revise our paper according to your constructive reviews.
> > >
> > > Best,
> > >
> > > Authors

---

### Official Review · Reviewer_kVyU · 2024-06-25

**Soundness:** 3
**Presentation:** 3
**Contribution:** 2
**Rating:** 5
**Confidence:** 3

**Summary:**

This paper studies provably efficient randomized exploration in parallel MDPs setting. The authors consider the linear setting, and propose two Thompson Sampling-style algorithms and establish the regret guarantees and communication cost. After that, they also conduct some experiments for evaluation.

**Strengths:**

The authors contribute some new algorithms in this cooperative learning setting, and derive their regret bounds. The paper writing is also easy to follow. The experiments results are provided to verify the performance of algorithms.

**Weaknesses:**

1. I'm not convinced that the setting considered in this paper should be called "multi-agent RL". I think it would be better to regard it as a multi-task RL setting (or maybe consider the terminology the authors used in paper: the "parallel MDP setting"), because the transition and reward functions for each agent here does not depend on the behavior of other agents, which I believe the key feature of MARL setting is missing.

2. The comparison with previous work is not precise and misleading:

    Line 210-211: "... matches the existing best single-agent results ...", which is not true. [1] considers a more general linear function approximation seeting and its regret only have $O(d)$ dependence on the dimension. I would also suggest to include the comparison with it in Table 1.

3. The setting requires more description. It seems to me there are two objectives: regret and communication complexity. It is unclear to me how the authors decide to trade-off them. I would suggest the authors to make it more clear. For the current version, it is unclear whether the regret bounds or the communication cost are optimized to optimal (or one of them are optimal while the other is better than previous).

    Besides, it seems that the agents can feel free to decide when to synchronize with the other agents to optimize the communication cost, which seems not practical in most of the times.

4. The authors motivated the randomized exploration by pointing out that the UCB style algorithms can be computational intractable beyond the linear setting. However, the objective in Eq. (3.5) and update rule Eq.(3.7) seem not easy to be generalized beyond linear setting. Besides, all the theoretical results are also limited in the linear setting. So the motivation of this paper seems not convincing to me.


[1] Zanette et al., Learning Near Optimal Policies with Low Inherent Bellman Error

**Questions:**

1. Can you highlight a bit more about the technique novelty of the proposed methods?

2. In Page 26, line 877, can you explain in details why the inequality holds?

3. In Page 28, line 891, I didn't find the definition of $w^{1,0}, \hat{w}^{1,0}, \Lambda^1, b^1$ in Algorithm 3. Can you explain it?

**Limitations:**

N.A.

---

> ### Author Rebuttal · Authors · 2024-08-07
>
> Thank you for your valuable time and providing positive feedback on our work. We hope our response will fully address all of your points.
>
> ### Q1. Discussion on multi-task RL and multi-agent RL
>
> Our work focuses on parallel MDPs, which have been categorized as multi-agent RL in the literature [1, 2, 4, 5]. While there are similarities with multi-task RL, especially in our extension to handle heterogeneity [6], our primary goal is to accelerate learning of individual MDPs by leveraging shared transition structures. This differs from multi-task RL, which involves solving multiple distinct tasks [7-9]. We will include relevant multi-task RL literature in the final version.
>
> ---
> ### Q2.  The comparison with previous work [3]
>
> Thank you for your suggestion! We would like to make our statement more accurate and concise. We intended to mean in line 221 that our result matches the existing best single-agent result using randomized exploration [10, 11]. We will polish our writing in our revision. [3] proposed ELEANOR, an optimistic generalization of the popular LSVI algorithm and derived the regret bound $\widetilde{O}(d H^2 \sqrt{K})$ under the same setting of ours. We will also add the comparison of [3] in Table 1 in the final version.
>
> ---
> ### Q3. Trade off between regret and communication complexity
>
> We first individually derive the regret bound (Theorem 4.2 and 4.3) and communication complexity with respect to $\gamma$. Then we choose a proper $\gamma=O(K/dM)$ to match the regret with other multi-agent results [1, 2]. Then we find our communication complexity matches the result of [2] and is better than [1]. We also further discuss the differences between synchronous and asynchronous settings in Remark 4.7.
>
> ---
> ### Q4. Concerns about synchronization
>
> We would like to clarify that our synchronization framework is a general setting and one main contribution of this work is to present how to incorporate random exploration strategies into this framework from both theoretical and experimental perspectives. Specifically, the synchronization condition in (3.3) contributes to the theoretical derivation of both regret and communication complexity. On the other hand, we mention in line 142 that we investigate three types of synchronization rules in the experiments. Although the synchronization rule from (3.3) still results in the most competitive performance, we show that other synchronization rules’ performance is also closer. We consider incorporating domain knowledge with criteria for specific tasks to constrain the synchronization in future work.
>
> ---
> ### Q5. Generalization beyond linear setting & Q6. About technical novelty of the proposed method
>
> Please refer to our **General Response** for all reviewers due to space limit.
>
> ---
> ### Q7. Proof explanation for the inequality in page 26, line 877
>
> This inequality holds because: 1. $(I+A_i)^{-1} \preccurlyeq \frac{2}{3} I$ (directly obtain from (F.2)) 2. we have set $\beta_{m, i}=\beta_K$ for all $i \in[k]$ and $m \in \mathcal{M}$ 3. $A_i^{2 J_i}(\Lambda_{m, h}^i)^{-1} = A_i^{J_i}(\Lambda_{m, h}^i)^{-1} A_i^{J_i}$ because $A_i=I-2 \eta_{m, i} \Lambda^i_{m, h}$.
>
> ---
> ### Q8. Explanation for the definition in page 28, line 891
>
> First, we would like to emphasize again that to simplify the notations in the proof for CoopTS-LMC, we eliminate the index $n$ (the multi-sampling number) before Lemma E.7 because the previous lemmas have nothing to do with multi-sampling. This has been mentioned at the beginning of the proof (line 779).
> So $w_{m, h}^{1,0}$ is $w_{m, h}^{k, j, n}$ with $k=1,j=0$ and eliminated $n$, here we initialize $w^{1,0}_{m, h}=0$.
>
> Moreover, $\widehat{w}^1_{m, h}, \Lambda^1_{m, h}, b^1_{m, h}$ is defined in line 640-642 with $k=1$.
>
> ---
> We hope we have addressed all of your questions. If you have any further questions, we would be happy to answer them and if you don’t, would you kindly consider increasing your score?
>
>
> ### References:
>
> [1] Dubey, Abhimanyu, and Alex Pentland. "Provably efficient cooperative multi-agent reinforcement learning with function approximation." arXiv preprint arXiv:2103.04972 (2021).
>
> [2] Min, Yifei, et al. "Cooperative multi-agent reinforcement learning: Asynchronous communication and linear function approximation." International Conference on Machine Learning. PMLR, 2023.
>
> [3] Zanette, Andrea, et al. "Learning near optimal policies with low inherent bellman error." International Conference on Machine Learning. PMLR, 2020.
>
> [4] Lidard, Justin, et al. “Provably Efficient Multi-Agent Reinforcement Learning with Fully Decentralized Communication.” IEEE American Control Conference (ACC), 2022.
>
> [5] Cisneros-Velarde, Pedro, et al. “One Policy is Enough: Parallel Exploration with a Single Policy is Near-Optimal for Reward-Free Reinforcement Learning.” International Conference on Artificial Intelligence and Statistics, 2023
>
> [6] Zhang, Chicheng, et al. “Provably efficient multi-task reinforcement learning with model transfer.” Advances in Neural Information Processing Systems 34 (2021)
>
> [7] Shi, Chengshuai, et al. “Provably Efficient Offline Reinforcement Learning with Perturbed Data Sources.” International Conference on Machine Learning. PMLR, 2023.
>
> [8] Sodhani, Shagun, et al. "Multi-Task Reinforcement Learning with Context-based Representations." Proceedings of the 38th International Conference on Machine
> Learning, 2021.
>
> [9] Amani, Sanae, et al . “Scaling Distributed Multi-task Reinforcement Learning with Experience Sharing.” arXiv preprint arXiv:2307.05834 (2023).
>
> [10] Ishfaq, Haque, et al. "Provable and Practical: Efficient Exploration in Reinforcement Learning via Langevin Monte Carlo." International Conference on Learning Representations, 2024.
>
> [11] Ishfaq, Haque, et al. "Randomized exploration in reinforcement learning with general value function approximation." International Conference on Machine Learning. PMLR, 2021.

---

> > ### Comment · Reviewer_kVyU · 2024-08-12
> >
> > Thanks for the detailed feedback. I do not have further questions and I will take the responses into consideration during the decision period.
> >
> > I would suggest include the discussion regarding comparison with [3], trade-off between regret and communication complexity, and clarification on the synchronization framework during the paper revision.

---

> > > ### Author Response · Authors · 2024-08-12
> > >
> > > Thank you very much for your positive feedback! We will add all these discussions to our final version.
> > >
> > > Best,
> > >
> > > Authors

---

### Official Review · Reviewer_kfpt · 2024-07-13

**Soundness:** 3
**Presentation:** 3
**Contribution:** 2
**Rating:** 5
**Confidence:** 4

**Summary:**

This paper considers randomized exploration in a multi-agent reinforcement learning setting called parallel MDPs. Two Thompson sampling-type algorithms are provided with a regret bound and communication complexity bound. The algorithms are empirically validated in multiple environments.

**Strengths:**

* The paper is well-written and easy to follow.
* Theoretical analysis is provided and it can match the performance of the state-of-the-art results with the potential to generalize to deep RL.
* Experiment validation is provided.

**Weaknesses:**

* The technical novelty and contribution seem limited. Can authors elaborate on the challenges in the analysis?

**Questions:**

* What is the tradeoff between using perturbed history exploration and Langevin Monte Carlo exploration?

**Limitations:**

I didn't see potential negative societal impact of this work.

---

> ### Author Rebuttal · Authors · 2024-08-07
>
> Thank you for your valuable time and effort in providing feedback on our work. We hope our response will fully address all of your points.
>
> ---
> ### Q1. Detailed explanation about challenges in theoretical analysis.
>
> We explain the specific improvements we made in our theoretical analysis here.
>
> 1. In our theoretical analysis, compared with UCB exploration, randomized exploration encounters more challenges to prove the lemma of optimism (Lemma E.10 and Lemma H.5) and the lemma of error bound (Lemma E.9 and Lemma H.6). For UCB type of algorithms, the property that the optimistic estimated value function is larger than the optimal value function can be directly guaranteed because of the added UCB bonus term. While for randomized exploration (TS-based exploration here), our optimism lemma is to prove a negative model prediction error (defined in Definition E.1). This can not be directly guaranteed because it can only be achieved with a probability. To ensure a high probability result, we use multi-sampling (such as line 3 in Algorithm 3), which causes some difficulty in analysis.
>
> 2. The multi-agent setting and the communications from synchronization in our algorithms further increase the challenges in our analysis compared to randomized exploration in the single-agent setting [2, 3]. To upper bound the self-normalized term summation in the multi-agent setting, we prove Lemma E.12, which is a modified and refined version compared with [4].
>
> 3. One big theoretical challenge is that we find and fix a non-negligible error in the regret decomposition that previous work ignored (we discuss this in Remark 4.5). To be specific, in proofs for both CoopTS-LMC and CoopTS-PHE we use a new $\varepsilon$-covering technique to prove that the optimism lemma holds for all $(s, a) \in \mathcal{S} \times \mathcal{A}$ instead of just the state-action pairs encountered by the algorithm, which is essential for the regret analysis. This was ignored by previous works [1] that use the same regret decomposition technique in the single-agent setting. Several following works using the same regret decomposition technique also ignore this error.
>
> 4. Additionally, in Appendix C, we also provide a refined analysis of communication complexity and achieve the state-of-the-art result. This is an improvement compared with previous work [4] under the same setting. This result matches the asynchronous setting result [5] and we discuss some interesting phenomena in Remark 4.7.
>
> We hope these illustrations could show our contributions on the theoretical analysis to you more clearly.
>
> ---
> ### Q2.  The tradeoff between using perturbed history exploration and Langevin Monte Carlo exploration
>
> We discuss the comparisons between PHE and LMC in the following three aspects: algorithm design, theoretical results and experiments.
>
> 1. Algorithm design: For PHE, we add i.i.d. random Gaussian noise to perturb reward and regularizer to realize randomized exploration. This requires a large number of i.i.d. Gaussian noise when the total episode $K$, the horizon length $H$ and the multi-sampling number $N$ are large. For LMC, we do the randomized exploration by performing noisy gradient descent. This requires the convergence of the LMC to the target distribution.
>
> 2. Theoretical results: Under linear MDP setting, we notice that CoopTS-PHE (Theorem 4.2) and CoopTS-LMC (Theorem 4.4) have the same order of regret, which is mentioned in Remark 4.5. While under the misspecified setting where the transition functions $P_{m,h}$ and the reward functions $r_{m,h}$ are heterogeneous across MDPs, by comparing Theorems D.3 and D.5, we find the result of CoopTS-LMC has an extra $\sqrt{d}$ factor worse than that of CoopTS-PHE, causing the chosen $\zeta$ in CoopTS-PHE has an extra $\sqrt{d}$ order over that in CoopTS-LMC. This indicates that CoopTS-PHE has better performance tolerance for the misspecified setting (we have discussed in Remark D.6).
>
> 3. Experiments: Based on our experimental results, it is hard to say which one exactly outperforms the other one. For example, in Figure 1, we find that CoopTS-LMC performs better in Mario tasks and CoopTS-PHE performs better in N-chain tasks.
>
> ---
> We hope we have addressed all of your questions/concerns. If you have any further questions, we would be more than happy to answer them and if you don’t, would you kindly consider increasing your score?
>
>
> ### References:
>
> [1] Cai, Qi, et al. "Provably efficient exploration in policy optimization." International Conference on Machine Learning. PMLR, 2020.
>
> [2] Ishfaq, Haque, et al. "Provable and Practical: Efficient Exploration in Reinforcement Learning via Langevin Monte Carlo." International Conference on Learning Representations, 2024.
>
> [3] Ishfaq, Haque, et al. "Randomized exploration in reinforcement learning with general value function approximation." International Conference on Machine Learning. PMLR, 2021.
>
> [4] Dubey, Abhimanyu, and Alex Pentland. "Provably efficient cooperative multi-agent reinforcement learning with function approximation." arXiv preprint arXiv:2103.04972 (2021).
>
> [5] Min, Yifei, et al. "Cooperative multi-agent reinforcement learning: Asynchronous communication and linear function approximation." International Conference on Machine Learning. PMLR, 2023.

---

### Official Review · Reviewer_CHJv · 2024-07-14

**Soundness:** 3
**Presentation:** 2
**Contribution:** 3
**Rating:** 5
**Confidence:** 3

**Summary:**

This paper investigates multi-agent reinforcement learning in cooperative scenarios. The main contribution is the extension of randomized exploration methods, including perturbed-history exploration and Langevin Monte Carlo exploration, to the multi-agent cooperative setting. The authors offer a regret analysis for the linear MDP case and present empirical results to validate the proposed method.

**Strengths:**

-  Although the results of this paper are not entirely new, as they extend randomized exploration methods from the single-agent to the cooperative multi-agent setting, the authors effectively highlight the technical challenges and their contributions (line 227-236).
- The algorithm proposed in this paper have better communication complexity than the UCB-type algorithm in the synchronous setting.
-  Extensive experiments are conducted to validate the effectiveness of the proposed method, which is an advantage for a theoretically oriented paper.

**Weaknesses:**

- About the contenders in experiments: It is unclear to me how the contenders DQN, Double DQN, and others were implemented. Are they running independently in multi-agent environments, with the average reward reported? If not, since Algorithm 1 provides a unified framework for parallel MDPs, the empirical comparison might be more complete if other contenders are also equipped with the synchronized steps. Besides, It is unclear why the performance of Bootstrapped DQN deteriorated significantly in the $N$-chain problem.

- Discrepancy between theory and empirical results: As shown by Theorem 4.3 and Theorem 4.4, the average performance of the proposed method improves with the growth of $M$. However, in Figure 1 and Figure 4 in the appendix, a slower convergence rate is observed when $m$ is larger. Additionally, the scaling of the x-axis in Figure 1(a) and Figure 1(b) is different, which can be misleading since the convergence rate in Figure 1(b) is actually slower than in Figure 1(a).

- Parameter setting: It appears that setting the threshold $\gamma$ requires knowledge of $K$. How should this parameter be set in practice?

Overall, this paper makes solid progress in developing randomized algorithms for cooperative multi-agent RL. However, I am concerned about the discrepancy between theory and experiment. I would raise my score if these concerns are adequately addressed.

**Questions:**

- Could you provide a more detailed explanation for the configuration of the contenders?
- Could you provide a more detailed explanation for the discrepancy between the theoretical guarantees and the experimental results?
- How should the parameter $\gamma$ be set in practice?

**Limitations:**

I do not find the negative societal impact of this work.

---

> ### Author Rebuttal · Authors · 2024-08-07
>
> Thank you for your valuable time and effort in providing detailed feedback on our work. We hope our response will fully address all of your points.
>
> ---
> ### Q1. Explanation for the configuration of the contenders
>
> All the baselines as well as our two proposed methods are run under the unified framework in Algorithm 1 for fair comparison. The only difference is how they explore. For example, the vanilla DQN follows $\epsilon$-greedy exploration strategy, but it is still equipped with the synchronized steps in your suggestion. We will emphasize this more in our final version of the manuscript. In addition to using a unified framework for all DQN baselines and our exploration strategies, the architecture for all of them remains consistent. Specifically, we detail the number of neural networks and layers for each task, as well as our hyper-parameter tuning process in Appendix Section L: Additional Experimental Details.
>
> It is generally observed that performance drops in a multi-agent parallel learning setting compared to a single-agent setting for all DQN-based approaches, including our two proposed strategies. This phenomenon is anticipated, as Theorems 4.2 and 4.4 indicate that regret increases with $\sqrt{M}$. Consequently, the single-agent performance serves as an upper bound for multi-agent parallel learning. In other words, empirical performance may decline with increasing $M$, regardless of the exploration strategy, as demonstrated by Bootstrapped DQN in this specific task. Our empirical results corroborate our theoretical findings.
>
> We would like to emphasize that the goal of our unified framework is to solve a single task efficiently when computational resources are limited. In this context, multiple agents can share the computational burden and accelerate training within our framework. CoopTS-PHE and CoopTS-LMC, combined with our synchronization conditions, facilitate efficient synchronization with minimal communication complexity.
>
>
> ---
> ### Q2. Explanation of the experimental results
>
> First, we clarify that both Theorem 4.2 (CoopTS-PHE) and Theorem 4.4 (CoopTS-LMC) show that the cumulative regret has the order of $\sqrt{M}$. Thus the average regret (defined as the cumulative regret divided by the number of agents) has the order of $\sqrt{1/M}$, meaning the average regret decreases when $M$ is larger, which indicates that the average performance improves when $M$ is larger. We would like to clarify that the plots in Figure 1 and Figure 4 in the original manuscript do not directly imply a convergence rate comparison between different numbers of agents due to the x-axis scaling.
>
> In particular, we appreciate the reviewer's observation regarding the x-axis scaling in the figures. These differing scales were intended to present comprehensive results. Specifically, our x-axis indicates the comparison from a sample efficiency perspective, which represents the total number of episodes shared among all agents, as mentioned in Line 274. Therefore, we can have a better understanding of how different RL algorithms perform with the same number of agents. To compare the convergence rate across settings with different numbers of agents, we should divide the x-axis by the number of agents involved in training. Please refer to Figure 1 in the attached PDF, where the y-axis is the average reward among all agents and the x-axis is the number of training episodes per agent. From this perspective, the convergence rate in Figure 1(b) is not slower than in Figure 1(a), especially that CoopTS-LMC has a significantly faster convergence rate when $n=3$ in Figure 1 (b). This indicates consistency with our theoretical results. We will update the plots in the final manuscript to ensure consistent scaling and enhance clarity.
>
>
> ---
> ### Q3. Practical setting of parameter $\gamma$
>
> We agree that there may be additional hyper-parameter tuning for $\gamma$ selection. There are prior works in the single-agent setting for all the experiments we conduct. Therefore, we set the initial total number of episodes among all agents $K$ based on prior works in single-agent settings. This initial setting of $K$ is used to decide our $\gamma$, followed by further fine-tuning of $K$ depending on the initial performance. We acknowledge that generalizing the synchronization rule from equation (3.3) to varying tasks may require more effort, despite its simple theoretical proof. As mentioned in Line 142, we actually investigated three types of synchronization rules in our experiments, including the rule from equation (3.3). Figure 6 demonstrates the results on the $N$-chain problem, indicating that all three synchronization rules perform consistently when $N=10$. Furthermore, Figure 11 shows a relatively better performance using the synchronization condition from equation (3.3) shown as the green curve when $N=25$. However, we recognize the practical value of the simplicity of other synchronization rules, which do not require knowledge of $K$ and result in closer performance. This flexibility highlights the practical utility of our unified framework.
>
>
> ---
> We hope we have addressed all of your questions/concerns. If you have any further questions, we would be more than happy to answer them and if you don’t, would you kindly consider increasing your score?

---

### Author Rebuttal · Authors · 2024-08-07

## General response

We would like to thank all reviewers for your insightful and detailed reviews and comments. We have addressed your comments and revised the manuscript accordingly. In the following, we would like to provide general responses to several common questions raised by reviewers.

---
### Q1. Generalization beyond linear setting

Empirically, we would like to clarify that our algorithm is designed for general function approximation. For general MDPs, we could simply utilize a more powerful and expressive function class to approximate the value function and then directly apply our proposed algorithms since their update rules ((3.5) and (3.7) in the paper) can work with any function classes. This is in contrast with UCB and vanilla TS based algorithms [6] which need to precisely compute the exploration bonus term based on the linear structure of the reward or value function.

Theoretically, when the transition is a linear MDP, it is equivalent to assuming the value function is linear and thus we can apply linear function classes with our algorithms. However, if we want to extend the theoretical results to richer MDPs beyond linear MDPs, it would require us to be able to analyze the convergence of the randomized strategies to the true posterior distribution which might be non-log-concave. We suspect this could be done by following some rigorous analyses in the approximate sampling literature for non-log-concave distributions [5]. Nevertheless, these results tend to have exponential dependency on the dimension or depend on specific assumptions on the properties of the posterior distributions, which could complicate the analysis and make the regret analysis vacuous without developing dedicated techniques. Thus we leave these interesting and challenging topics for future study.


---
### Q2. About technical novelty of the proposed method

1. Our work is the first study on provably efficient randomized exploration in cooperative multi-agent RL, with both theory and empirical evidence. No prior work has implemented provable randomized exploration in this multi-agent setting, there have been even no experiments involving UCB exploration in the multi-agent setting in previous studies.

2. We conduct extensive experiments on various benchmarks with comprehensive ablation studies, including $N$-chain that requires deep exploration, Super Mario Bros task in a misspecified setting, and a real-world problem in thermal control of building energy systems. Through empirical evaluation, we demonstrate that our proposed framework with synchronous communication scheme has a better performance and also compare different synchronization conditions (which we showed in Figure 6 and discuss in Appendix J). Our experiments also support that our randomized exploration strategies outperform existing deep $Q$ network (DQN)-based baselines. By proposing Algorithm 4, we also show that our random exploration strategies in cooperative MARL can be adapted to the existing federated RL framework when data transitions are not shared. We believe that our empirical results have enough contributions and provide solid support for relevant theoretical analysis.

3. In the linear Parallel MDPs setting, we give the first theoretical result for randomized exploration. Both our regret upper bound and communication complexity match the currently best results of UCB-type of algorithms in the same setting. Moreover, we extend our theoretical analysis to the misspecified setting where the transition and the reward functions are heterogeneous across different MDPs, which is more general than the previous work (we discuss in Remark D.2 in Appendix D). In Remark D.6, we also discuss that CoopTS-PHE has better performance tolerance than CoopTS-LMC  for the misspecified setting, which is aligned with our experimental results in Figure 1(b).

4. In the theoretical analysis, we also fixed a non-negligible error in the regret decomposition that previous work ignored (we discuss this in Remark 4.5). To be specific, in proofs for both CoopTS-LMC and CoopTS-PHE we use a new $\varepsilon$-covering technique to prove that the optimism lemma holds for all $(s, a) \in \mathcal{S} \times \mathcal{A}$ instead of just the state-action pairs encountered by the algorithm, which is essential for the regret analysis. This was ignored by previous work [1] and its follow-up work [2] that use the same regret decomposition technique. Furthermore, the multi-agent setting and the communications from synchronization in our algorithms also increase the challenges in our analysis compared to randomized exploration in the single-agent setting [2, 3]. In Appendix C, we also provide a refined analysis of communication complexity and get the currently best regret bound. This is an improvement compared with previous work [4].

### References:

[1] Cai, Qi, et al. "Provably efficient exploration in policy optimization." International Conference on Machine Learning. PMLR, 2020.

[2] Ishfaq, Haque, et al. "Provable and Practical: Efficient Exploration in Reinforcement Learning via Langevin Monte Carlo." International Conference on Learning Representations, 2024.

[3] Ishfaq, Haque, et al. "Randomized exploration in reinforcement learning with general value function approximation." International Conference on Machine Learning. PMLR, 2021.

[4] Dubey, Abhimanyu, and Alex Pentland. "Provably efficient cooperative multi-agent reinforcement learning with function approximation." arXiv preprint arXiv:2103.04972 (2021).

[5] Dalalyan, Arnak S. "Theoretical guarantees for approximate sampling from smooth and log-concave densities." Journal of the Royal Statistical Society Series B: Statistical Methodology 79.3 (2017): 651-676.

[6] Jin, Chi, et al. "Provably efficient reinforcement learning with linear function approximation." Conference on learning theory. PMLR, 2020.

---

### Decision · Program_Chairs · 2024-09-25

**Decision:**

Accept (poster)

**Comment:**

This paper addresses randomized exploration in cooperative multi-agent reinforcement learning (MARL), focusing on parallel MDPs. The authors propose two Thompson Sampling (TS)-based algorithms, CoopTS-PHE and CoopTS-LMC, and provide theoretical guarantees regarding their regret bounds and communication complexity. The empirical results are validated through experiments on several benchmark environments, including a deep exploration problem, a video game, and a real-world energy systems task. The paper claims to make a significant contribution by providing the first theoretical results for randomized exploration in cooperative MARL.

Pros & Cons:

+ Unified algorithm framework for randomized exploration in parallel MDPs.
+ Theoretical guarantees for randomized exploration in cooperative MARL.
+ Extensive experiments across various environments.
+ Generally well-written.

- Limited Novelty: proposed algorithms are somewhat straightforward extensions of existing randomized exploration methods
- The setting described in the paper could be more appropriately characterized as multi-task RL rather than multi-agent RL, as the transition and reward functions do not depend on the actions of other agents.
- The current work is heavily focused on linear MDPs, and the generalization to richer MDP classes is not well-explored.

In sum, considering the significant theoretical contributions, robust empirical validation, and the detailed clarifications provided in the rebuttal, I recommend accepting this paper.